# DiffImpute: Tabular Data Imputation With Denoising Diffusion Probabilistic Model

## Abstract

Tabular data plays a crucial role in various domains but often suffers from missing values, thereby curtailing its potential utility. Traditional imputation techniques frequently yield suboptimal results and impose substantial computational burdens, leading to inaccuracies in subsequent modeling tasks. To address these challenges, we propose `DiffImpute`, a novel Denoising Diffusion Probabilistic Model (DDPM). Specifically, `DiffImpute` is trained on complete tabular datasets, ensuring that it can produce credible imputations for missing entries without undermining the authenticity of the existing data. Innovatively, it can be applied to various settings of Missing Completely At Random (MCAR) and Missing At Random (MAR). To effectively handle the tabular features in DDPM, we tailor four tabular denoising networks, spanning MLP, ResNet, Transformer, and U-Net. We also propose `Harmonization` to enhance coherence between observed and imputed data by infusing the data back and denoising them multiple times during the sampling stage. To enable efficient inference while maintaining imputation performance, we propose a refined non-Markovian sampling process that works along with `Harmonization`. Empirical evaluations on seven diverse datasets underscore the prowess of `DiffImpute`. Specifically, when paired with the Transformer as the denoising network, it consistently outperforms its competitors, boasting an average ranking of 1.7 and the most minimal standard deviation. In contrast, the next best method lags with a ranking of 2.8 and a standard deviation of 0.9. The code is available at `https://anonymous.4open.science/r/anonymization-C1B5`.

## 1 Introduction

Tabular data, ubiquitous across domains like healthcare, finance, and customer relationship management, is foundational for data management and decision-making. However, the utility of tabular data is often compromised by missing values because most deep-learning methods can only be applied to complete datasets. Yet, missing data is common because it can stem from many factors, such as human errors, privacy issues, and the inherent complexities of data collection (Tan et al., 2013). To counter this, researchers resort to imputation methods to replace missing entries. Broadly, imputation methods are bifurcated into single and multiple imputation (Rubin, 1987). Single imputation, characterized by techniques like mean and median imputation, is simple but can introduce bias by homogenizing missing entries with singular values. This approach can lead to a misrepresentation of the genuine data distribution (Roderick J. A. Little, 2002). On the opposite spectrum, multiple imputation suggests a gamut of plausible values for missing entries, leveraging iterative methods (Raghunathan et al., 2000; Buuren et al., 2006; van Buuren & Groothuis-Oudshoorn, 2011) and deep generative models (Gondara & Wang, 2018; Nazabal et al., 2020; Ivanov et al., 2019; Richardson et al., 2020). Yet, these methods come with strings attached. Iterative methods might strain computational resources and demand robust data assumptions. Deep generative models, such as Generative Adversarial Networks (GANs) and Variation AutoEncoders (VAEs), grapple with challenges like mode collapse and posterior distribution alignment (Kingma & Welling, 2019; Goodfellow et al., 2014), which leads to suboptimal imputation performance. In light of these challenges, we propose `DiffImpute`, a Denoising Diffusion Probabilistic Model (DDPM) specifically tailored for tabular data imputation. Unlike GANs and VAEs which are confined to Missing Completely At Random (MCAR) settings (Jarrett et al., 2022), the diffusion models can be applied to more generous settings

like Missing At Random (MAR). Drawing inspiration from the principles of image inpainting (Lugmayr et al., 2022), our method first involves training the DDPM (Ho et al., 2020) on complete datasets. During inference, our method effectively replaces the missing entries within an observed dataset while preserving the integrity of the observed values. `DiffImpute` addresses mode collapse challenges observed in GAN-based approaches (Salimans et al., 2016; Goodfellow, 2015) by the stability and simplicity of our training and inference process. Additionally, `DiffImpute` improves traceability by incorporating Gaussian noise throughout the diffusion process, as opposed to the prevalent practice of zero-padding in VAE-based approaches (Mattei & Frellsen, 2019). Correspondingly, we propose a novel `Time Step Tokenizer` to embed temporal order information into the denoising network. Based on this, we explore four different denoising network architectures, including MLP, ResNet, U-Net, and Transformer, to demonstrate the improvement of incorporating time information in the imputation process. Additionally, to produce an intricately continuous data distribution, we propose `Harmonization`. Specifically, `Harmonization` meticulously aligns the synthetically generated tabular entries in data-deficient regions with the observed datasets through iterative processes of diffusion and denoising. This can further help model to learn dependencies among variables like MAR. Lastly, addressing efficiency concerns while keeping the imputation quality, our research introduces the `Impute-DDIM`. This method, inspired by the non-Markovian Denoising Diffusion Implicit Models (DDIM) (Song et al., 2022), offers a significant boost to the imputation speed, where our adaptation is laser-focused on tabular data.

Our major contributions are four-fold:

- We introduce `DiffImpute`, a method that trains a diffusion model on complete data. `DiffImpute` offers a more stable and simplified training and inference process compared to other generative approaches. Furthermore, it enables imputation for various missing mechanisms of both MCAR and MAR.

- DDPM, originally developed for image data, is adapted for tabular data by introducing the `Time Step Tokenizer` to encode temporal order information. This modification enables the customization of four tabular denoising network architectures: MLP, ResNet, Transformer, and U-Net in our experiment.

- We also introduce `Harmonization` to enhance coherence between imputed and observed data during the sampling stage.

- To accelerate the inference and keep enhanced coherence, we extend the applicability of `Harmonization` beyond consecutive time step sequences by proposing `Impute-DDIM`. This modified approach supports repetitive and condensed time step sequences during the non-Markovian sampling process (Song et al., 2022).

Correspondingly, we conduct extensive experiments on seven tabular datasets which suggest Transformer as the denoising network demonstrates faster training and inference, along with state-of-the-art performance.

## 2 RELATED WORKS

**Missing Tabular Data Imputation.** Most deep learning solutions often encounter challenges when dealing with missing data, while ensemble learning approaches tend to experience a decrease in predictive power due to the presence of missing data. Missing data originates from a myriad of sources including human error, equipment malfunction, and data loss (Tan et al., 2013) and basic single imputation methods such as mean and median imputation, while convenient, are notorious for introducing bias (Roderick J. A. Little, 2002). To tackle this, the field has advanced toward more complex imputation strategies, broadly categorized into iterative and generative methods. Iterative techniques like Multiple Imputation by Chained Equations (MICE) (van Buuren & Groothuis-Oudshoorn, 2011) and MissForest (Stekhoven & Bühlmann, 2011) harness the conditional distributions between features to iteratively estimate missing values. On the other hand, generative models like GAIN (Yoon et al., 2018) and MIWAE (Mattei & Frellsen, 2019) use deep function approximators to capture the joint probability distribution of features and impute missing values accordingly. Despite their sophistication, these approaches have limitations, including complicated optimization landscapes (Jarrett et al., 2022) and strong assumptions about data missingness patterns (Li et al., 2019; Yoon & Sull, 2020; Nazabal et al., 2020).

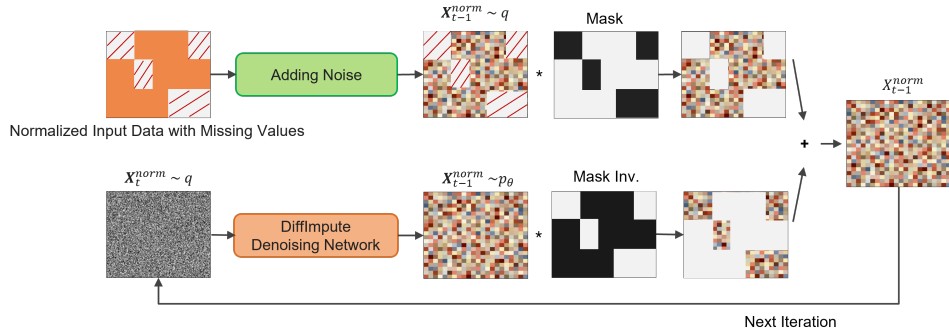

Figure 1: Schematic representation of `DiffImpute`. During inference, noisy data is extracted from known regions and supplemented with data imputed from the unknown region.

**Diffusion Models for Tabular Data.** Generative models like GANs and VAEs have carved a niche in realms such as computer vision and natural language processing (Rombach et al., 2022; Chen et al., 2022), but their foray into tabular data is still in its nascency. The reasons for this limited penetration are multifaceted, including the constrained sample sizes and the intricate task of integrating domain knowledge (Liu et al., 2023). Stepping into this milieu are diffusion models, which uniquely harness Markov chains to emulate the target distribution (Sohl-Dickstein et al., 2015; Ho et al., 2020). Their distinctive edge is twofold: the capacity to spawn high-caliber samples (Ho et al., 2020) and the simplicity and robustness of their training paradigm (Goodfellow et al., 2014; Sohl-Dickstein et al., 2015). In fact, burgeoning literature indicates that DDPMs can potentially overshadow their generative counterparts (Dhariwal & Nichol, 2021; Nichol & Dhariwal, 2021). Yet, the potential of diffusion models in the tabular data context remains under-leveraged. A handful of pioneering studies have blazed the trail, Tashiro et al. (2021) charted a course with a score-based diffusion model targeted at imputing lacunae in time series data, while Zheng & Charoenphakdee (2022) broadened this scope to envelop general tabular data imputation. Moreover, previous work (Ouyang et al., 2023) delineated an innovative score-centric approach, grounded on the gradient of the log-density score function. However, the landscape still lacks a simple but efficient denoising diffusion stratagem crafted explicitly for tabular data imputation.

## 3 METHODS

In this section, we elaborate on `DiffImpute` and unpack the four denoising network architectures correspondingly. Specifically, `DiffImpute` encompasses two stages: (1) the training of a diffusion model using complete tabular data; (2) the imputation of missing data from observed values.

### 3.1 TRAINING STAGE OF DIFFIMPUTE.

The training phase of `DiffImpute` leverages DDPM on complete tabular data, denoted as $\mathbf{x}_0 = (x_0^1, x_0^2, \cdots, x_0^k) \in \mathbb{R}^k$, where $k$ signifies the tabular data's dimensionality *i.e.,* the number of columns. Within DDPM, Gaussian noise $\epsilon$ is introduced to drive the transition from input $\mathbf{x}_0$ to distorted latent feature $\mathbf{x}_t$ across a span of $t$ time steps (Ho et al., 2020). Then, the objective during the training of `DiffImpute` is to adeptly approximate the authentic data distribution of the complete tabular set. To accomplish this, a denoising network is trained to acutely predict the noise profile $\epsilon$ that has been infused into $\mathbf{x}_t$. Specifically, we employ the smooth L1 loss function, motivated by the function's proficiency in discerning the discrepancies between the anticipated and the genuine noise (Gokcesu & Gokcesu, 2021).

### 3.2 SAMPLING STAGE OF DIFFIMPUTE.

**Missing Data Imputation.** In the sampling stage, the observed tabular data $\mathbf{x}$ is categorized into two distinct regions (Lugmayr et al., 2022). The "known region" defined by truly observed values is represented as $\mathbf{m} \odot \mathbf{x}$, where $\mathbf{m} \in \{0, 1\}^k$ is a Boolean mask pinpointing the known data with $\odot$

denoting element-wise multiplication. Conversely, the "unknown region" harbors the missing values, denoted by $(1 - \mathbf{m}) \odot \mathbf{x}$. Imputation is executed by leveraging our trained denoising network within `DiffImpute`, symbolized as $f_\theta(\mathbf{x}_t, t)$. This network focuses on the unknown region while retaining the values in the known sector, as illustrated in Fig. 1. Diving deeper, this denoising network embarks on a stepwise refinement of the "unknown region", commencing with unadulterated Gaussian noise. By tapping into the Markov Chain property of DDPM, Gaussian noise is injected at each time step $t$ to aid in sampling from the known region, $\mathbf{m} \odot \mathbf{x}$, depicted as follows:

$$\mathbf{x}_{t-1}^{\text{known}} = \sqrt{\bar{\alpha}_{t-1}} \cdot \mathbf{x}_0 + \sqrt{1 - \bar{\alpha}_{t-1}} \cdot \boldsymbol{\epsilon}, \tag{1}$$

where $\bar{\alpha}_{t-1}$ signifies the aggregate diffusion level or noise imposed on the initial input data $\mathbf{x}_0$ until time step $t - 1$, and $\boldsymbol{\epsilon} \in \mathbb{R}^k$ is drawn from a Gaussian distribution. However, for the unknown territories, the denoising network facilitates the sampling of progressively refined data with every backward step as follows:

$$\mathbf{x}_{t-1}^{\text{unknown}} = \frac{1}{\sqrt{\alpha_t}} \cdot \left( \mathbf{x_t} - \frac{1 - \alpha_t}{\sqrt{1 - \bar{\alpha}_t}} \cdot f_\theta(\mathbf{x}_t, t) \right) + \sigma_t \cdot \boldsymbol{\epsilon}, \tag{2}$$

where $\alpha_t$ represents the diffusion coefficient at time step $t$, $\sigma_t$ denotes the posterior standard deviation at time step $t$. To synthesize the imputed data, the segments $\mathbf{x}_{t-1}^{\text{known}}$ and $\mathbf{x}_{t-1}^{\text{unknown}}$ are amalgamated based on their respective masks, yielding $\mathbf{x}_{t-1}$ at the $t - 1$ time step:

$$\mathbf{x}_{t-1} = \mathbf{m} \odot \mathbf{x}_{t-1}^{\text{known}} + (1 - \mathbf{m}) \odot \mathbf{x}_{t-1}^{\text{unknown}}. \tag{3}$$

This procedure is reiterated in every reverse step until the final imputed data, $x_0$, emerges.

To further bolster the quality of our imputation, we propose `Harmonization` as a means to enhance the coherence between $\mathbf{x}_{t-1}^{\text{known}}$ and $\mathbf{x}_{t-1}^{\text{unknown}}$, thereby improving the quality of imputation. While `Harmonization` promises improved performance, extended time steps might inadvertently prolong inference runtime. To counterbalance this, we design `Impute-DDIM` to expedite the sampling process.

**Harmonization.** During the sampling of $\mathbf{x}_{t-1}^{\text{known}}$, we observed notable inconsistencies despite the model's active efforts to harmonize data at each interval (Lugmayr et al., 2022), because the current methodologies are suboptimal in leveraging the generated components from the entire dataset. To overcome this challenge and enhance the consistency during the sampling stage, we introduce `Harmonization` to retrace the output $\mathbf{x}_{t-1}$ in Eq. (3) back by one or more steps to $\mathbf{x}_{t-1+j}$ by calculating $\sqrt{\bar{\alpha}_{t-1+j}} \cdot \mathbf{x}_0 + \sqrt{1 - \bar{\alpha}_{t-1+j}} \cdot \boldsymbol{\epsilon}$, where $j \geq 1$ represents the number of steps retraced. For instance, $j = 1$ indicates a single-step retrace. It should be noted that as $j$ increases, the semantic richness of the data is amplified. However, a trade-off emerges as the run-time during the inference phase grows since the denoising network having to initiate its operation from the time step $t - 1 + j$.

**Impute-DDIM.** To accelerate the sampling stage without compromising the benefits of `Harmonization`, we introduced `Impute-DDIM`, inspired by DDIM (Song et al., 2022). Central to its merit is the capacity to sample data at a substantially condensed time step $\tau$ for $\mathbf{x}_{t-1}^{\text{unknown}}$ during inference. By honing in on the forward procedure, specifically within the subset $\mathbf{x}_{\tau 1}, \ldots, \mathbf{x}_{\tau S}$ where $S \in \{1, \ldots, T\}$, the computational weight tied to inference is appreciably reduced. Here, $\tau$ represents a sequentially increasing subset extracted from the range $\{1, \ldots, T\}$. It's worth noting that the derivation of $\mathbf{x}_{t-1}^{\text{unknown}}$ from its preceding time step $\mathbf{x}_t^{\text{unknown}}$ underwent a slight alteration:

$$\mathbf{x}_{t-1}^{\text{unknown}} = \sqrt{\alpha_{t-1}} \cdot \left( \frac{\mathbf{x}_t - \sqrt{1 - \alpha_t} f_\theta(\mathbf{x}_t^{\text{unknown}}, t)}{\sqrt{\alpha_t}} \right) + \sqrt{1 - \alpha_{t-1} - \sigma_t^2} \cdot f_\theta(\mathbf{x}_t^{\text{unknown}}, t) + \sigma_t \boldsymbol{\epsilon},$$

where $f_\theta(\mathbf{x}_t^{\text{unknown}}, t)$ refers to the predicted noise at time step for the unknown region of $\mathbf{x}$ using a trained denoising model.

**Overview.** In brief, the overall sampling process of `DiffImpute` is summarized in Alg. 1. Starting at time step $T$ and backtracking to 1, the initial step involves drawing the noise-laden observation $\mathbf{x}_{t-1}^{\text{known}}$ at time step $t - 1$. This is followed by its multiplication with the mask $\mathbf{m}$ to derive the known section. For the unknown region $(1 - \mathbf{m}) \odot \mathbf{x}$, $\mathbf{x}_{t-1}^{\text{unknown}}$ is sourced using the reverse procedure. The denoising network $f_\theta(\mathbf{x}_t, t)$ underpins this reverse modeling. Subsequently, the algorithm amalgamates the known and uncertain data facets to compute the imputed value at $t - 1$. When the `Harmonization` setting with $j = 1$ is active, a diffusion of the output $\mathbf{x}_{t-1}$ back to $\mathbf{x}_t$ is executed.

---

**Algorithm 1** Pseudo code for the sampling stage of `DiffImpute` with `Harmonization`.

---

1: **input:** Observed tabular data $\mathbf{x} \subseteq \mathbb{R}^k$, retraced step $J$, the Boolean mask for the known region $\mathbf{m}$, time step $T$, denoising network $f_\theta(\mathbf{x}_t, t)$
2: **for** $t = T, \dots, 1$ **do**                                    ▷ Loop through every time step $t$ reversely
3:    **for** $j = 1, \dots, J$ **do**                               ▷ `Harmonization` parameter: retraced steps
4:        $\boldsymbol{\epsilon} \sim \mathcal{N}(\mathbf{0}, \mathbf{I})$ if $t > 1$, else $\boldsymbol{\epsilon} = 0$                        ▷ Sampling random noise
5:        $\mathbf{x}_{t-1}^{\text{known}} = \sqrt{\bar{\alpha}_t} \cdot \mathbf{x}_0 + \sqrt{1 - \bar{\alpha}_t} \cdot \boldsymbol{\epsilon}$    ▷ Calculate the noisy observation at time step $t-1$
6:        $\mathbf{x}_{t-1}^{\text{unknown}} = \frac{1}{\sqrt{\alpha_t}} \cdot \left(\mathbf{x}_t - \frac{1-\alpha_t}{\sqrt{1-\bar{\alpha}_t}} \cdot f_\theta(\mathbf{x}_t, t)\right) + \alpha_t \cdot \boldsymbol{\epsilon}$          ▷ Sampling denoised data
7:        $\mathbf{x}_{t-1} = \mathbf{m} \cdot \mathbf{x}_t^{\text{known}} - 1 + (1 - \mathbf{m}) \cdot \mathbf{x}_{t-1}^{\text{unknown}}$ ▷ Combining known and unknown regions
8:        **if** $j < J$ and $t > 1$ **then**
9:            $\mathbf{x}_{t-1+j} = \sqrt{\bar{\alpha}_{t-1+j}} \cdot \mathbf{x}_{t-1} + \sqrt{1 - \bar{\alpha}_{t-1+j}} \cdot \boldsymbol{\epsilon}$        ▷ Diffuse $\mathbf{x}_{t-1}$ back to $\mathbf{x}_{t-1+j}$
10:        **end if**
11:    **end for**
12: **end for**
13: **return** $\mathbf{x}_0$

---

### 3.3 DENOISING NETWORK ARCHITECTURE.

To obtain a denoising network tailored specifically for tabular data, we introduce the `Time Step Tokenizer` to encode temporal information into the denoising procedure. Building upon this foundational component, we have adapted four prominent denoising network architectures: MLP, ResNet, Transformer, and U-Net, as illustrated in Fig. 2.

**Time Step Tokenizer.**   Time step tokenizer is designed to encapsulate the information of time step $t \in \mathbb{R}$, written as $\mathbf{t}_{\text{emb}} = \texttt{TimeStepTokenier}(t) \in \mathbb{R}^{2k}$. The tokenizer achieves this by formulating two distinct embeddings for scale and shift respectively, denoted as $\mathbf{t}_{\text{emb}} = \texttt{Concate}[\mathbf{t}_{\text{emb\_scale}}, \mathbf{t}_{\text{emb\_shift}}] \in \mathbb{R}^{2k}$, where `Concat` signifies the concatenation of the two tensors $\mathbf{t}_{\text{emb\_scale}}$ and $\mathbf{t}_{\text{emb\_shift}}$ along the same dimension. These learnable embeddings, $\mathbf{t}_{\text{emb\_scale}}$ and $\mathbf{t}_{\text{emb\_shift}}$, are inspired by the fixed sine and cosine transformations of $t$ (Vaswani et al., 2017), defined as:

$$
\begin{aligned}
\mathbf{t}_{\text{emb}} &= \texttt{Concat}[\mathbf{t}_{\text{emb\_scale}}, \mathbf{t}_{\text{emb\_shift}}] \\
&= \texttt{Linear}(\texttt{SiLU}(\texttt{Linear}(\texttt{GELU}(\texttt{Linear}[\mathbf{t}_{\text{scale}}, \mathbf{t}_{\text{shift}}])))), \\
\mathbf{t}_{\text{scale}} &= \sin(t \cdot \exp\left(\frac{-\log(10000)}{k} \cdot [0, 1, 2, \dots, k-1]\right)) \in \mathbb{R}^k, \\
\mathbf{t}_{\text{shift}} &= \cos(t \cdot \exp\left(\frac{-\log(10000)}{k} \cdot [0, 1, 2, \dots, k-1]\right)) \in \mathbb{R}^k,
\end{aligned}
\tag{4}
$$

where `Linear` is a learnable linear layer, `SiLU` refers to the Sigmoid Linear Unit activation (Elfwing et al., 2017), and `GeLU` applies the Gaussian Error Linear Units function (Hendrycks & Gimpel, 2023). Thus, each of the $\mathbf{t}_{\text{emb\_scale}}, \mathbf{t}_{\text{emb\_shift}}$ maintain the same dimension with $\mathbf{x}_t \in \mathbb{R}^k$. To seamlessly integrate these time step embeddings with the feature $\mathbf{x}$, we compute the update as $\mathbf{x} \cdot (\mathbf{t}_{\text{emb\_scale}} + 1) + \mathbf{t}_{\text{emb\_shift}}$, as depicted by "Add & Multiply" in Fig. 2(b).

**MLP.**   By leveraging the time step tokenizer, we can adapt the MLP (Gorishniy et al., 2021) to serve as a denoising network by incorporating $t$ as an auxiliary input. Specifically, we introduce the time embedding, $\mathbf{t}_{\text{emb}}$, derived from the time step tokenizer, into a modified block named `TimeStepMLP`. This new block is an evolution of the traditional MLP Block. The architecture of this adaptation is depicted in Fig. 2(b) and can be mathematically represented as

$$
\begin{aligned}
\texttt{MLP}(\mathbf{x}, \mathbf{t}_{\text{emb}}) &= \texttt{Linear}(\texttt{TimeStepMLP}(\dots(\texttt{TimeStepMLP}(\mathbf{x}, \mathbf{t}_{\text{emb}})))), \\
\texttt{TimeStepMLP}(\mathbf{x}, \mathbf{t}_{\text{emb}}) &= \texttt{Dropout}(\texttt{ReLU}(\texttt{Linear}(\mathbf{x}) \cdot (\mathbf{t}_{\text{emb\_scale}} + 1) + \mathbf{t}_{\text{emb\_shift}})),
\end{aligned}
\tag{5}
$$

where `Dropout` randomly zeroes some of the elements of the input tensor using samples from a Bernoulli distribution, and `ReLU` stands for the rectified linear unit function (Agarap, 2019).

**ResNet.**   Building on the foundation of the `TimeStepMLP`, we then introduce a variant of ResNet (Gorishniy et al., 2021) tailored for tabular DDPM. In this design, the `TimeStepMLP`

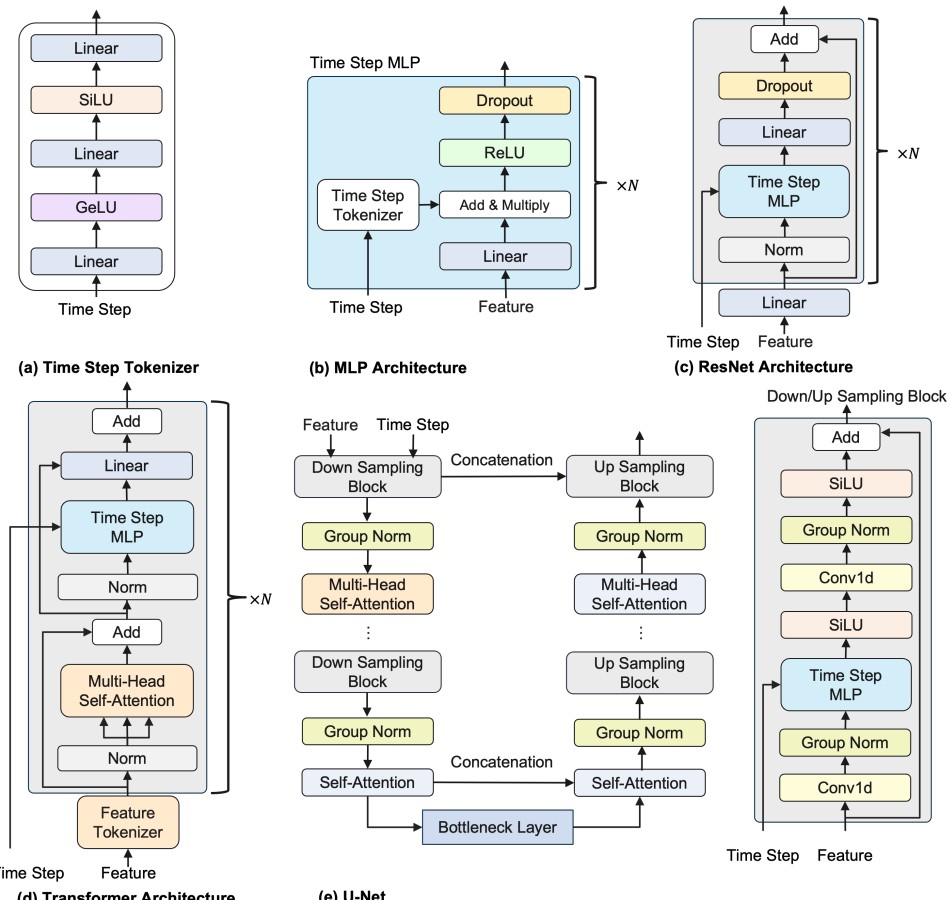

Figure 2: Four types of denoising network architecture for tabular data. (a) Time Step Tokenzier, (b) MLP; (c) ResNet; (d) Transformer; (e) U-Net.

block is seamlessly integrated into each ResNet block, as illustrated in Fig. 2(c). We hypothesize that due to the depth of its representations, this ResNet variant will outperform the MLP-based models. Formally, the representation of our ResNet architecture is:

$$\texttt{ResNet}(\mathbf{x}, \mathbf{t}_{\text{emb}}) = \texttt{Prediction}(\texttt{ResBlock}(\dots(\texttt{ResBlock}(\texttt{Linear}(\mathbf{x}), \mathbf{t}_{\text{emb}})))),$$
$$\texttt{ResBlock}(\mathbf{x}, \mathbf{t}_{\text{emb}}) = \mathbf{x} + \texttt{Dropout}(\texttt{Linear}(\texttt{TimeStepMLP}(\texttt{BatchNorm}(\mathbf{x}), \mathbf{t}_{\text{emb}}))), \quad (6)$$
$$\texttt{Prediction}(\mathbf{x}) = \texttt{Linear}(\texttt{ReLU}(\texttt{BatchNorm}(\mathbf{x}))),$$

where `BatchNorm` refers to the 1D batch normalization (Ioffe & Szegedy, 2015).

**Transformer.** To further enhance our imputation capabilities, we adapt the Transformer architecture to tailor it explicitly for the tabular domain, as shown in Fig. 2(d). The transformer processes the feature and time step embeddings through a series of sequential layers, with each layer focusing on the feature level associated with a specific time stamp, $t$. To elevate the representation of input tabular data, $\mathbf{x}$, we employ a learnable linear layer, aptly named `Feature Tokenizer` (Gorishniy et al., 2021). Then, for a given feature $\mathbf{x} = (x^1, \cdots, x^k) \in \mathbb{R}^k$, its embeddings are constructed as $\mathbf{x}_{\text{emb}}^k = \mathbf{b}^k + x^k \cdot \mathbf{W}^k \in \mathbb{R}^d$, where $\mathbf{b}^k \in \mathbb{R}^d$ is the learnable bias and $\mathbf{W}^k \in \mathbb{R}^d$ represents the learnable weight. The aggregated embeddings are then represented as $\mathbf{x}_{\text{emb}} = [\mathbf{x}_{\text{emb}}^1, \dots, \mathbf{x}_{\text{emb}}^k] \in \mathbb{R}^{k \times d}$, with $d$ being the feature embedding dimension. To capture global contexts and further enhance the model's performance on downstream tasks, we introduce the $[\textbf{CLS}] \in \mathbb{R}^d$ token (Devlin et al., 2019). This token is concatenated with the embedding matrix $\mathbf{x}_{\text{emb}}$, resulting in $\texttt{Concat}([\textbf{CLS}], \mathbf{x}_{\text{emb}}) \in \mathbb{R}^{(k+1) \times d}$. The architecture can be mathematically

described as:

$$
\begin{aligned}
\texttt{Transformer}(\mathbf{x}, \mathbf{t}_{\text{emb}}) &= \texttt{Prediction}(\texttt{TransBlock}(\dots(\texttt{TransBlock}(\\
&\quad \texttt{Concat}([\textbf{CLS}], \texttt{FeatureTokenizer}(\mathbf{x})), \mathbf{t}_{\text{emb}}))))\\
\texttt{TransBlock}(\mathbf{x}, \mathbf{t}_{\text{emb}}) &= \texttt{ResPreNorm}(\texttt{FFN}_{\mathbf{t}_{\text{emb}}}, \texttt{ResPreNorm}(\texttt{MHSA}, x)),\\
\texttt{ResPreNorm}(\texttt{Operator}, \mathbf{x}) &= \mathbf{x} + \texttt{Dropout}(\texttt{Operator}(\texttt{LayerNorm}(\mathbf{x}))),\\
\texttt{FFN}_{\mathbf{t}_{\text{emb}}}(\mathbf{x}) &= \texttt{Linear}(\texttt{TimeStepMLP}(\mathbf{x}, \mathbf{t}_{\text{emb}})),\\
\texttt{Prediction}(\mathbf{x}) &= \texttt{Linear}(\texttt{ReLU}(\texttt{LayerNorm}(\mathbf{x}))),
\end{aligned}
\tag{7}
$$

where `LayerNorm` refers to layer normalization (Ba et al., 2016), while `MHSA` denotes the Multi-Head Self-Attention layer (Vaswani et al., 2017) and we set $n_{\text{heads}} = 8$.

**U-Net.** U-Net (Ronneberger et al., 2015) has garnered significant acclaim in the domain of diffusion models. Historically, its prowess has been predominantly demonstrated in image and text sequence processing. This has inadvertently led to a dearth of U-Net architectures specifically fine-tuned for tabular data. Addressing this gap, we introduce a novel U-Net tailored for tabular data, integrating both an encoder and decoder, as illustrated in Fig. 2(e). This design uniquely amalgamates a variant of `TimeStepMLP` and self-attention mechanisms, ensuring optimal performance for tabular data. Mathematically, our U-Net is represented as:

$$
\begin{aligned}
\texttt{UNet}(\mathbf{x}, \mathbf{t}_{\text{emb}}) &= \texttt{Linear}(\texttt{DecoderBlock}(\cdots(\texttt{DecoderBlock}((\\
&\quad \texttt{BottleneckBlock}(\cdots(\texttt{EncoderBlock}(\cdots \texttt{EncoderBlock}((\mathbf{x}, \mathbf{t}_{\text{emb}})))))))))),\\
\texttt{DecoderBlock}(\mathbf{x}, \mathbf{t}_{\text{emb}}) &= \texttt{MHSA}(\texttt{ResBlock}_{\text{UNet}}(\texttt{UpsampleBlock}(\mathbf{x}, \mathbf{t}_{\text{emb}}))),\\
\texttt{EncoderBlock}(\mathbf{x}, \mathbf{t}_{\text{emb}}) &= \texttt{MHSA}(\texttt{ResBlock}_{\text{UNet}}(\texttt{DownsampleBlock}(\mathbf{x}, \mathbf{t}_{\text{emb}}))),\\
\texttt{ResBlock}_{\text{UNet}}(\mathbf{x}) &= \texttt{GroupNorm}(\mathbf{x}) + \mathbf{x},
\end{aligned}
\tag{8}
$$

where `GroupNorm` refers to Group Normalization (Wu & He, 2018), while `Conv1d` signifies 1D convolution (Kiranyaz et al., 2019). The `DownSampleBlock`, `UpSampleBlock`, and `BottleneckBlock` components, although distinct in their roles, share analogous layers with variations primarily in input and output channel sizes. Specifically, the `DownSampleBlock` commences with 64 channels, amplifying to 512, capturing intricate semantic information. In contrast, the `UpSampleBlock` initiates with 768 channels, tapering to 1, facilitating the restoration of feature map dimensions by harnessing the insights from the `DownSampleBlock`. This restoration is achieved through a skip connection, merging upsampled feature maps with their counterparts from the downsampling trajectory. The `BottleneckBlock` serves as a conduit, preserving consistent input and output channel dimensions, and distilling pivotal features from the downsampling phase. A comprehensive formulation is provided in the Appendix.

**Denoising Network Formulation.** Consequently, the denoising network is formulated as $f_\theta(\mathbf{x}, t) = \texttt{Network}(\mathbf{x}, \texttt{TimeTokenizer}(t))$. Here, `Network` can be any of the following architectures: `MLP`, `ResNet`, `Transformer`, or `U-Net`.

## 4 EXPERIMENTS

### 4.1 DATASET AND IMPLEMENTATIONS.

**Dataset.** We leverage seven publicly accessible datasets, offering a diverse representation of domains. These datasets are: (1) California Housing (CA), real estate data (R. Kelley Pace, 1997); (2) Helena (HE) and (3) Jannis (JA) are both anonymized datasets (Guyon et al., 2019); (4) Higgs (HI), simulated data of physical particles (P. Baldi, 2014), where we adopted the version housing 98K samples from the OpenML repository (Vanschoren et al., 2013); (5) ALOI (AL), an image-centric dataset (Geusebroek et al., 2005); (6) Year (YE), dataset capturing audio features (Bertin-Mahieux et al., 2011); (7) Covertype (CO), it describes forest characteristics (Blackard & Dean, 1999).

**Data Preprocessing.** To ensure equitable benchmarking, we administer a consistent preprocessing strategy for all datasets and models. Specifically, we scale each feature to a $(0, 1)$ range by subtracting its minimum and then dividing by its range. This transformation, conveniently integrated within the Scikit-learn library (Pedregosa et al., 2011), has been applied to both training and test data.

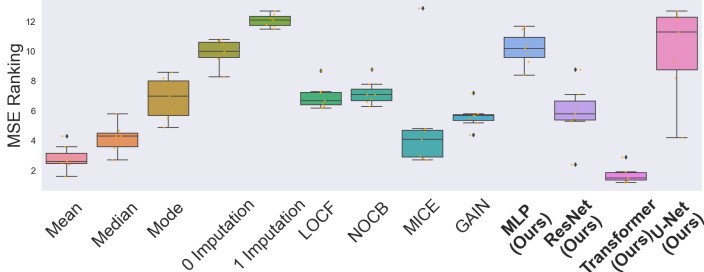

Figure 3: Imputation performance rankings of imputation methods in terms of MSE. The lower the better.

**Evaluation Metrics.** To gauge the precision of imputed values, we manually induce random masks on the test set data. The randomness of the mask is characterized by a percentage $p_{\text{random}} \in \{10\%, \dots, 90\%\}$ for each row (MCAR) and column mask (MAR) number $p_{\text{col}} \in \{1, \dots, 4\}$. Three evaluative criteria have been established: (1) Mean Squared Error (MSE); (2) Pearson Correlation Coefficient; (3) Downstream Tasks Performance. To mitigate potential biases from randomness during mask generation, we instantiate five distinct random seeds for each missing percentage. Given the inherent variability in data masking and diffusion inference, each random setting undergoes 25 inferences, arising from 5 unique data masks and 5 independent inferences per mask. For each mask generated using a unique random seed, the imputed data is multiplied by one-fifth for each inference, and the results are accumulated over five inferences. Subsequently, the sum of these accumulated results is employed to calculate the MSE for the particular generated mask. The final outcome for each mask setting is determined by averaging the five MSE results obtained from each generated mask from the corresponding random seed.

## 4.2 RESULTS.

**Comparison on Imputation Performance and Downstream Tasks.** We start our evaluation by contrasting the performance of `DiffImpute` with a range of established single and iterative tabular imputation methods. As illustrated in Fig. 3 and Tab. 1, when equipped with a Transformer as the denoising network, `DiffImpute` consistently surpasses its peers, both in terms of MSE that measures the imputation performance and downstream tasks on the imputed data. However, an anomaly is observed with the HI dataset. Its second-place performance can be traced back to the dataset's distinct characteristics, notably its dominant normal distributions and scant tail densities.

Table 1: Downstream task performance comparison using the imputed dataset. As different datasets apply different metrics, we report the performance rankings as the measurement.

| Imputation Methods | CA | HE | JA | HI | AL | YE | CO | Mean | Std |
|---|---|---|---|---|---|---|---|---|---|
| Mean Imputation | 3.9 | 4.5 | 6.5 | **1.8** | 6.9 | 3.9 | 4.3 | 4.5 | 1.7 |
| Median Imputation | 5.2 | 5.6 | 6.9 | 2.9 | 3.7 | 3.7 | 2.9 | 4.4 | 1.5 |
| Mode Imputation | 6.6 | 7.3 | 5.8 | 4.1 | 5.5 | 6.9 | 6.2 | 6.0 | 1.1 |
| 0 Imputation | 10.1 | 9.2 | 8.1 | 7.6 | 7.9 | 8.0 | 9.5 | 8.7 | 1.0 |
| 1 Imputation | 10.7 | 11.0 | 10.2 | 11.5 | 11.3 | 9.7 | 10.6 | 10.7 | 0.6 |
| LOCF Imputation | 8.2 | 10.5 | 10.1 | 9.7 | 11.5 | 10.5 | 8.5 | 9.9 | 1.2 |
| NOCB Imputation | 9.2 | 12.1 | 12.1 | 12.0 | 12.0 | 12.2 | 10.0 | 11.4 | 1.2 |
| MICE | 2.8 | 2.1 | 3.0 | 6.0 | 2.8 | 3.9 | 9.6 | 4.3 | 2.6 |
| GAIN | 4.9 | 3.5 | 4.0 | 7.3 | 4.9 | 5.2 | 7.7 | 5.4 | 1.6 |
| `DiffImpute w/ MLP` | 8.5 | 8.5 | 7.7 | 8.5 | 10.2 | 8.9 | 8.2 | 8.7 | 0.8 |
| `DiffImpute w/ ResNet` | 6.2 | 5.1 | 5.4 | 6.6 | 6.6 | 6.1 | 3.3 | 5.6 | 1.2 |
| `DiffImpute w/ Transformer` | **1.5** | **2.2** | **2.4** | 2.4 | **1.4** | **3.4** | **1.4** | 2.1 | 0.7 |
| `DiffImpute w/ U-Net` | 12.1 | 9.0 | 8.2 | 10.1 | 5.2 | 6.1 | 6.2 | 8.1 | 2.5 |

Table 2: Ablation on `Time Step Tokenizer` ('TST') and `Harmonization` ('H') with four denoising networks. We use the CA dataset and report the imputation performance in terms of MSE.

| TST | H | MLP | ResNet | Transformer | U-Net |
|---|---|---|---|---|---|
| ✗ | ✗ | 0.0212 | 0.0457 | 0.0210 | 0.0497 |
| ✓ | ✗ | 0.0585 | 0.0498 | 0.0194 | 0.6831 |
| ✗ | ✓ | 0.0164 | 0.0199 | 0.0174 | 0.0184 |
| ✓ | ✓ | 0.0268 | 0.0181 | 0.0191 | 4.2497 |

Table 3: Ablation on `Impute-DDIM` with four denoising networks. Note that when $\tau = 500$, no `Impute-DDIM` is applied.

| $\tau$ | MLP | ResNet | Transformer | U-Net |
|---|---|---|---|---|
| 10 | 0.2791 | 0.2574 | 0.2576 | 0.2741 |
| 25 | 0.2396 | 0.1892 | 0.1808 | 0.2274 |
| 50 | 0.1895 | 0.1164 | 0.0986 | 0.1727 |
| 100 | 0.1252 | 0.0525 | 0.0353 | 0.1145 |
| 250 | 0.0556 | 0.0240 | 0.0193 | 0.0795 |
| 500 | 0.0585 | 0.0498 | 0.0194 | 0.6831 |

This particular outcome accentuates the effectiveness of the mean imputation technique. Interestingly, mean imputation not only holds its own but even outperforms well-regarded methods such as MICE, GAIN, and `DiffImpute` with ResNet. While MICE does outshine mean imputation in specific datasets like HE, AL, and YE, its overall rank suffers due to variable performance on other datasets. Within the sphere of deep generative models, GAIN's performance parallels that of `DiffImpute` with ResNet, albeit at a slower inference speed.

**Effect of Denoising Network Architectures.** Among the four denoising networks, the Transformer consistently stands out, marking its dominance in the tabular data domain. ResNets, on the other hand, serve as a robust baseline, delivering both impressive performance and swift inference speeds, thereby outperforming other models. The MLP and U-Net architectures face challenges in grasping sequential data, such as time step inputs. However, U-Net exhibits exceptional performance on the AL dataset, aligning with its foundational design for image data processing. Yet, its extended training and inference times make it a less optimal choice for tabular imputation. In summary, the Transformer within `DiffImpute` emerges as a leading solution.

**Ablation Study.** To gain deeper insights into the contributions of individual components, we conducted an ablation study on the time embedding layers, `Harmonization`, and `Impute-DDIM` on the CA dataset. We initiated our investigation by excluding the `time step tokenizer` from the denoising network. Interestingly, the impact on MSE performance was not uniform across models. This omission led to a noticeable decline in performance for the Transformer achitecture, with a 7.96% drop in MSE performance and 6.28% drop in the downstream task efficacy respectively. The U-Net and MLP architectures experienced significant improvements, recording a 63.81% and 94.76% enhancement in MSE, respectively. Subsequently, we evaluated the impact of incorporating the `Harmonization` with $j = 5$. The results, as detailed in Tab. 2, highlight the performance boosts achieved by `Harmonization` across various architectures. To illustrate, when integrated into the `DiffImpute` with the MLP model, there was a remarkable 53.81% increase in MSE and a 22.84% improvement in downstream task performance for the CA dataset. Lastly, we assessed the efficacy of `Impute-DDIM` in enhancing the inference speed, experimenting with different $\tau$ sampling steps, specifically $\tau \in \{10, 25, 50, 100, 250\}$. We also set $j = 5$. As shown in Tab. 3, when $\tau$ increases, the quality of imputation improves. Remarkably, with `Impute-DDIM` and a $\tau$ setting of 250, we managed to double the inference speed without compromising the MSE performance for both our MLP and Transformer models.

## 5 CONCLUSION

In this work, we introduce `DiffImpute`, a novel denoising diffusion model for imputing missing tabular data. By seamlessly incorporating the `Time Step Tokenizer`, we have adapted four distinct denoising network architectures to enhance the capabilities of `DiffImpute`. Moreover, the amalgamation of the `Harmonization` technique and `Impute-DDIM` ensures that `DiffImpute` delivers superior performance without incurring extended sampling time. Our empirical evaluations, spanning seven diverse datasets, underscore the potential of `DiffImpute` as a foundational tool, poised to catalyze future innovations in the realm of tabular data imputation. One future direction is to further accelerate the sampling stage by distillation (Salimans & Ho, 2022). Additionally, we envision broadening the scope of `DiffImpute` to cater to missing multimodal scenarios, given that latent space features can be intuitively treated as tabular data.

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
