## A  DATASET DETAILS

**Dataset Descriptions and Statistics.**   We employed seven benchmark datasets in our experiments, the specifics of which are elaborated in Tab. 4. These datasets span two primary tasks, namely classification and regression. For evaluation, we adopt the mean square error (MSE) for regression tasks and the accuracy score for classification tasks. The data distribution for each dataset is structured such that 80% is allocated for training and the remaining 20% for testing.

Table 4: Statistics of the seven datasets used in our experiments. Regression tasks utilize mean square error (MSE) for evaluation, while classification tasks employ accuracy score.

| Name | Abbr. | # Train | # Test | # Num | Task Type | Batch Size |
|---|---|---|---|---|---|---|
| California Housing | CA | 16512 | 4128 | 8 | Regression | 256 |
| Helena | HE | 52156 | 13040 | 27 | Multiclass | 256 |
| Jannis | JA | 66986 | 16747 | 54 | Multiclass | 256 |
| Higgs Small | HI | 78439 | 19610 | 28 | Binclass | 256 |
| ALOI | AL | 86400 | 21600 | 128 | Multiclass | 256 |
| Year | YE | 463715 | 51630 | 90 | Regression | 256 |
| Covtype | CO | 464809 | 116203 | 54 | Multiclass | 256 |

**Download Link.**   All datasets, formatted as `Numpy.darray`, are accessible for download from `https://www.dropbox.com/s/o53umyg6mn3zhxy/data.tar.gz?dl=1`. The source of these datasets is `https://github.com/Yura52/tabular-dl-revisiting-models`.

**Preprocessing.**   For preprocessing, we standardized the numerical features and target values of each dataset using the `scikit-learn` library (Pedregosa et al., 2011). The standardization is based on the following equations:

$$\mathbf{X}_{\text{std}} = \frac{(\mathbf{X} - X_{\min})}{X_{\max} - X_{\min}},$$
$$\mathbf{X}_{\text{scaled}} = \mathbf{X}_{\text{std}} \cdot (\max - \min) + \min. \tag{9}$$

This preprocessing is applied to all variables, excluding the classification labels $\mathbf{y}$ for datasets CA, HE, JA, HI, AL, and CO. The feature values are scaled to lie between 0 and 1, with min=0 and max=1. Then we maintain a consistent 80% and 20% train-test split across all datasets, enabling uniform evaluation.

## B  METHODOLOGICAL DETAILS

This section elaborates on the details of the methodology.

**Detailed Formulation of of U-Net.**   The following equations given the formal definition of the `DownSampleBlock`, `UpSampleBlock`, and the `BottleneckBlock`:

$$
\begin{aligned}
&\texttt{DownSampleBlock}(\mathbf{x}, \mathbf{t}_{\text{emb}}) = \texttt{SiLU}(\texttt{GroupNorm}(\texttt{Conv1d}(\texttt{SiLU}(\\
&\quad \texttt{TimeStepMLP}(\texttt{GroupNorm}(\texttt{Conv1d}(\mathbf{x}))), \mathbf{t}_{\text{emb}})))) + \mathbf{x}\\
&\texttt{UpSampleBlock}(\texttt{Concat}(\mathbf{x}, \texttt{DownSampleBlock}(\mathbf{x}, \mathbf{t}_{\text{emb}})), \mathbf{t}_{\text{emb}}) = \texttt{SiLU}(\\
&\quad \texttt{GroupNorm}(\texttt{Conv1d}(\texttt{SiLU}(\texttt{TimeStepMLP}(\texttt{GroupNorm}(\texttt{Conv1d}(\\
&\quad \texttt{Concat}(\mathbf{x}, \texttt{DownSampleBlock}(\mathbf{x}, \mathbf{t}_{\text{emb}}))))), \mathbf{t}_{\text{emb}})))) + \texttt{Concat}(\mathbf{x},\\
&\quad \texttt{DownSampleBlock}(\mathbf{x}, \mathbf{t}_{\text{emb}}))\\
&\texttt{BottleneckBlock}(\mathbf{x}) = \texttt{MHSA}(\texttt{Res}_{\text{U-Net}}(\texttt{SiLU}(\texttt{GroupNorm}(\texttt{Conv1d}(\texttt{SiLU}(\\
&\quad \texttt{TimeStepMLP}(\texttt{GroupNorm}(\texttt{Conv1d}(\mathbf{x}))), \mathbf{t}_{\text{emb}})))) + \mathbf{x})).
\end{aligned} \tag{10}
$$

**Pseudo Code for the Training Stage.**   The pseudo code of `DiffImpute` training is summarized in Alg. 2.

---

**Algorithm 2** Pseudo code for the training stage of `DiffImpute` on a complete dataset $\mathbf{x}$.

---

1: **input:** Complete training data $\mathbf{x} \subseteq \mathbb{R}^k$, batch size $N$, time steps $T$, denoising network $f_\theta$, and smooth L1 loss scaling parameter $\beta_{\text{L1}} = 1$.
2: **for** $epoch = 1, 2, \dots$ **do**
3:     **for** sampled mini-batch $\{\mathbf{x}\}^N\} \in \mathcal{X}$ **do**
4:         $t \sim \text{Uniform}(\{1, \dots, T\})$ ▷ Uniformly sample time steps for denoising model training
5:         $\epsilon \sim \mathcal{N}(\mathbf{0}, \mathbf{I})$         ▷ Sample random noise from the Gaussian distribution
6:         Compute the $\mathbf{x}_t$ based on $\mathbf{x}_0$: $\sqrt{\bar{\alpha}_t}\mathbf{x}_0 + \sqrt{1 - \bar{\alpha}_t}\epsilon$     ▷ Diffuse $\mathbf{x}_0$ to the noisy data $\mathbf{x}_t$
    based on Eq. (1)
7:         Compute the predicted noise $\epsilon_\theta = f(\mathbf{x}_t^i, t)$
8:         Define the smooth L1 loss function $\mathcal{L} := \begin{cases} 0.5\,(\epsilon - \epsilon_\theta)^2/\beta_{\text{L1}}, & \text{if } |\epsilon - \epsilon_\theta| < \beta_{\text{L1}} \\ |\epsilon - \epsilon_\theta| - 0.5 \cdot \beta_{\text{L1}} & \text{otherwise} \end{cases}$ ▷
    Calculate the loss between predicted noise $\epsilon_\theta$ and ground truth noise $\epsilon$
9:         Update neural network $f_\theta(\mathbf{x}_t, t)$ to minimize $\mathcal{L}$ using AdamW optimizer.
10:     **end for**
11: **end for**
12: **return** denoising network $f_\theta(\mathbf{x}_t, t)$

---

**Algorithms for `Impute-DDIM` Step Schedule.** Pseudo code for the `Impute-DDIM` skip type schedule function definition is depicted in code snippet. 1.

Code Listing 1: `Impute-DDIM` skip type schedule function

```python
def skip_seq(num_timesteps, timesteps, skip_type="uniform"):
    if skip_type == "uniform":
        skip = num_timesteps // timesteps
        seq = range(0, num_timesteps, skip)
        return seq
    elif skip_type == "quad":
        seq = (
            np.linspace(
                0, np.sqrt(num_timesteps * 0.8), timesteps
            )
            ** 2
        )
        seq = [int(s) for s in list(seq)]
        return ddim_seq
    else:
        raise NotImplementedError
```

**Algorithms for `Harmonization with Impute-DDIM` Schedule.** Pseudo code for the `Harmonization` schedule function definition is illustrated in code snippet. 2, where working with the `Impute-DDIM`. The ddim_seq argument is the output of the function of skip_seq from the code snippet. 1.

Code Listing 2: `Harmonization` schedule function definition

```python
def get_schedule_jump_DDIM(ddim_seq, jump_length, jump_n_sample):
    jumps = {}
    for j in range(0, len(ddim_seq)-jump_length, jump_n_sample):
        jumps[ddim_seq[j]] = jump_n_sample - 1

    t = len(ddim_seq)
    ts = []

    while t >= 1:
        t = t-1
        ts.append(ddim_seq[t])

        if jumps.get(ddim_seq[t], 0) > 0:
            jumps[ddim_seq[t]] = jumps[ddim_seq[t]]-1
            for _ in range(jump_length):
                t = t + 1
                ts.append(ddim_seq[t])

    ts.append(-1)

    return ts
```

**Schematic Illustration.**    To elucidate the diffusing and denoising process, we present a visual representation in Fig. 4. This diagram captures the intricate dynamics of noise addition and subsequent denoising. Specifically, it illustrates how the data distribution gradually morphs into a Gaussian distribution during the noise addition phase and reverts during the denoising phase.

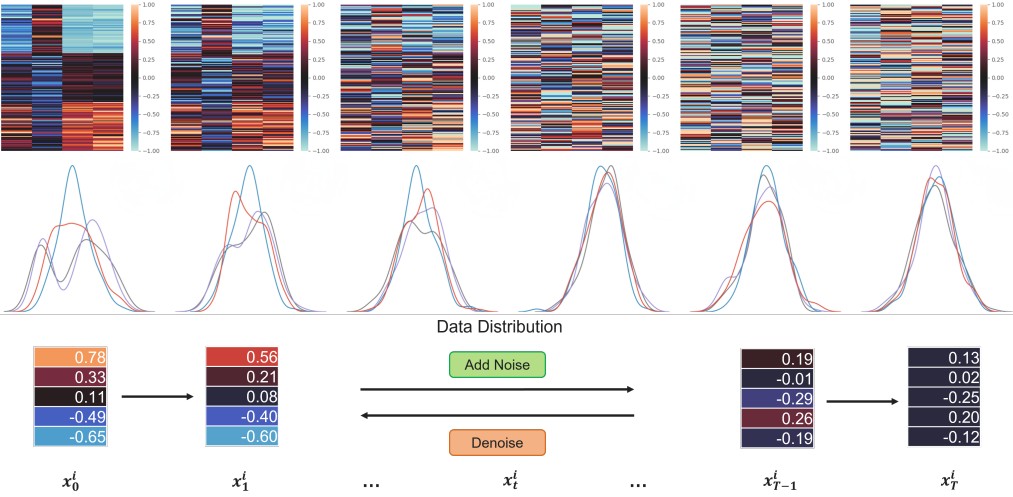

Figure 4: This visualization captures the dual processes of noise addition and denoising. As noise is added, the data distribution converges towards a Gaussian shape, which is then reversed during the denoising phase.

## C    IMPLEMENTATION DETAILS

**Hardware Platforms.**    Our implementation followed a structured workflow:

- We did the data preprocessing on any suitable hardware.
- Model training, inference, and evaluation were exclusively performed on an NVIDIA Tesla 3090 24GB GPU, boasting 35.6 TFLOPS. The software environment was consistent across all experiments, utilizing Python version 3.10.9 and Pytorch version 2.0.1+cu117.

**Training Settings.** While `DiffImpute` is trained on complete data, it performs imputation on test data, thereby leveraging insights from the complete dataset. To ensure a fair comparison, we also provided the training data as contextual information for all competing methods during their test data imputation.

**Hyper-parameters for `DiffImpute`.** `DiffImpute` is trained over 20 epochs using batch sizes of 64. Across all denoising network architectures and datasets, we employed an initial learning rate of $1e-3$, complemented by a learning rate decay of $1e-5$, optimized via AdamW. A notable deviation is observed in the U-Net architecture for the YE dataset, which operates without feature learning rate decay and adopts an initial rate of 0.01. During training, we designate the time step as $T_{\text{training}} = 1000$. Conversely, during the sampling phase, it's set to $T_{\text{sampling}=500}$, representing the reverse process steps. The diffusion coefficient, $\alpha_t$, is derived from the forward process variance $\beta_t$, defined as $\alpha_t := 1 - \beta_t$. We adopt the $\beta_t$ schedule from a cosine schedule (Nichol & Dhariwal, 2021). The posterior variance calculation follows: $\sigma_t = \frac{1 - \bar{\alpha}_{t-1}}{1 - \bar{\alpha}_t} \cdot \beta_t$ (Ho et al., 2020). For the `Impute-DDIM` acceleration, we partition the sampling step $T_{\text{sampling}}$ by a condensed time step $S$, uniformly distributing $T_{\text{sampling}}$ across $S$ steps. For clarity, we set $\eta = 0$, resulting in $\sigma_t = 0$, where $\sigma_t(\eta) = \eta\sqrt{(1 - \alpha_{t-1}) / (1 - \alpha_t)}\sqrt{1 - \alpha_t / \alpha_{t-1}}$ (Song et al., 2022). Tabs. 5 to 8 describe the implementation and configuration details of the four denoising networks.

Table 5: MLP model hyper-parameters as denoising network architecture in `DiffImpute`.

| Imputation Methods | CA | HE | JA | HI | AL | YE | CO |
|---|---|---|---|---|---|---|---|
| Layer count | 3 | 3 | 3 | 3 | 3 | 3 | 3 |
| Feature embedding size | / | / | / | / | / | / | / |
| Head count | 8 | 8 | 8 | 8 | 8 | 8 | 8 |
| Activation & FFN size factor | (ReLU, /) | (ReLU, /) | (ReLU, /) | (ReLU, /) | (ReLU, /) | (ReLU, /) | (ReLU, /) |
| Attention dropout | 0.2 | 0.2 | 0.2 | 0.2 | 0.2 | 0.2 | 0.2 |
| FFN dropout | 0.1 | 0.1 | 0.1 | 0.1 | 0.1 | 0.1 | 0.1 |
| Residual droupout | 0 | 0 | 0 | 0 | 0 | 0 | 0 |
| Initialization | / | / | / | / | / | / | / |
| Parameter count | 2376 | 18,279 | 65,718 | 19,516 | 65,718 | 174,330 | 65,718 |
| Optimizer | AdamW | AdamW | AdamW | AdamW | AdamW | AdamW | AdamW |
| Learning rate | 1e-3 | 1e-3 | 1e-3 | 1e-3 | 1e-3 | 1e-3 | 1e-3 |
| Weight decay | 1e-5 | 1e-5 | 1e-5 | 1e-5 | 1e-5 | 1e-5 | 1e-5 |

Table 6: ResNet model hyper-parameters as denoising network architecture in `DiffImpute`.

| Imputation Methods | CA | HE | JA | HI | AL | YE | CO |
|---|---|---|---|---|---|---|---|
| Layer count | 3 | 3 | 3 | 3 | 3 | 3 | 3 |
| Feature embedding size | 192 | 4.5 | 3.0 | 4.3 | 3.3 | 2.0 | 5.8 |
| Head count | 8 | 8 | 8 | 8 | 8 | 8 | 8 |
| Activation & FFN size factor | (ReLU, /) | (ReLU, /) | (ReLU, /) | (ReLU, /) | (ReLU, /) | (ReLU, /) | (ReLU, /) |
| Attention dropout | 0.2 | 0.2 | 0.2 | 0.2 | 0.2 | 0.2 | 0.2 |
| FFN dropout | 0.1 | 0.1 | 0.1 | 0.1 | 0.1 | 0.1 | 0.1 |
| Residual droupout | 0 | 0 | 0 | 0 | 0 | 0 | 0 |
| Initialization | / | / | / | / | / | / | / |
| Parameter count | 3784 | 22,119 | 73,014 | 23,484 | 73,014 | 186,234 | 73,014 |
| Optimizer | AdamW | AdamW | AdamW | AdamW | AdamW | AdamW | AdamW |
| Learning rate | 1e-3 | 1e-3 | 1e-3 | 1e-3 | 1e-3 | 1e-3 | 1e-3 |
| Weight decay | 1e-5 | 1e-5 | 1e-5 | 1e-5 | 1e-5 | 1e-5 | 1e-5 |

Table 7: U-Net model hyper-parameters as denoising network architecture in `DiffImpute`.

| Imputation Methods | CA | HE | JA | HI | AL | YE | CO |
|---|---|---|---|---|---|---|---|
| **Layer count** | 3 | 3 | 3 | 3 | 3 | 3 | 3 |
| **Feature embedding size** | / | / | / | / | / | / | / |
| **Head count** | 8 | 8 | 8 | 8 | 8 | 8 | 8 |
| **Activation & FFN size factor** | (SiLU, /) | (SiLU, /) | (SiLU) | (SiLU, /) | (SiLU, /) | (SiLU, /) | (SiLU, /) |
| **Attention dropout** | 0.2 | 0.2 | 0.2 | 0.2 | 0.2 | 0.2 | 0.2 |
| **FFN dropout** | 0.1 | 0.1 | 0.1 | 0.1 | 0.1 | 0.1 | 0.1 |
| **Residual droupout** | 0 | 0 | 0 | 0 | 0 | 0 | 0 |
| **Initialization** | / | / | / | / | / | / | / |
| **Parameter count** | 5,284,664 | 5,590,792 | 6,051,898 | 5,607,324 | 6,051,898 | 6,714,334 | 6,051,898 |
| **Optimizer** | AdamW | AdamW | AdamW | AdamW | AdamW | AdamW | AdamW |
| **Learning rate** | 1e-3 | 1e-3 | 1e-3 | 1e-3 | 1e-3 | 1e-3 | 1e-3 |
| **Weight decay** | 1e-2 | 1e-2 | 1e-2 | 1e-2 | 1e-2 | 1e-2 | 1e-2 |

Table 8: Transformer model hyper-parameters as denoising network architecture in `DiffImpute`.

| Imputation Methods | CA | HE | JA | HI | AL | YE | CO |
|---|---|---|---|---|---|---|---|
| **Layer count** | 3 | 3 | 3 | 3 | 3 | 3 | 3 |
| **Feature embedding size** | 192 | 192 | 192 | 192 | 192 | 192 | 192 |
| **Head count** | 8 | 8 | 8 | 8 | 8 | 8 | 8 |
| **Activation & FFN size factor** | (ReGLU, 4/3) | (ReGLU, 4/3) | (ReGLU, 4/3) | (ReGLU, 4/3) | (ReGLU, 4/3) | (ReGLU, 4/3) | (ReGLU, 4/3) |
| **Attention dropout** | 0.2 | 0.2 | 0.2 | 0.2 | 0.2 | 0.2 | 0.2 |
| **FFN dropout** | 0.1 | 0.1 | 0.1 | 0.1 | 0.1 | 0.1 | 0.1 |
| **Residual droupout** | 0 | 0 | 0 | 0 | 0 | 0 | 0 |
| **Initialization** | kaiming | kaiming | kaiming | kaiming | kaiming | kaiming | kaiming |
| **Parameter count** | 3,997,448 | 4,008,411 | 4,023,990 | 4,008,988 | 4,023,990 | 4,044,762 | 4,023,990 |
| **Optimizer** | AdamW | AdamW | AdamW | AdamW | AdamW | AdamW | AdamW |
| **Learning rate** | 1e-3 | 1e-3 | 1e-3 | 1e-3 | 1e-3 | 1e-3 | 1e-3 |
| **Weight decay** | 1e-5 | 1e-5 | 1e-5 | 1e-5 | 1e-5 | 1e-5 | 1e-5 |

**Evaluation Metrics.** To assess imputation performance, we employ the following metrics. We first denote the imputed data as $\hat{\mathbf{x}} \in \mathbb{R}^k$ and the ground truth as $\mathbf{x} \in \mathbb{R}^k$. Here, $\hat{x}i$ represents the $i$-th imputed value, and $x_i$ is the corresponding $i$-th ground truth value. We use $N_{\text{miss}}$ to signify the total number of missing values.

- **Mean Squared Error (MSE):** This metric quantifies the average squared discrepancy between the imputed and actual values.

$$\text{MSE}(\mathbf{x}, \hat{\mathbf{x}}) = \frac{\sum_{i=0}^{N_{\text{miss}}-1}(x_i - \hat{x}_i)^2}{N_{\text{miss}}} \tag{11}$$

- **Pearson Correlation Coefficient:** This evaluates the linear relationship between the actual and imputed values.

$$\text{R}(\mathbf{x}, \hat{\mathbf{x}}) = \frac{\sum_{i=0}^{N_{\text{miss}}-1}((x_i - mean(\mathbf{x})) \cdot (\hat{x}_i - mean(\hat{\mathbf{x}})))}{\sqrt{\sum_{i=0}^{N_{\text{miss}}-1}(x_i - mean(\mathbf{x}))^2} \cdot \sqrt{\sum_{i=0}^{N_{\text{miss}}-1}(\hat{x}_i - mean(\hat{\mathbf{x}}))^2}}$$

- **Downstream Tasks Performance:** For evaluating the performance on downstream tasks, we consistently use the same training and test sets. Depending on the nature of the downstream task, we employ either the root mean squared error (RMSE) for regression or the accuracy score for classification.

  - **RMSE:** For regression tasks, the RMSE metric is used, where $y_i$ and $\hat{y}_i$ denote the $i$-th actual and predicted values, respectively, and $N$ is the total number of values, defined as:

  $$\text{RMSE}(\mathbf{y}, \hat{\mathbf{y}}) = \sqrt{\frac{\sum_{i=0}^{N-1}(y_i - \hat{y}_i)^2}{N}} \tag{12}$$

  - **Accuracy Score:** For classification tasks, we utilize the accuracy score, as defined in the `Scikit-learn` library (Pedregosa et al., 2011). Here, $\mathbb{1}_{[\hat{y}_i=y_i]}$ is an indicator function that returns 1 if the condition $\hat{x}_i = y_i$ holds true.

  $$\text{Accuracy Score}(\mathbf{y}, \hat{\mathbf{y}}) = \frac{\sum_{i=0}^{N}\mathbb{1}_{[\hat{x}_i=y_i]}}{N} \tag{13}$$

**Compared Methods.** Our research endeavors to benchmark various imputation techniques and model architectures across a suite of seven datasets. It's crucial to note that we refrained from fine-tuning model parameters or employing model-agnostic deep learning enhancements like pretraining, additional loss functions, or data augmentation. Although these methods can potentially elevate model performance, our core objective remains to gauge the intrinsic efficacy of the diverse model architectures under uniform conditions. Below, we elaborate on a concise synopsis of the methods under comparison:

- **Mean Imputation**: Substitutes missing values with the feature's mean.

- **Median Imputation**: Uses the median of available values for imputation.

- **Mode Imputation**: Fills missing slots with the most frequent value.

- **0 Imputation**: Directly replaces missing values with 0.

- **1 Imputation**: Uses 1 as the replacement.

- **LOCF Imputation**: Fills gaps with the last observed value.

- **NOCB Imputation**: Uses the subsequent observed value for imputation.

- **MICE (linear) Imputation**: Employs multiple imputations based on regularized linear regression (van Buuren & Groothuis-Oudshoorn, 2011).

- **GAIN Imputation**: Leverages Generative Adversarial Nets for imputation (Yoon et al., 2018).

**Hyper-parameters for Compared Methods.** Below, we detail the hyper-parameters of the compared methods used in our experiments:

- **MICE:** We fix and do not tune the following hyper-parameters:
  - $n_{imputations} = 1$
  - $max_{iter} = 100$
  - $initial_{strategy} = 0$
  - $imputation_{order} = 0$
  - $random_{state}$ is set to the current time.
- **GAIN:** We fix and do not tune the following hyper-parameters:
  - $batch_{size} = 256$
  - $n_{epochs} = 1000$
  - $hint_{rate} = 0.9$
  - $loss_{alpha} = 10$

# D MORE RESULTS

## D.1 IMPUTATION PERFORMANCE IN TERMS OF MSE.

We present the mean squared error (MSE) results for imputed data, evaluated under various missingness mechanisms across our seven benchmark datasets.

**Random Mask.** In this segment, we focus on the imputation performance under the random mask settings. This mechanism aligns with the Missing Completely At Random (MCAR). The results for each of the seven datasets are detailed in the subsequent tables, referenced as Tabs. 9 to 15.

Table 9: Imputation performance comparison in terms of random mask setting, *i.e.* Missing Completely At Random (MCAR), on CA using MSE. Optimal results are highlighted in **bold**.

| Imputation Methods | 10% | 20% | 30% | 40% | 50% | 60% | 70% | 80% | 90% |
|---|---|---|---|---|---|---|---|---|---|
| Mean Imputation | 0.0210 | 0.0212 | 0.0214 | 0.0212 | 0.0213 | **0.0212** | **0.0212** | **0.0213** | **0.0212** |
| Median Imputation | 0.0254 | 0.0256 | 0.0257 | 0.0254 | 0.0257 | 0.0256 | 0.0256 | 0.0256 | 0.0256 |
| Mode Imputation | 0.0689 | 0.0843 | 0.0683 | 0.0689 | 0.0683 | 0.0681 | 0.0533 | 0.0536 | 0.0534 |
| 0 Imputation | 0.1055 | 0.1054 | 0.1067 | 0.1070 | 0.1073 | 0.1072 | 0.1070 | 0.1069 | 0.1069 |
| 1 Imputation | 0.6892 | 0.6896 | 0.6881 | 0.6874 | 0.6868 | 0.6871 | 0.6872 | 0.6875 | 0.6876 |
| LOCF Imputation | 0.0422 | 0.0421 | 0.0418 | 0.0426 | 0.0421 | 0.0422 | 0.0425 | 0.0425 | 0.0426 |
| NOCB Imputation | 0.0420 | 0.0436 | 0.0438 | 0.0425 | 0.0437 | 0.0432 | 0.0429 | 0.0430 | 0.0431 |
| MICE (linear) | 0.0192 | 0.0230 | 0.0252 | 0.0270 | 0.0314 | 0.0333 | 0.0367 | 0.0376 | 0.0400 |
| GAIN | 0.0224 | 0.0232 | 0.0238 | 0.0290 | 0.0422 | 0.0532 | 0.0739 | 0.0907 | 0.1024 |
| DiffImpute w/ MLP | 0.0495 | 0.0526 | 0.0554 | 0.0582 | 0.0609 | 0.0639 | 0.0670 | 0.0701 | 0.0734 |
| DiffImpute w/ ResNet | 0.0160 | 0.0171 | **0.0182** | 0.0196 | 0.0218 | 0.0254 | 0.0321 | 0.0449 | 0.0680 |
| DiffImpute w/ Transformer | **0.0155** | **0.0170** | 0.0184 | **0.0195** | **0.0210** | 0.0221 | 0.0233 | 0.0246 | 0.0259 |
| DiffImpute w/ U-Net | 0.6323 | 0.6540 | 0.6759 | 0.6895 | 0.7005 | 0.7077 | 0.7155 | 0.7206 | 0.7252 |

Table 10: Imputation performance comparison in terms of random mask setting, *i.e.* Missing Completely At Random (MCAR), on HE using MSE. Optimal results are highlighted in **bold**.

| Imputation Methods | 10% | 20% | 30% | 40% | 50% | 60% | 70% | 80% | 90% |
|---|---|---|---|---|---|---|---|---|---|
| Mean Imputation | 0.0285 | 0.0285 | 0.0285 | 0.0284 | 0.0285 | 0.0285 | 0.0285 | 0.0285 | **0.0285** |
| Median Imputation | 0.0294 | 0.0294 | 0.0293 | 0.0293 | 0.0293 | 0.0293 | 0.0293 | 0.0293 | 0.0293 |
| Mode Imputation | 0.0965 | 0.0960 | 0.0961 | 0.0960 | 0.0960 | 0.0960 | 0.0959 | 0.0958 | 0.0965 |
| 0 Imputation | 0.2547 | 0.2547 | 0.2544 | 0.2543 | 0.2542 | 0.2542 | 0.2542 | 0.2543 | 0.2543 |
| 1 Imputation | 0.3942 | 0.3943 | 0.3949 | 0.3950 | 0.3951 | 0.3951 | 0.3951 | 0.3950 | 0.3950 |
| LOCF Imputation | 0.0570 | 0.0573 | 0.0573 | 0.0572 | 0.0571 | 0.0571 | 0.0570 | 0.0570 | 0.0570 |
| NOCB Imputation | 0.0573 | 0.0574 | 0.0573 | 0.0572 | 0.0572 | 0.0572 | 0.0572 | 0.0572 | 0.0572 |
| MICE (linear) | 0.0125 | 0.0137 | 0.0156 | 0.0180 | 0.0205 | 0.0246 | 0.0296 | 0.0365 | 0.0453 |
| GAIN | 0.0227 | 0.0220 | 0.0241 | 0.0298 | 0.0544 | 0.1342 | 0.1550 | 0.1401 | 0.2536 |
| **DiffImpute w/ MLP** | 0.1116 | 0.1292 | 0.1482 | 0.1684 | 0.1902 | 0.2130 | 0.2371 | 0.2619 | 0.2876 |
| **DiffImpute w/ ResNet** | 0.0122 | 0.0136 | 0.0154 | 0.0178 | 0.0218 | 0.0291 | 0.0442 | 0.0757 | 0.1381 |
| **DiffImpute w/ Transformer** | **0.0088** | **0.0101** | **0.0117** | **0.0137** | **0.0162** | **0.0193** | **0.0227** | **0.0268** | 0.0314 |
| **DiffImpute w/ U-Net** | 0.2464 | 0.2579 | 0.2705 | 0.2894 | 0.3026 | 0.3233 | 0.3475 | 0.3759 | 0.4098 |

Table 11: Imputation performance comparison in terms of random mask setting, *i.e.* Missing Completely At Random (MCAR), on JA using MSE. Optimal results are highlighted in **bold**.

| Imputation Methods | 10% | 20% | 30% | 40% | 50% | 60% | 70% | 80% | 90% |
|---|---|---|---|---|---|---|---|---|---|
| Mean Imputation | 0.0295 | 0.0295 | 0.0295 | 0.0295 | 0.0295 | 0.0295 | **0.0295** | **0.0295** | **0.0295** |
| Median Imputation | 0.0303 | 0.0303 | 0.0303 | 0.0303 | 0.0303 | 0.0303 | 0.0303 | 0.0303 | 0.0303 |
| Mode Imputation | 0.1003 | 0.0998 | 0.0998 | 0.1009 | 0.1011 | 0.1012 | 0.1013 | 0.1033 | 0.1005 |
| 0 Imputation | 0.2262 | 0.2263 | 0.2261 | 0.2262 | 0.2262 | 0.2263 | 0.2262 | 0.2262 | 0.2262 |
| 1 Imputation | 0.4131 | 0.4128 | 0.4132 | 0.4130 | 0.4129 | 0.4128 | 0.4129 | 0.4128 | 0.4128 |
| LOCF Imputation | 0.0590 | 0.0589 | 0.0588 | 0.0589 | 0.0588 | 0.0588 | 0.0588 | 0.0588 | 0.0589 |
| NOCB Imputation | 0.0589 | 0.0588 | 0.0589 | 0.0588 | 0.0589 | 0.0588 | 0.0588 | 0.0587 | 0.0586 |
| MICE (linear) | 0.0366 | 0.0376 | 0.0384 | 0.0396 | 0.0410 | 0.0428 | 0.0456 | 0.0487 | 0.0533 |
| GAIN | 0.0407 | 0.0375 | 0.0436 | 0.0538 | 0.0733 | 0.1355 | 0.0904 | 0.0804 | 0.2039 |
| DiffImpute w/ MLP | 0.2158 | 0.2521 | 0.2880 | 0.3230 | 0.3569 | 0.3902 | 0.4229 | 0.4547 | 0.4857 |
| **DiffImpute w/ ResNet** | 0.0242 | 0.0253 | 0.0270 | 0.0301 | 0.0358 | 0.0470 | 0.0679 | 0.1035 | 0.1599 |
| **DiffImpute w/ Transformer** | **0.0233** | **0.0240** | **0.0249** | **0.0260** | **0.0273** | **0.0288** | 0.0305 | 0.0325 | 0.0347 |
| **DiffImpute w/ U-Net** | 0.3720 | 0.4570 | 0.5631 | 0.6937 | 0.8462 | 1.016 | 1.1949 | 1.3656 | 1.4972 |

Table 12: Imputation performance comparison in terms of random mask setting, *i.e.* Missing Completely At Random (MCAR), on HI using MSE. Optimal results are highlighted in **bold**.

| Imputation Methods | 10% | 20% | 30% | 40% | 50% | 60% | 70% | 80% | 90% |
|---|---|---|---|---|---|---|---|---|---|
| Mean Imputation | 0.0570 | 0.0572 | **0.0570** | **0.0570** | **0.0570** | **0.0570** | **0.0569** | **0.0569** | **0.0568** |
| Median Imputation | 0.0681 | 0.0698 | 0.0711 | 0.0739 | 0.0724 | 0.0738 | 0.0737 | 0.0737 | 0.0739 |
| Mode Imputation | 0.1028 | 0.1013 | 0.1015 | 0.1014 | 0.1004 | 0.0995 | 0.0977 | 0.1019 | 0.0984 |
| 0 Imputation | 0.1844 | 0.1849 | 0.1847 | 0.1845 | 0.1845 | 0.1844 | 0.1845 | 0.1845 | 0.1845 |
| 1 Imputation | 0.5811 | 0.5807 | 0.5806 | 0.5808 | 0.5808 | 0.5808 | 0.5807 | 0.5807 | 0.5806 |
| LOCF Imputation | 0.1135 | 0.1144 | 0.1139 | 0.1140 | 0.1140 | 0.1138 | 0.1135 | 0.1135 | 0.1135 |
| NOCB Imputation | 0.1135 | 0.1140 | 0.1137 | 0.1138 | 0.1141 | 0.1137 | 0.1137 | 0.1137 | 0.1137 |
| MICE (linear) | 0.0838 | 0.0875 | 0.0913 | 0.0956 | 0.0990 | 0.1022 | 0.1059 | 0.1088 | 0.1114 |
| GAIN | 0.0867 | 0.0811 | 0.0806 | 0.0955 | 0.1026 | 0.1330 | 0.1381 | 0.1483 | 0.1778 |
| **DiffImpute w/ MLP** | 0.1523 | 0.1652 | 0.1781 | 0.1921 | 0.2071 | 0.2226 | 0.2384 | 0.2544 | 0.2708 |
| **DiffImpute w/ ResNet** | **0.0545** | **0.0568** | 0.0592 | 0.0626 | 0.0680 | 0.0767 | 0.0911 | 0.1142 | 0.1501 |
| **DiffImpute w/ Transformer** | 0.0594 | 0.0613 | 0.0625 | 0.0638 | 0.0650 | 0.0661 | 0.0670 | 0.0680 | 0.0688 |
| **DiffImpute w/ U-Net** | 0.7151 | 0.7265 | 0.7362 | 0.7465 | 0.7575 | 0.7676 | 0.7777 | 0.7877 | 0.7975 |

Table 13: Imputation performance comparison in terms of random mask setting, *i.e.* Missing Completely At Random (MCAR), on AL using MSE. Optimal results are highlighted in **bold**.

| Imputation Methods | 10% | 20% | 30% | 40% | 50% | 60% | 70% | 80% | 90% |
|---|---|---|---|---|---|---|---|---|---|
| Mean Imputation | 0.0175 | 0.0176 | 0.0175 | 0.0175 | 0.0175 | 0.0175 | 0.0175 | 0.0175 | **0.0175** |
| Median Imputation | 0.0209 | 0.0209 | 0.0209 | 0.0209 | 0.0209 | 0.0209 | 0.0209 | 0.0209 | 0.0209 |
| Mode Imputation | 0.0255 | 0.0255 | 0.0255 | 0.0255 | 0.0255 | 0.0255 | 0.0255 | 0.0255 | 0.0255 |
| 0 Imputation | 0.0386 | 0.0386 | 0.0387 | 0.0386 | 0.0386 | 0.0386 | 0.0386 | 0.0386 | 0.0386 |
| 1 Imputation | 0.8833 | 0.8832 | 0.8831 | 0.8831 | 0.8833 | 0.8831 | 0.8832 | 0.8832 | 0.8832 |
| LOCF Imputation | 0.0351 | 0.0351 | 0.0351 | 0.0351 | 0.0351 | 0.0351 | 0.0351 | 0.0351 | 0.0351 |
| NOCB Imputation | 0.0351 | 0.0351 | 0.0351 | 0.0351 | 0.0351 | 0.0351 | 0.0351 | 0.0351 | 0.0351 |
| MICE (linear) | 0.0065 | 0.0071 | 0.0079 | 0.0087 | 0.0099 | 0.0114 | 0.0136 | 0.0169 | 0.0224 |
| GAIN | 0.0067 | 0.0079 | 0.0126 | 0.0154 | 0.0183 | 0.0203 | 0.0257 | 0.0302 | 0.0343 |
| **DiffImpute w/ MLP** | 0.2710 | 0.3174 | 0.3541 | 0.3857 | 0.4129 | 0.4370 | 0.4584 | 0.4776 | 0.4949 |
| **DiffImpute w/ ResNet** | 0.0098 | 0.0105 | 0.0115 | 0.0133 | 0.0168 | 0.0229 | 0.0327 | 0.0469 | 0.0652 |
| **DiffImpute w/ Transformer** | **0.0048** | **0.0054** | **0.0062** | **0.0071** | **0.0083** | **0.0100** | **0.0120** | **0.0146** | 0.0177 |
| **DiffImpute w/ U-Net** | 0.0130 | 0.0139 | 0.0148 | 0.0158 | 0.0169 | 0.0182 | 0.0197 | 0.0217 | 0.0242 |

Table 14: Imputation performance comparison in terms of random mask setting, *i.e.* Missing Completely At Random (MCAR), on YE using MSE. Optimal results are highlighted in **bold**.

| Imputation Methods | 10% | 20% | 30% | 40% | 50% | 60% | 70% | 80% | 90% |
|---|---|---|---|---|---|---|---|---|---|
| Mean Imputation | 0.0009 | 0.0009 | 0.0009 | 0.0009 | 0.0009 | 0.0009 | 0.0009 | **0.0009** | **0.0009** |
| Median Imputation | 0.0009 | 0.0009 | 0.0009 | 0.0009 | 0.0009 | 0.0009 | 0.0009 | 0.0009 | 0.0009 |
| Mode Imputation | 0.0010 | 0.0010 | 0.0010 | 0.0010 | 0.0010 | 0.0010 | 0.0010 | 0.0010 | 0.0010 |
| 0 Imputation | 0.2251 | 0.2252 | 0.2252 | 0.2251 | 0.2252 | 0.2251 | 0.2251 | 0.2251 | 0.2252 |
| 1 Imputation | 0.3553 | 0.3552 | 0.3552 | 0.3552 | 0.3552 | 0.3552 | 0.3552 | 0.3552 | 0.3552 |
| LOCF Imputation | 0.0018 | 0.0018 | 0.0018 | 0.0018 | 0.0018 | 0.0018 | 0.0018 | 0.0018 | 0.0018 |
| NOCB Imputation | 0.0018 | 0.0018 | 0.0018 | 0.0018 | 0.0018 | 0.0018 | 0.0018 | 0.0018 | 0.0018 |
| MICE (linear) | 0.0001 | 0.0002 | 0.0003 | 0.0004 | 0.0005 | 0.0007 | 0.0012 | 0.0014 | 0.0016 |
| GAIN | 0.0641 | 0.0015 | 0.0019 | 0.0032 | 0.0128 | 0.0843 | 0.0148 | 0.1877 | 0.2252 |
| **DiffImpute w/ MLP** | 0.2011 | 0.2672 | 0.3260 | 0.3795 | 0.4282 | 0.4729 | 0.5143 | 0.5526 | 0.5885 |
| **DiffImpute w/ ResNet** | 0.0013 | 0.0014 | 0.0016 | 0.0023 | 0.0048 | 0.0132 | 0.0346 | 0.0759 | 0.1440 |
| **DiffImpute w/ Transformer** | **0.0006** | **0.0006** | **0.0006** | **0.0007** | **0.0007** | **0.0008** | **0.0008** | **0.0009** | 0.0010 |
| **DiffImpute w/ U-Net** | 0.0036 | 0.0045 | 0.0057 | 0.0750 | 0.0106 | 0.0171 | 0.0313 | 0.0606 | 0.1161 |

Table 15: Imputation performance comparison in terms of random mask setting, *i.e.* Missing Completely At Random (MCAR), on CO using MSE. Optimal results are highlighted in **bold**.

| Imputation Methods | 10% | 20% | 30% | 40% | 50% | 60% | 70% | 80% | 90% |
|---|---|---|---|---|---|---|---|---|---|
| Mean Imputation | 0.0333 | 0.0333 | 0.0333 | 0.0333 | 0.0333 | 0.0333 | **0.0333** | **0.0333** | **0.0333** |
| Median Imputation | 0.0425 | 0.0424 | 0.0424 | 0.0424 | 0.0424 | 0.0424 | 0.0425 | 0.0425 | 0.0425 |
| Mode Imputation | 0.0472 | 0.0471 | 0.0471 | 0.0471 | 0.0471 | 0.0471 | 0.0471 | 0.0471 | 0.0471 |
| 0 Imputation | 0.0909 | 0.0908 | 0.0908 | 0.0908 | 0.0908 | 0.0908 | 0.0908 | 0.0908 | 0.0908 |
| 1 Imputation | 0.8479 | 0.8481 | 0.8480 | 0.8480 | 0.8480 | 0.8480 | 0.8480 | 0.8480 | 0.8480 |
| LOCF Imputation | 0.0666 | 0.0665 | 0.0666 | 0.0665 | 0.0666 | 0.0666 | 0.0666 | 0.0666 | 0.0666 |
| NOCB Imputation | 0.0665 | 0.0664 | 0.0665 | 0.0664 | 0.0665 | 0.0665 | 0.0665 | 0.0666 | 0.0667 |
| MICE (linear) | 29550 | 33301 | 880.73 | 7965.1 | 154.84 | 5.7013 | 0.46 | 8976.6 | 4148.3 |
| GAIN | 0.0290 | 0.0292 | 0.0314 | 0.0405 | 0.0663 | 0.0768 | 0.0751 | 0.784 | 0.0893 |
| **DiffImpute w/ MLP** | 0.1555 | 0.1827 | 0.2100 | 0.2373 | 0.2642 | 0.2910 | 0.3180 | 0.3443 | 0.3701 |
| **DiffImpute w/ ResNet** | 0.0200 | 0.0220 | 0.0243 | 0.0268 | 0.0300 | 0.0342 | 0.0368 | 0.0388 | 0.0407 |
| **DiffImpute w/ Transformer** | **0.0176** | **0.0206** | **0.0235** | **0.0263** | **0.0290** | **0.0315** | 0.0345 | 0.0368 | 0.0390 |
| **DiffImpute w/ U-Net** | 0.1098 | 0.1249 | 0.1447 | 0.1703 | 0.2047 | 0.2504 | 0.3122 | 0.3949 | 0.5069 |

**Column Mask.** In this segment, we assess the imputation performance under the column mask settings, aligning with the Missing At Random (MAR). The results for each of the seven datasets are detailed in the subsequent tables, referenced as Tabs. 16 to 22. It's important to highlight that the NOCB imputation method is not suitable for the column mask setting, given the absence of a subsequent observation to utilize for imputation.

Table 16: Imputation performance comparison in terms of column mask setting, *i.e.* Missing At Random (MAR), on CA using MSE. The best results are in **bold**.

| Imputation Methods | 1 | 2 | 3 | 4 |
|---|---|---|---|---|
| Mean Imputation | 0.0228 | 0.0245 | 0.0220 | 0.0137 |
| Median Imputation | 0.0273 | 0.0314 | 0.0266 | 0.0162 |
| Mode Imputation | 0.0702 | 0.0544 | 0.0565 | 0.0281 |
| 0 Imputation | 0.1043 | 0.1214 | 0.0944 | 0.0818 |
| 1 Imputation | 0.7275 | 0.6591 | 0.7035 | 0.7407 |
| LOCF Imputation | 0.0419 | 0.0453 | 0.0462 | 0.0246 |
| NOCB Imputation | / | / | / | / |
| MICE (linear) | 0.1012 | **0.0009** | **0.0111** | **0.0030** |
| GAIN | 0.0610 | 0.0011 | 0.0062 | 0.0067 |
| **DiffImpute w/ MLP** | 0.0492 | 0.0586 | 0.0550 | 0.0469 |
| **DiffImpute w/ ResNet** | 0.0849 | 0.0225 | 0.0846 | 0.0902 |
| **DiffImpute w/ Transformer** | **0.0184** | 0.0208 | 0.0173 | 0.0088 |
| **DiffImpute w/ U-Net** | 0.6117 | 0.6188 | 0.7210 | 0.7079 |

Table 17: Imputation performance comparison in terms of column mask setting, *i.e.* Missing At Random (MAR), on HE using MSE. The best results are in **bold**.

| Imputation Methods | 1 | 2 | 3 | 4 |
|---|---|---|---|---|
| Mean Imputation | 0.0225 | 0.0200 | 0.0351 | 0.0421 |
| Median Imputation | 0.0231 | 0.0202 | 0.0360 | 0.0437 |
| Mode Imputation | 0.1043 | 0.0239 | 0.1337 | 0.1733 |
| 0 Imputation | 0.2856 | 0.3412 | 0.2279 | 0.2301 |
| 1 Imputation | 0.3066 | 0.3333 | 0.4127 | 0.4674 |
| LOCF Imputation | 0.0266 | 0.0316 | 0.0469 | 0.0504 |
| NOCB Imputation | / | / | / | / |
| MICE (linear) | 0.0015 | **0.0014** | 0.0207 | 0.0321 |
| GAIN | **0.0009** | 0.0024 | **0.0143** | 0.0286 |
| **DiffImpute w/ MLP** | 0.0983 | 0.1067 | 0.1234 | 0.1322 |
| **DiffImpute w/ ResNet** | 0.2633 | 0.3497 | 0.2640 | 0.2210 |
| **DiffImpute w/ Transformer** | 0.0021 | 0.0149 | 0.0151 | **0.0149** |
| **DiffImpute w/ U-Net** | 0.1920 | 0.3147 | 0.2874 | 0.2284 |

Table 18: Imputation performance comparison in terms of column mask setting, *i.e.* Missing At Random (MAR), on JA using MSE. The best results are in **bold**.

| Imputation Methods | 1 | 2 | 3 | 4 |
|---|---|---|---|---|
| Mean Imputation | 0.0347 | 0.0279 | 0.0294 | 0.0379 |
| Median Imputation | 0.0358 | 0.0281 | 0.0303 | 0.0389 |
| Mode Imputation | 0.0550 | 0.0776 | 0.0880 | 0.0719 |
| 0 Imputation | 0.1891 | 0.1987 | 0.2332 | 0.3026 |
| 1 Imputation | 0.4190 | 0.4063 | 0.3930 | 0.3380 |
| LOCF Imputation | 0.0582 | 0.0338 | 0.0631 | 0.0846 |
| NOCB Imputation | / | / | / | / |
| MICE (linear) | 0.0568 | 0.0561 | 0.0205 | 0.0272 |
| GAIN | 0.0303 | 0.0348 | 0.0164 | **0.0190** |
| **DiffImpute w/ MLP** | 0.2041 | 0.2014 | 0.1993 | 0.2253 |
| **DiffImpute w/ ResNet** | 0.2059 | 0.3091 | 0.2522 | 0.2880 |
| **DiffImpute w/ Transformer** | **0.0299** | **0.0253** | **0.0114** | 0.0197 |
| **DiffImpute w/ U-Net** | 0.3295 | 0.3412 | 0.3179 | 0.4265 |

Table 19: Imputation performance comparison in terms of column mask setting, *i.e.* Missing At Random (MAR), on HI using MSE. The best results are in **bold**.

| Imputation Methods | 1 | 2 | 3 | 4 |
|---|---|---|---|---|
| Mean Imputation | 0.0263 | 0.0492 | 0.0635 | 0.0534 |
| Median Imputation | 0.0264 | 0.0701 | 0.0842 | 0.0536 |
| Mode Imputation | 0.0707 | 0.0768 | 0.1187 | 0.1066 |
| 0 Imputation | 0.1307 | 0.1545 | 0.1905 | 0.1830 |
| 1 Imputation | 0.6354 | 0.6198 | 0.5846 | 0.5696 |
| LOCF Imputation | 0.0664 | 0.1227 | 0.1460 | 0.1021 |
| NOCB Imputation | / | / | / | / |
| MICE (linear) | **0.0018** | 0.0043 | 0.0543 | 0.1110 |
| GAIN | **0.0018** | **0.0030** | **0.0314** | 0.0723 |
| `DiffImpute` w/ MLP | 0.1090 | 0.1334 | 0.1473 | 0.1437 |
| `DiffImpute` w/ ResNet | 0.0788 | 0.1824 | 0.1983 | 0.1881 |
| `DiffImpute` w/ Transformer | 0.0301 | 0.0536 | 0.0676 | **0.0562** |
| `DiffImpute` w/ U-Net | 0.6449 | 0.6786 | 0.7392 | 0.7265 |

Table 20: Imputation performance comparison in terms of column mask setting, *i.e.* Missing At Random (MAR), on AL using MSE. The best results are in **bold**.

| Imputation Methods | 1 | 2 | 3 | 4 |
|---|---|---|---|---|
| Mean Imputation | 0.0086 | 0.0214 | 0.0138 | 0.0185 |
| Median Imputation | 0.0102 | 0.0265 | 0.0171 | 0.0227 |
| Mode Imputation | 0.0102 | 0.0287 | 0.0171 | 0.0233 |
| 0 Imputation | 0.0102 | 0.0331 | 0.0171 | 0.0368 |
| 1 Imputation | 0.9433 | 0.8718 | 0.9328 | 0.8881 |
| LOCF Imputation | 0.0102 | 0.1004 | 0.0509 | 0.0752 |
| NOCB Imputation | / | / | / | / |
| MICE (linear) | 0.0106 | 0.0208 | 0.0068 | 0.0101 |
| GAIN | 0.0099 | 0.0201 | 0.0058 | 0.0086 |
| `DiffImpute` w/ MLP | 0.1989 | 0.2244 | 0.2204 | 0.2348 |
| `DiffImpute` w/ ResNet | 0.0507 | 0.0476 | 0.0225 | 0.0791 |
| `DiffImpute` w/ Transformer | **0.0029** | **0.0069** | **0.0037** | **0.0064** |
| `DiffImpute` w/ U-Net | 0.0068 | 0.0057 | 0.0166 | 0.0142 |

Table 21: Imputation performance comparison in terms of column mask setting, *i.e.* Missing At Random (MAR), on YE using MSE. The best results are in **bold**.

| Imputation Methods | 1 | 2 | 3 | 4 |
|---|---|---|---|---|
| Mean Imputation | 0.0007 | 0.0011 | 0.0010 | 0.0013 |
| Median Imputation | 0.0007 | 0.0011 | 0.0010 | 0.0013 |
| Mode Imputation | 0.0007 | 0.0014 | 0.0012 | 0.0015 |
| 0 Imputation | 0.3638 | 0.2321 | 0.2263 | 0.2119 |
| 1 Imputation | 0.2126 | 0.4028 | 0.3276 | 0.3496 |
| LOCF Imputation | 0.0009 | 0.0016 | 0.0011 | 0.0017 |
| NOCB Imputation | / | / | / | / |
| MICE (linear) | 0.0008 | 0.0012 | 0.0004 | 0.0007 |
| GAIN | 0.0006 | 0.0019 | 0.0003 | 0.0011 |
| **DiffImpute w/ MLP** | 0.1465 | 0.1535 | 0.1629 | 0.1756 |
| **DiffImpute w/ ResNet** | 0.3666 | 0.3285 | 0.2516 | 0.2469 |
| **DiffImpute w/ Transformer** | **0.0004** | **0.0007** | **0.0007** | **0.0009** |
| **DiffImpute w/ U-Net** | 0.0013 | 0.0011 | 0.0015 | 0.0014 |

Table 22: Imputation performance comparison in terms of column mask setting, *i.e.* Missing At Random (MAR), on CO using MSE. The best results are in **bold**.

| Imputation Methods | 1 | 2 | 3 | 4 |
|---|---|---|---|---|
| Mean Imputation | 0.0378 | 0.0333 | 0.0323 | 0.0303 |
| Median Imputation | 0.0409 | 0.0353 | 0.0341 | 0.0321 |
| Mode Imputation | 0.0595 | 0.0394 | 0.0509 | 0.0566 |
| 0 Imputation | 0.0622 | 0.1206 | 0.0633 | 0.0759 |
| 1 Imputation | 0.8684 | 0.7813 | 0.8494 | 0.7801 |
| LOCF Imputation | 0.0444 | 0.2175 | 0.1031 | 0.0499 |
| NOCB Imputation | / | / | / | / |
| MICE (linear) | NaN | NaN | NaN | NaN |
| GAIN | NaN | NaN | NaN | NaN |
| **DiffImpute w/ MLP** | 0.1474 | 0.1451 | 0.1396 | 0.1430 |
| **DiffImpute w/ ResNet** | 0.0366 | 0.0322 | 0.0325 | 0.0292 |
| **DiffImpute w/ Transformer** | **0.0245** | **0.0213** | **0.0253** | **0.0230** |
| **DiffImpute w/ U-Net** | 0.1034 | 0.1022 | 0.0926 | 0.1111 |

**Imputation Performance Rankings.** In this segment, we showcase the consolidated rankings of imputation performance, measured by mean squared error (MSE), under various masking mechanisms, specifically Missing Completely At Random (MCAR) and Missing At Random (MAR). These rankings span seven datasets, as detailed in Tabs. 23 to 25. Within each dataset, the performance metrics are sorted to determine the rankings. The column labeled "rank" represents the average ranking across the different missingness settings.

Table 23: Overall imputation performance rankings under the random mask setting (MCAR) evaluated by MSE. `DiffImpute` with Transformer has the best overall performance. The `DiffImpute` with the Transformer architecture outperform other methods in six datasets out of seven datasets. The best results are in **bold**.

| Imputation Methods | CA | HE | JA | HI | AL | YE | CO | Mean | Std |
|---|---|---|---|---|---|---|---|---|---|
| Mean Imputation | 2.1 | 3.4 | 2.0 | **1.2** | 4.3 | 2.4 | 2.6 | 2.6 | 0.9 |
| Median Imputation | 4.2 | 4.6 | 3.1 | 3.6 | 5.9 | 2.4 | 4.4 | 4.0 | 1.0 |
| Mode Imputation | 9.2 | 8.4 | 8.6 | 5.8 | 7.3 | 4.6 | 5.4 | 7.0 | 1.7 |
| 0 Imputation | 11.0 | 10.8 | 10.1 | 10.3 | 10.8 | 11.0 | 9.0 | 10.4 | 0.7 |
| 1 Imputation | 12.3 | 12.9 | 11.8 | 12.0 | 13.0 | 12.3 | 12.1 | 12.3 | 0.4 |
| LOCF Imputation | 6.4 | 6.3 | 6.7 | 7.8 | 8.8 | 6.6 | 7.3 | 7.1 | 0.8 |
| NOCB Imputation | 7.1 | 7.1 | 6.3 | 7.8 | 8.8 | 6.6 | 6.8 | 7.2 | 0.8 |
| MICE | 4.4 | 3.1 | 4.7 | 5.6 | 2.2 | 2.3 | 12.9 | 5.0 | 3.4 |
| GAIN | 7.2 | 6.8 | 7.1 | 7.1 | 5.1 | 8.8 | 5.7 | 6.8 | 1.1 |
| **DiffImpute w/ MLP** | 9.0 | 10.2 | 11.2 | 10.7 | 12.0 | 12.6 | 10.8 | 10.9 | 1.1 |
| **DiffImpute w/ ResNet** | 3.6 | 3.9 | 4.7 | 3.7 | 6.0 | 7.8 | 2.4 | 4.6 | 1.7 |
| **DiffImpute w/ Transformer** | **1.7** | **1.2** | **1.7** | 2.4 | **1.1** | **1.9** | **1.3** | 1.6 | 0.4 |
| **DiffImpute w/ U-Net** | 12.7 | 12.0 | 12.9 | 13.0 | 4.7 | 9.2 | 10.2 | 10.7 | 2.8 |

Table 24: Overall imputation performance rankings under the column mask setting (MAR) evaluated by MSE. `DiffImpute` with Transformer has the best overall performance. The `DiffImpute` with the Transformer architecture outperform other methods in six datasets out of seven datasets. The best results are in **bold**.

| Imputation Methods | CA | HE | JA | HI | AL | YE | CO | Mean | Std |
|---|---|---|---|---|---|---|---|---|---|
| Mean Imputation | 3.8 | 4.0 | 3.3 | **2.5** | 4.3 | 3.3 | 2.8 | 3.4 | 0.6 |
| Median Imputation | 4.8 | 5.0 | 4.3 | 4.0 | 5.8 | 3.3 | 4.0 | 4.4 | 0.7 |
| Mode Imputation | 7.5 | 7.5 | 6.3 | 6.3 | 6.3 | 5.8 | 5.5 | 6.4 | 0.7 |
| 0 Imputation | 9.8 | 10.5 | 8.8 | 9.3 | 6.8 | 10.3 | 6.8 | 8.8 | 1.4 |
| 1 Imputation | 11.8 | 11.5 | 11.8 | 11.0 | 12.0 | 11.5 | 10.0 | 11.4 | 0.6 |
| LOCF Imputation | 5.8 | 6.3 | 6.0 | 6.3 | 8.5 | 7.0 | 6.8 | 6.6 | 0.9 |
| NOCB Imputation | / | / | / | / | / | / | / | / | / |
| MICE | 3.3 | 2.3 | 4.5 | 3.0 | 4.8 | 3.5 | NaN | 3.5 | 0.9 |
| GAIN | 2.8 | 1.5 | 2.5 | 1.8 | 2.8 | 3.5 | NaN | 2.5 | 0.7 |
| **DiffImpute w/ MLP** | 7.3 | 7.3 | 8.5 | 8.3 | 11.0 | 9.0 | 8.8 | 8.5 | 1.2 |
| **DiffImpute w/ ResNet** | 7.8 | 10.3 | 9.8 | 9.5 | 9.5 | 11.3 | 2.3 | 8.8 | 2.8 |
| **DiffImpute w/ Transformer** | **2.5** | **2.3** | **1.3** | 4.0 | **1.3** | **1.8** | **1.0** | 2.0 | 1.0 |
| **DiffImpute w/ U-Net** | 11.3 | 9.8 | 11.3 | 12.0 | 3.0 | 6.0 | 7.3 | 8.6 | 3.1 |

Table 25: Overall imputation performance rankings under the random mask (MCAR) and the column mask (MAR) settings evaluated by MSE. `DiffImpute` with Transformer has the best overall performance. The `DiffImpute` with the Transformer architecture outperform other methods in six datasets out of seven datasets. The best results are in **bold**.

| Imputation Methods | CA | HE | JA | HI | AL | YE | CO | Mean | Std |
|---|---|---|---|---|---|---|---|---|---|
| Mean Imputation | 2.6 | 3.6 | 2.4 | **1.6** | 4.3 | 2.7 | 2.6 | 2.8 | 0.9 |
| Median Imputation | 4.4 | 4.7 | 3.5 | 3.7 | 5.8 | 2.7 | 4.3 | 4.2 | 1.0 |
| Mode Imputation | 8.7 | 8.2 | 7.8 | 5.9 | 7.0 | 4.9 | 5.5 | 6.9 | 1.4 |
| 0 Imputation | 10.6 | 10.7 | 9.7 | 10.0 | 9.5 | 10.8 | 8.3 | 9.9 | 0.9 |
| 1 Imputation | 12.2 | 12.5 | 11.8 | 11.7 | 12.7 | 12.1 | 11.5 | 12.0 | 0.4 |
| LOCF Imputation | 6.2 | 6.3 | 6.5 | 7.3 | 8.7 | 6.7 | 7.2 | 7.0 | 0.9 |
| NOCB Imputation | 7.1 | 7.1 | 6.3 | 7.8 | 8.8 | 6.6 | 6.8 | 7.2 | 0.8 |
| MICE | 4.1 | 2.8 | 4.6 | 4.8 | 3.0 | 2.7 | 12.9 | 5.0 | 3.6 |
| GAIN | 5.8 | 5.2 | 5.7 | 5.5 | 4.4 | 7.2 | 5.7 | 5.6 | 0.8 |
| **DiffImpute w/ MLP** | 8.5 | 9.3 | 10.4 | 9.9 | 11.7 | 11.5 | 10.2 | 10.2 | 1.1 |
| **DiffImpute w/ ResNet** | 4.8 | 5.8 | 6.2 | 5.5 | 7.1 | 8.8 | 2.4 | 5.8 | 2.0 |
| **DiffImpute w/ Transformer** | **1.9** | **1.5** | **1.5** | 2.9 | **1.2** | **1.8** | **1.2** | 1.7 | 0.6 |
| **DiffImpute w/ U-Net** | 12.2 | 11.3 | 12.4 | 12.7 | 4.2 | 8.2 | 9.3 | 10.0 | 3.1 |

**Visualization of the Imputation Performance.** Fig. 5 demonstrates the imputation process through the time steps of four denoising networks for the CA dataset with 90% random mask. The ResNet and Transformer architectures utilized in `DiffImpute` exhibit superior imputation capability.

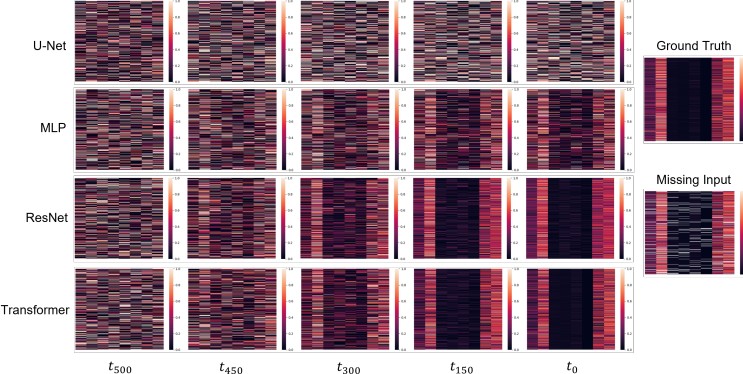

Figure 5: Sampling Process on CA dataset at 90% random mask of different model architectures.

## D.2 IMPUTATION PERFORMANCE IN TERMS OF PEARSON CORRELATION.

The following tables display the Pearson correlation performance between the ground truth data and the imputed data under different missingness mechanisms across seven datasets.

**Random Mask.** This section presents the evaluation of imputation performance using Pearson correlation under random mask settings, which correspond to the Missing Completely At Random (MCAR) mechanism, across seven datasets (Tabs. 26 to 32).

Table 26: Imputation performance comparison in terms of random mask setting, *i.e.* Missing Completely At Random (MCAR), on CA using Pearson correlation. The best results are in **bold**.

| Imputation Methods | 10% | 20% | 30% | 40% | 50% | 60% | 70% | 80% | 90% |
|---|---|---|---|---|---|---|---|---|---|
| Mean Imputation | 0.8138 | 0.8118 | 0.8126 | 0.8145 | 0.8142 | **0.8144** | **0.8139** | **0.8136** | **0.8140** |
| Median Imputation | 0.7780 | 0.7763 | 0.7777 | 0.7788 | 0.7782 | 0.7789 | 0.7784 | 0.7783 | 0.7787 |
| Mode Imputation | 0.7162 | 0.6895 | 0.7154 | 0.6926 | 0.7154 | 0.7156 | 0.7383 | 0.7376 | 0.7385 |
| LOCF Imputation | 0.6597 | 0.6649 | 0.6683 | 0.6620 | 0.6673 | 0.6654 | 0.6651 | 0.6640 | 0.6608 |
| NOCB Imputation | 0.6649 | 0.6528 | 0.6538 | 0.6613 | 0.6555 | 0.6584 | 0.6608 | 0.6599 | 0.6571 |
| MICE | 0.8418 | 0.8222 | 0.8012 | 0.7848 | 0.7527 | 0.7369 | 0.7125 | 0.7009 | 0.6832 |
| GAIN | 0.8211 | 0.8160 | 0.7947 | 0.7480 | 0.6207 | 0.5500 | 0.4414 | 0.3555 | 0.4523 |
| DiffImpute w/ MLP | 0.5824 | 0.5649 | 0.5529 | 0.5408 | 0.5290 | 0.5151 | 0.5015 | 0.4881 | 0.4758 |
| DiffImpute w/ ResNet | 0.8628 | 0.8533 | **0.8461** | **0.8380** | **0.8262** | 0.8104 | 0.7804 | 0.7105 | 0.5272 |
| DiffImpute w/ Transformer | **0.8680** | **0.8543** | 0.8429 | 0.8325 | 0.8187 | 0.8072 | 0.7950 | 0.7814 | 0.7687 |
| DiffImpute w/ U-Net | -0.0219 | -0.0392 | -0.0573 | -0.0678 | -0.0730 | -0.0775 | -0.0803 | -0.0839 | -0.0827 |

Table 27: Imputation performance comparison in terms of random mask setting, *i.e.* Missing Completely At Random (MCAR), on HE using Pearson correlation. The best results are in **bold**.

| Imputation Methods | 10% | 20% | 30% | 40% | 50% | 60% | 70% | 80% | 90% |
|---|---|---|---|---|---|---|---|---|---|
| Mean Imputation | 0.7682 | 0.7685 | 0.7690 | 0.7692 | 0.7691 | 0.7692 | 0.7692 | 0.7691 | **0.7692** |
| Median Imputation | 0.7646 | 0.7653 | 0.7656 | 0.7656 | 0.7655 | 0.7656 | 0.7656 | 0.7655 | 0.7656 |
| Mode Imputation | 0.3891 | 0.3929 | 0.3966 | 0.3962 | 0.3953 | 0.4021 | 0.4066 | 0.4101 | 0.4056 |
| LOCF Imputation | 0.5907 | 0.5884 | 0.5888 | 0.5893 | 0.5899 | 0.5905 | 0.5910 | 0.5910 | 0.5910 |
| NOCB Imputation | 0.5891 | 0.5885 | 0.5893 | 0.5896 | 0.5896 | 0.5897 | 0.5899 | 0.5899 | 0.5899 |
| MICE (linear) | 0.9100 | 0.9019 | 0.8882 | 0.8711 | 0.8528 | 0.8236 | 0.7575 | 0.7381 | 0.6740 |
| GAIN | 0.8414 | 0.8390 | 0.8259 | 0.7960 | 0.7095 | 0.3360 | 0.2034 | 0.3752 | 0.2500 |
| DiffImpute w/ MLP | 0.3977 | 0.3533 | 0.3138 | 0.2783 | 0.2464 | 0.2183 | 0.1938 | 0.1728 | 0.1542 |
| DiffImpute w/ ResNet | 0.9092 | 0.8987 | 0.8858 | 0.8689 | 0.8450 | 0.8107 | 0.7560 | 0.6594 | 0.4552 |
| DiffImpute w/ Transformer | **0.9354** | **0.9252** | **0.9130** | **0.8971** | **0.8769** | **0.8521** | **0.8229** | **0.7882** | 0.7425 |
| DiffImpute w/ U-Net | 0.2709 | 0.2677 | 0.2671 | 0.2648 | 0.2613 | 0.2562 | 0.2520 | 0.2474 | 0.2437 |

Table 28: Imputation performance comparison in terms of random mask setting, *i.e.* Missing Completely At Random (MCAR), on JA using Pearson correlation. The best results are in **bold**.

| Imputation Methods | 10% | 20% | 30% | 40% | 50% | 60% | 70% | 80% | 90% |
|---|---|---|---|---|---|---|---|---|---|
| Mean Imputation | 0.7182 | 0.7179 | 0.7182 | 0.7180 | 0.7182 | 0.7181 | **0.7180** | **0.7179** | **0.7179** |
| Median Imputation | 0.7140 | 0.7137 | 0.7140 | 0.7137 | 0.7141 | 0.7140 | 0.7139 | 0.7138 | 0.7139 |
| Mode Imputation | 0.3168 | 0.3176 | 0.3155 | 0.3127 | 0.3094 | 0.3062 | 0.3042 | 0.3034 | 0.3013 |
| LOCF Imputation | 0.5163 | 0.5162 | 0.5166 | 0.5164 | 0.5165 | 0.5163 | 0.5162 | 0.5160 | 0.5162 |
| NOCB Imputation | 0.5162 | 0.5164 | 0.5164 | 0.5164 | 0.5160 | 0.5165 | 0.5167 | 0.5167 | 0.5174 |
| MICE (linear) | 0.6996 | 0.6916 | 0.6855 | 0.6759 | 0.6638 | 0.6489 | 0.6262 | 0.6004 | 0.5631 |
| GAIN | 0.6658 | 0.6803 | 0.6514 | 0.6283 | 0.5944 | 0.3190 | 0.4965 | 0.4867 | 0.0952 |
| DiffImpute w/ MLP | 0.1892 | 0.1691 | 0.1509 | 0.1359 | 0.1236 | 0.1130 | 0.1032 | 0.0943 | 0.0864 |
| DiffImpute w/ ResNet | 0.7773 | 0.7672 | 0.7530 | 0.7310 | 0.6974 | 0.6442 | 0.5630 | 0.4463 | 0.2906 |
| DiffImpute w/ Transformer | **0.7904** | **0.7827** | **0.7739** | **0.7630** | **0.7503** | **0.7351** | 0.7176 | 0.6970 | 0.6743 |
| DiffImpute w/ U-Net | 0.1525 | 0.1473 | 0.1472 | 0.1500 | 0.1536 | 0.1561 | 0.1571 | 0.1568 | 0.1563 |

Table 29: Imputation performance comparison in terms of random mask setting, *i.e.* Missing Completely At Random (MCAR), on HI using Pearson correlation. The best results are in **bold**.

| Imputation Methods | 10% | 20% | 30% | 40% | 50% | 60% | 70% | 80% | 90% |
|---|---|---|---|---|---|---|---|---|---|
| Mean Imputation | 0.6248 | 0.6240 | **0.6243** | **0.6243** | **0.6240** | **0.6243** | **0.6248** | **0.6250** | **0.6251** |
| Median Imputation | 0.5513 | 0.5419 | 0.5329 | 0.5157 | 0.5246 | 0.5161 | 0.5171 | 0.5172 | 0.5176 |
| Mode Imputation | 0.3910 | 0.3947 | 0.3880 | 0.3860 | 0.3922 | 0.3977 | 0.4012 | 0.3907 | 0.4091 |
| LOCF Imputation | 0.3912 | 0.3878 | 0.3896 | 0.3899 | 0.3899 | 0.3911 | 0.3922 | 0.3922 | 0.3922 |
| NOCB Imputation | 0.3919 | 0.3911 | 0.3920 | 0.3913 | 0.3900 | 0.3913 | 0.3907 | 0.3907 | 0.3907 |
| MICE (linear) | 0.5469 | 0.5316 | 0.5093 | 0.4861 | 0.4688 | 0.4521 | 0.4329 | 0.4180 | 0.4032 |
| GAIN | 0.4461 | 0.4754 | 0.4829 | 0.4305 | 0.4532 | 0.3956 | 0.3797 | 0.4449 | 0.2699 |
| `DiffImpute w/ MLP` | 0.2938 | 0.2781 | 0.2667 | 0.2552 | 0.2437 | 0.2341 | 0.2261 | 0.2189 | 0.2129 |
| `DiffImpute w/ ResNet` | **0.6475** | **0.6317** | 0.6138 | 0.5914 | 0.5593 | 0.5124 | 0.4430 | 0.3457 | 0.2271 |
| `DiffImpute w/ Transformer` | 0.6133 | 0.5994 | 0.5885 | 0.5774 | 0.5671 | 0.5574 | 0.5496 | 0.5416 | 0.5339 |
| `DiffImpute w/ U-Net` | 0.0052 | 0.0041 | 0.0036 | 0.0031 | 0.0014 | -0.0001 | -0.0012 | -0.0024 | -0.0036 |

Table 30: Imputation performance comparison in terms of random mask setting, *i.e.* Missing Completely At Random (MCAR), on AL using Pearson correlation. The best results are in **bold**.

| Imputation Methods | 10% | 20% | 30% | 40% | 50% | 60% | 70% | 80% | 90% |
|---|---|---|---|---|---|---|---|---|---|
| Mean Imputation | 0.6797 | 0.6793 | 0.6802 | 0.6800 | 0.6798 | 0.6798 | 0.6798 | 0.6797 | 0.6796 |
| Median Imputation | 0.6310 | 0.6304 | 0.6313 | 0.6311 | 0.6309 | 0.6310 | 0.6310 | 0.6309 | 0.6308 |
| Mode Imputation | 0.5520 | 0.5508 | 0.5519 | 0.5515 | 0.5514 | 0.5551 | 0.5515 | 0.5514 | 0.5513 |
| LOCF Imputation | 0.4617 | 0.4617 | 0.4617 | 0.4617 | 0.4617 | 0.4617 | 0.4617 | 0.4617 | 0.4617 |
| NOCB Imputation | 0.4612 | 0.4612 | 0.4612 | 0.4612 | 0.4612 | 0.4612 | 0.4612 | 0.4612 | 0.4612 |
| MICE (linear) | 0.9006 | 0.8912 | 0.8789 | **0.8660** | 0.8486 | 0.8248 | 0.7907 | 0.7395 | 0.6545 |
| GAIN | 0.8993 | 0.8804 | 0.7993 | 0.7464 | 0.6900 | 0.6576 | 0.5322 | 0.4854 | 0.4521 |
| `DiffImpute w/ MLP` | 0.0752 | 0.0546 | 0.0406 | 0.0304 | 0.0227 | 0.0161 | 0.0112 | 0.0069 | 0.0034 |
| `DiffImpute w/ ResNet` | 0.8360 | 0.8239 | 0.8049 | 0.7705 | 0.7035 | 0.5849 | 0.4190 | 0.2437 | 0.0939 |
| `DiffImpute w/ Transformer` | **0.9233** | **0.9133** | **0.9009** | 0.8845 | **0.8627** | **0.8335** | **0.7952** | **0.7448** | **0.6819** |
| `DiffImpute w/ U-Net` | 0.7762 | 0.7597 | 0.7428 | 0.7241 | 0.7035 | 0.6788 | 0.6509 | 0.6156 | 0.5722 |

Table 31: Imputation performance comparison in terms of random mask setting, *i.e.* Missing Completely At Random (MCAR), on YE using Pearson correlation. The best results are in **bold**.

| Imputation Methods | 10% | 20% | 30% | 40% | 50% | 60% | 70% | 80% | 90% |
|---|---|---|---|---|---|---|---|---|---|
| Mean Imputation | 0.9877 | 0.9876 | 0.9876 | 0.9876 | 0.9876 | 0.9876 | 0.9876 | **0.9876** | **0.9876** |
| Median Imputation | 0.9876 | 0.9875 | 0.9875 | 0.9875 | 0.9875 | 0.9875 | 0.9875 | 0.9875 | 0.9875 |
| Mode Imputation | 0.9864 | 0.9863 | 0.9863 | 0.9864 | 0.9863 | 0.9864 | 0.9864 | 0.9864 | 0.9865 |
| LOCF Imputation | 0.9755 | 0.9754 | 0.9754 | 0.9754 | 0.9754 | 0.9754 | 0.9755 | 0.9755 | 0.9755 |
| NOCB Imputation | 0.9754 | 0.9754 | 0.9754 | 0.9754 | 0.9754 | 0.9754 | 0.9754 | 0.9754 | 0.9754 |
| MICE (linear) | **0.9989** | **0.9977** | **0.9963** | **0.9947** | **0.9928** | **0.9906** | 0.9829 | 0.9810 | 0.9782 |
| GAIN | 0.9830 | 0.9815 | 0.9777 | 0.9719 | 0.9384 | 0.7138 | 0.9052 | 0.2598 | 0.2119 |
| DiffImpute w/ MLP | 0.2620 | 0.2079 | 0.1741 | 0.1506 | 0.1340 | 0.1218 | 0.1124 | 0.1052 | 0.0994 |
| DiffImpute w/ ResNet | 0.9818 | 0.9809 | 0.9789 | 0.9740 | 0.9602 | 0.9206 | 0.8206 | 0.6173 | 0.2984 |
| DiffImpute w/ Transformer | 0.9921 | 0.9917 | 0.9912 | 0.9906 | 0.9900 | 0.9892 | **0.9883** | 0.9874 | 0.9862 |
| DiffImpute w/ U-Net | 0.9499 | 0.9375 | 0.9229 | 0.9045 | 0.8818 | 0.8512 | 0.8102 | 0.7602 | 0.7064 |

Table 32: Imputation performance comparison in terms of random mask setting, *i.e.* Missing Completely At Random (MCAR), on CO using Pearson correlation. The best results are in **bold**.

| Imputation Methods | 10% | 20% | 30% | 40% | 50% | 60% | 70% | 80% | 90% |
|---|---|---|---|---|---|---|---|---|---|
| Mean Imputation | 0.7499 | 0.7499 | 0.7500 | 0.7499 | 0.7500 | 0.7501 | **0.7501** | **0.7501** | **0.7501** |
| Median Imputation | 0.6827 | 0.6828 | 0.6828 | 0.6828 | 0.6827 | 0.6829 | 0.6828 | 0.6827 | 0.6828 |
| Mode Imputation | 0.6520 | 0.6520 | 0.6520 | 0.6520 | 0.6520 | 0.6521 | 0.6521 | 0.6520 | 0.6521 |
| LOCF Imputation | 0.5622 | 0.5628 | 0.5625 | 0.5626 | 0.5623 | 0.5622 | 0.5624 | 0.5627 | 0.5623 |
| NOCB Imputation | 0.5628 | 0.5631 | 0.5631 | 0.5632 | 0.5630 | 0.5630 | 0.5627 | 0.5625 | 0.5618 |
| MICE (linear) | -0.0150 | -0.0070 | 0.0036 | -0.0510 | -0.1070 | -0.0390 | 0.1820 | 0.0021 | -0.0020 |
| GAIN | 0.7928 | 0.7928 | 0.7874 | 0.7475 | 0.5077 | 0.3772 | 0.4619 | 0.4580 | 0.2975 |
| DiffImpute w/ MLP | 0.2707 | 0.2231 | 0.1846 | 0.1526 | 0.1263 | 0.1044 | 0.0852 | 0.0693 | 0.0556 |
| DiffImpute w/ ResNet | 0.8604 | 0.8441 | 0.8267 | 0.8064 | 0.7815 | 0.7475 | 0.7218 | 0.7054 | 0.6888 |
| DiffImpute w/ Transformer | **0.8780** | **0.8543** | **0.8317** | **0.8094** | **0.7880** | **0.7671** | 0.7425 | 0.7223 | 0.7034 |
| DiffImpute w/ U-Net | 0.3785 | 0.3344 | 0.2863 | 0.2348 | 0.1806 | 0.1257 | 0.0703 | 0.0167 | -0.0340 |

**Column Mask.** In this segment, we delve into the imputation performance assessment using the Pearson correlation metric under the column mask settings. This approach aligns with the Missing At Random (MAR) paradigm. The detailed results for each of the seven datasets are provided in Tabs. 33 to 39. It's pertinent to mention that the column mask setting renders the NOCB imputation method inapplicable, given the lack of a subsequent observation for imputation purposes.

Table 33: Imputation performance comparison in terms of column mask setting, *i.e.* Missing Completely At Random (MCAR), on CA using Pearson correlation. The best results are in **bold**.

| Imputation Methods | 1 | 2 | 3 | 4 |
|---|---|---|---|---|
| Mean Imputation | **0.8140** | **0.8140** | **0.8140** | **0.8140** |
| Median Imputation | 0.7787 | 0.7787 | 0.7787 | 0.7787 |
| Mode Imputation | 0.7385 | 0.7385 | 0.7385 | 0.7385 |
| LOCF Imputation | 0.6615 | 0.6615 | 0.6615 | 0.6615 |
| NOCB Imputation | / | / | / | / |
| MICE (linear) | 0.1814 | 0.2818 | 0.6596 | 0.9691 |
| GAIN | 0.0323 | 0.2640 | 0.7887 | 0.9685 |
| DiffImpute w/ MLP | 0.0317 | 0.3627 | 0.3746 | 0.5798 |
| DiffImpute w/ ResNet | 0.1733 | -0.0002 | 0.3057 | -0.0469 |
| DiffImpute w/ Transformer | 0.2575 | 0.6394 | 0.7743 | 0.9175 |
| DiffImpute w/ U-Net | -0.0022 | -0.0640 | -0.0143 | -0.0897 |

Table 34: Imputation performance comparison in terms of column mask setting, *i.e.* Missing Completely At Random (MCAR), on HE using Pearson correlation. The best results are in **bold**.

| Imputation Methods | 1 | 2 | 3 | 4 |
|---|---|---|---|---|
| Mean Imputation | 0.7692 | 0.7692 | 0.7692 | 0.7692 |
| Median Imputation | 0.7656 | 0.7656 | 0.7656 | 0.7656 |
| Mode Imputation | 0.4056 | 0.4056 | 0.4056 | 0.4056 |
| LOCF Imputation | 0.5911 | 0.5911 | 0.5911 | 0.5911 |
| NOCB Imputation | / | / | / | / |
| MICE (linear) | 0.0797 | **0.9836** | 0.7218 | 0.7660 |
| GAIN | 0.0509 | 0.9713 | 0.7839 | 0.7937 |
| `DiffImpute w/ MLP` | 0.0457 | 0.2731 | 0.1824 | 0.3239 |
| `DiffImpute w/ ResNet` | 0.5779 | 0.2973 | 0.5354 | 0.5509 |
| `DiffImpute w/ Transformer` | **0.7734** | 0.8365 | **0.8169** | **0.8914** |
| `DiffImpute w/ U-Net` | 0.0572 | 0.2208 | 0.0971 | 0.1945 |

Table 35: Imputation performance comparison in terms of column mask setting, *i.e.* Missing Completely At Random (MCAR), on JA using Pearson correlation. The best results are in **bold**.

| Imputation Methods | 1 | 2 | 3 | 4 |
|---|---|---|---|---|
| Mean Imputation | **0.7179** | **0.7179** | 0.7179 | 0.7179 |
| Median Imputation | 0.7138 | 0.7139 | 0.7139 | 0.7139 |
| Mode Imputation | 0.3013 | 0.3013 | 0.3013 | 0.3013 |
| LOCF Imputation | 0.5162 | 0.5162 | 0.5162 | 0.5162 |
| NOCB Imputation | / | / | / | / |
| MICE (linear) | -0.0090 | 0.2864 | 0.8519 | 0.7471 |
| GAIN | 0.0141 | 0.3346 | **0.8844** | 0.8060 |
| `DiffImpute w/ MLP` | 0.0138 | 0.0548 | 0.0941 | 0.1827 |
| `DiffImpute w/ ResNet` | 0.1849 | 0.2422 | 0.3213 | 0.4518 |
| `DiffImpute w/ Transformer` | 0.1979 | 0.3747 | 0.8622 | **0.8505** |
| `DiffImpute w/ U-Net` | -0.0180 | 0.0699 | 0.0916 | 0.2285 |

Table 36: Imputation performance comparison in terms of column mask setting, *i.e.* Missing Completely At Random (MCAR), on HI using Pearson correlation. The best results are in **bold**.

| Imputation Methods | 1 | 2 | 3 | 4 |
|---|---|---|---|---|
| Mean Imputation | **0.6251** | **0.6251** | 0.6251 | 0.6251 |
| Median Imputation | 0.5176 | 0.5176 | 0.5176 | 0.5176 |
| Mode Imputation | 0.4091 | 0.4091 | 0.4091 | 0.4091 |
| LOCF Imputation | 0.3911 | 0.3911 | 0.3911 | 0.3911 |
| NOCB Imputation | / | / | / | / |
| MICE (linear) | 0.5234 | 0.3278 | 0.5956 | 0.5393 |
| GAIN | 0.3119 | 0.2392 | **0.7328** | **0.6325** |
| `DiffImpute w/ MLP` | 0.0024 | 0.1180 | 0.2406 | 0.2306 |
| `DiffImpute w/ ResNet` | 0.2010 | -0.0460 | 0.3727 | 0.1481 |
| `DiffImpute w/ Transformer` | 0.4956 | 0.3981 | 0.5255 | 0.5383 |
| `DiffImpute w/ U-Net` | -0.0030 | 0.0196 | 0.0095 | 0.5255 |

Table 37: Imputation performance comparison in terms of column mask setting, *i.e.* Missing Completely At Random (MCAR), on AL using Pearson correlation. The best results are in **bold**.

| Imputation Methods | 1 | 2 | 3 | 4 |
|---|---|---|---|---|
| Mean Imputation | 0.6796 | 0.6796 | 0.6796 | 0.6796 |
| Median Imputation | 0.6308 | 0.6308 | 0.6308 | 0.6308 |
| Mode Imputation | 0.5513 | 0.5513 | 0.5513 | 0.5513 |
| LOCF Imputation | 0.4617 | 0.4617 | 0.4617 | 0.4617 |
| NOCB Imputation | / | / | / | / |
| MICE (linear) | 0.7555 | 0.8037 | 0.8102 | 0.8228 |
| GAIN | 0.7392 | 0.7910 | 0.8314 | 0.8373 |
| **DiffImpute w/ MLP** | 0.0329 | 0.0131 | 0.0256 | 0.0562 |
| **DiffImpute w/ ResNet** | 0.4733 | 0.4771 | 0.4027 | 0.3236 |
| **DiffImpute w/ Transformer** | **0.8375** | **0.8549** | **0.8666** | **0.8738** |
| **DiffImpute w/ U-Net** | 0.5533 | 0.6889 | 0.6778 | 0.7592 |

Table 38: Imputation performance comparison in terms of column mask setting, *i.e.* Missing Completely At Random (MCAR), on YE using Pearson correlation. The best results are in **bold**.

| Imputation Methods | 1 | 2 | 3 | 4 |
|---|---|---|---|---|
| Mean Imputation | **0.9876** | **0.9876** | 0.9876 | 0.9876 |
| Median Imputation | 0.9875 | 0.9875 | 0.9875 | 0.9875 |
| Mode Imputation | 0.9863 | 0.9863 | 0.9863 | 0.9863 |
| LOCF Imputation | 0.9839 | 0.9839 | 0.9839 | 0.9839 |
| NOCB Imputation | / | / | / | / |
| MICE (linear) | 0.3292 | 0.8135 | **0.9912** | **0.9918** |
| GAIN | 0.0309 | 0.6259 | 0.9925 | 0.9883 |
| **DiffImpute w/ MLP** | -0.0009 | 0.3019 | 0.1618 | 0.1459 |
| **DiffImpute w/ ResNet** | 0.0516 | 0.0828 | -0.2318 | -0.2155 |
| **DiffImpute w/ Transformer** | 0.5469 | 0.9382 | 0.9049 | 0.9478 |
| **DiffImpute w/ U-Net** | 0.0254 | 0.7423 | 0.9545 | 0.9638 |

Table 39: Imputation performance comparison in terms of column mask setting, *i.e.* Missing Completely At Random (MCAR), on CO using Pearson correlation. The best results are in **bold**.

| Imputation Methods | 1 | 2 | 3 | 4 |
|---|---|---|---|---|
| Mean Imputation | **0.7501** | **0.7501** | **0.7501** | **0.7501** |
| Median Imputation | 0.6828 | 0.6828 | 0.6828 | 0.6828 |
| Mode Imputation | 0.6521 | 0.6521 | 0.6521 | 0.6521 |
| LOCF Imputation | 0.5621 | 0.5621 | 0.5621 | 0.5621 |
| NOCB Imputation | / | / | / | / |
| MICE (linear) | NaN | NaN | NaN | NaN |
| GAIN | NaN | NaN | NaN | NaN |
| **DiffImpute w/ MLP** | 0.0121 | 0.1933 | 0.1223 | 0.1786 |
| **DiffImpute w/ ResNet** | 0.1872 | 0.5201 | 0.4335 | 0.6617 |
| **DiffImpute w/ Transformer** | 0.4553 | 0.7481 | 0.6273 | 0.7497 |
| **DiffImpute w/ U-Net** | -0.0028 | 0.1780 | 0.2590 | 0.2288 |

**Pearson Correlation Performance Rankings.** This section presents overall Pearson correlation performance rankings under different mask settings (MCAR, and MAR) across seven datasets, as shown in Tabs. 40 to 42.

Table 40: Overall Pearson correlation rankings under the random mask setting (MCAR). `DiffImpute` with Transformer outperform other methods in six datasets out of seven datasets. The best results are in **bold**.

| Imputation Methods | CA | HE | JA | HI | AL | YE | CO | Mean | Std |
|---|---|---|---|---|---|---|---|---|---|
| Mean Imputation | 2.6 | 3.7 | 2.1 | **1.2** | 4.6 | 2.4 | 2.6 | 2.7 | 1.0 |
| Median Imputation | 4.4 | 4.7 | 3.2 | 3.6 | 6.0 | 3.4 | 4.4 | 4.3 | 0.9 |
| Mode Imputation | 5.9 | 8.7 | 8.8 | 7.1 | 7.3 | 4.6 | 5.4 | 6.8 | 1.5 |
| LOCF Imputation | 7.4 | 6.8 | 6.9 | 7.8 | 8.6 | 6.7 | 7.2 | 7.3 | 0.6 |
| NOCB Imputation | 8.2 | 7.1 | 6.7 | 7.4 | 9.6 | 7.2 | 6.7 | 7.6 | 1.0 |
| MICE | 4.6 | 2.7 | 4.6 | 5.0 | 2.1 | 2.3 | 10.7 | 4.6 | 2.7 |
| GAIN | 7.4 | 6.6 | 7.0 | 6.4 | 5.2 | 8.3 | 5.9 | 6.7 | 0.9 |
| **DiffImpute w/ MLP** | 9.7 | 10.4 | 10.7 | 10.0 | 11.0 | 11.0 | 9.8 | 10.4 | 0.5 |
| **DiffImpute w/ ResNet** | 2.8 | 3.9 | 4.2 | 3.9 | 6.0 | 7.9 | 2.4 | 4.4 | 1.8 |
| **DiffImpute w/ Transformer** | **2.0** | **1.2** | **1.6** | 2.4 | **1.0** | **2.2** | **1.3** | 1.7 | 0.5 |
| **DiffImpute w/ U-Net** | 11.0 | 10.3 | 10.2 | 11.0 | 4.6 | 9.4 | 9.6 | 9.4 | 2.1 |

Table 41: Overall Pearson correlation rankings under the random mask setting (MCAR). `DiffImpute` with Transformer outperform other methods in two datasets out of seven datasets. The mean imputaion methods outperform other methods in five datasets. The best results are in **bold**.

| Imputation Methods | CA | HE | JA | HI | AL | YE | CO | Mean | Std |
|---|---|---|---|---|---|---|---|---|---|
| Mean Imputation | **1.8** | 3.0 | **2.5** | **1.5** | 4.5 | **2.0** | **1.0** | 2.3 | 1.1 |
| Median Imputation | 3.0 | 4.3 | 3.5 | 4.0 | 5.8 | 3.0 | 2.5 | 3.7 | 1.0 |
| Mode Imputation | 4.3 | 7.3 | 6.5 | 5.3 | 7.0 | 4.0 | 3.8 | 5.4 | 1.4 |
| LOCF Imputation | 5.3 | 5.5 | 4.5 | 6.5 | 8.5 | 5.0 | 5.0 | 5.8 | 1.3 |
| NOCB Imputation | / | / | / | / | / | / | / | / | / |
| MICE | 5.3 | 4.3 | 5.5 | 3.5 | 2.5 | 3.8 | NaN | 4.1 | 1.0 |
| GAIN | 5.0 | 3.8 | 3.8 | 4.0 | 2.5 | 4.8 | NaN | 4.0 | 0.8 |
| **DiffImpute w/ MLP** | 7.8 | 9.3 | 9.3 | 8.8 | 10.0 | 9.3 | 7.5 | 8.8 | 0.8 |
| **DiffImpute w/ ResNet** | 8.5 | 6.8 | 7.0 | 9.0 | 8.5 | 9.3 | 5.5 | 7.8 | 1.3 |
| **DiffImpute w/ Transformer** | 4.3 | **1.5** | 3.0 | 4.0 | **1.0** | 6.5 | 3.3 | 3.4 | 1.7 |
| **DiffImpute w/ U-Net** | 10.0 | 9.5 | 9.5 | 8.5 | 4.8 | 7.5 | 7.5 | 8.2 | 1.7 |

Table 42: Overall Pearson correlation rankings of MCAR and MAR (MSE). `DiffImpute` with Transformer outperform other methods in four datasets and the mean imputation methods outperform other methods in three datasets. The best results are in **bold**.

| Imputation Methods | CA | HE | JA | HI | AL | YE | CO | Mean | Std |
|---|---|---|---|---|---|---|---|---|---|
| Mean Imputation | **2.3** | 3.5 | 2.2 | **1.3** | 4.5 | **2.3** | 2.1 | 2.6 | 1.1 |
| Median Imputation | 4.0 | 4.5 | 3.3 | 3.7 | 5.9 | 3.3 | 3.8 | 4.1 | 0.9 |
| Mode Imputation | 5.4 | 8.2 | 8.1 | 6.5 | 7.2 | 4.4 | 4.9 | 6.4 | 1.5 |
| LOCF Imputation | 6.8 | 6.4 | 6.2 | 7.4 | 8.5 | 6.2 | 6.5 | 6.8 | 0.9 |
| NOCB Imputation | 8.2 | 7.1 | 6.7 | 7.4 | 9.6 | 7.2 | 6.7 | 7.6 | 1.0 |
| MICE | 4.8 | 3.2 | 4.8 | 4.5 | 2.2 | 2.8 | 10.7 | 4.7 | 2.8 |
| GAIN | 6.7 | 5.7 | 6.0 | 5.7 | 4.4 | 7.2 | 5.9 | 5.9 | 0.9 |
| `DiffImpute w/ MLP` | 9.1 | 10.1 | 10.2 | 9.6 | 10.7 | 10.5 | 9.1 | 9.9 | 0.6 |
| `DiffImpute w/ ResNet` | 4.5 | 4.8 | 5.1 | 5.5 | 6.8 | 8.3 | 3.4 | 5.5 | 1.6 |
| `DiffImpute w/ Transformer` | 2.7 | **1.3** | **2.0** | 2.9 | **1.0** | 3.5 | **1.9** | 2.2 | 0.9 |
| `DiffImpute w/ U-Net` | 10.7 | 10.1 | 10.0 | 10.2 | 4.6 | 8.8 | 8.9 | 9.1 | 2.1 |

## D.3 Performance on Downstream Tasks.

In this section, we present the performance metrics of downstream tasks for imputed data, considering various missingness mechanisms across our seven benchmark datasets. Specifically, for regression tasks, we employ the root mean squared error (RMSE) as the evaluation metric, while classification tasks are gauged using the accuracy score. Our focus here is on the random mask settings, which align with the Missing Completely At Random (MCAR) setting.

**Random Mask.** Delving deeper into the random mask settings, we evaluate the downstream task performance in the context of the Missing Completely At Random (MCAR). Detailed results for each of the seven datasets are provided in Tabs. 43 to 49.

Table 43: Downstream task performance comparison in random mask setting (MCAR) on CA, evaluated by RMSE. For each missing setting, the best results are in **bold**.

| Imputation Methods | 10% | 20% | 30% | 40% | 50% | 60% | 70% | 80% | 90% |
|---|---|---|---|---|---|---|---|---|---|
| Mean Imputation | 0.8707 | 1.0113 | 1.0974 | 1.1683 | 1.2189 | 1.2532 | 1.2680 | 1.2615 | 1.2461 |
| Median Imputation | 0.8986 | 1.0449 | 1.1319 | 1.2037 | 1.2480 | 1.2753 | 1.2795 | 1.2527 | **1.2150** |
| Mode Imputation | 0.9982 | 1.3324 | 1.3552 | 1.6428 | 1.5582 | 1.6270 | 1.3985 | 1.3580 | 1.2889 |
| 0 Imputation | 1.1661 | 1.4571 | 1.6696 | 1.8366 | 1.9694 | 2.073 | 2.1479 | 2.2096 | 2.2443 |
| 1 Imputation | 1.3509 | 1.6528 | 1.8069 | 1.8886 | 1.9268 | 1.9520 | 1.9805 | 2.0049 | 2.0774 |
| LOCF Imputation | 1.5345 | 1.6405 | 1.6802 | 1.4143 | 1.7231 | 1.7528 | 1.7746 | 1.787 | 1.8204 |
| NOCB Imputation | 1.5317 | 1.6512 | 1.6996 | 1.4195 | 1.7400 | 1.7782 | 1.8056 | 1.8163 | 1.8216 |
| MICE(linear) | 0.7643 | 0.8571 | 0.9543 | 1.0534 | 1.1461 | 1.2349 | 1.3023 | 1.3927 | 1.4240 |
| GAIN | 0.8464 | 0.9473 | 0.9991 | 1.1548 | 1.2405 | 1.3517 | 1.8428 | 2.1072 | 2.2291 |
| `DiffImpute w/ MLP` | 0.9986 | 1.2324 | 1.4155 | 1.5677 | 1.7011 | 1.8234 | 1.9264 | 2.0195 | 2.1030 |
| `DiffImpute w/ ResNet` | 0.7917 | 0.8916 | 0.9637 | 1.0388 | 1.1239 | 1.2563 | 1.5024 | 1.9100 | 2.2878 |
| `DiffImpute w/ Transformer` | **0.7614** | **0.8365** | **0.8951** | **0.9633** | **1.0286** | **1.0874** | **1.1465** | **1.1994** | 1.2527 |
| `DiffImpute w/ U-Net` | 1.2736 | 1.6123 | 1.8475 | 2.0147 | 2.1314 | 2.2267 | 2.2929 | 2.3461 | 2.3812 |

Table 44: Downstream task performance comparison in random mask setting (MCAR) on HE, evaluated by accuracy score, the best results are in **bold**.

| Imputation Methods | 10% | 20% | 30% | 40% | 50% | 60% | 70% | 80% | 90% |
|---|---|---|---|---|---|---|---|---|---|
| Mean Imputation | 0.3172 | 0.2723 | 0.2291 | 0.1874 | 0.1484 | 0.1149 | 0.0866 | 0.0643 | 0.0511 |
| Median Imputation | 0.3160 | 0.2705 | 0.2288 | 0.1874 | 0.1481 | 0.1131 | 0.0832 | 0.0567 | 0.0344 |
| Mode Imputation | 0.2931 | 0.2361 | 0.1877 | 0.1484 | 0.1176 | 0.0914 | 0.0694 | 0.0531 | 0.0412 |
| 0 Imputation | 0.2295 | 0.1584 | 0.1203 | 0.0975 | 0.0810 | 0.0706 | 0.0646 | 0.0606 | 0.0596 |
| 1 Imputation | 0.2238 | 0.1453 | 0.0963 | 0.0692 | 0.0524 | 0.0400 | 0.0323 | 0.0261 | 0.0207 |
| LOCF Imputation | 0.0234 | 0.0266 | 0.0252 | 0.0260 | 0.0256 | 0.0250 | 0.0240 | 0.0240 | 0.0240 |
| NOCB Imputation | 0.0243 | 0.0270 | 0.0262 | 0.0256 | 0.0266 | 0.0246 | 0.0256 | 0.0256 | 0.0256 |
| MICE (linear) | 0.3345 | 0.3083 | 0.2812 | 0.2433 | **0.2036** | **0.1600** | **0.1206** | **0.0875** | 0.0538 |
| GAIN | 0.3246 | 0.2798 | 0.2425 | 0.1968 | 0.1304 | 0.0937 | 0.0747 | 0.0655 | 0.0601 |
| DiffImpute w/ MLP | 0.2695 | 0.2007 | 0.1486 | 0.1115 | 0.0866 | 0.0701 | 0.0579 | 0.0499 | 0.0440 |
| DiffImpute w/ ResNet | 0.3313 | 0.2980 | 0.2621 | 0.2199 | 0.1726 | 0.1266 | 0.0868 | 0.0671 | **0.0610** |
| DiffImpute w/ Transformer | **0.3397** | **0.3145** | **0.2826** | **0.2439** | 0.1986 | 0.1567 | 0.1148 | 0.0780 | 0.0485 |
| DiffImpute w/ U-Net | 0.2531 | 0.1800 | 0.1327 | 0.1036 | 0.0826 | 0.0685 | 0.0578 | 0.0518 | 0.0474 |

Table 45: Downstream task performance comparison in random mask setting (MCAR) on JA, evaluated by accuracy score, the best results are in **bold**.

| Imputation Methods | 10% | 20% | 30% | 40% | 50% | 60% | 70% | 80% | 90% |
|---|---|---|---|---|---|---|---|---|---|
| Mean Imputation | 0.6863 | 0.6547 | 0.6173 | 0.5762 | 0.5307 | 0.4782 | 0.4215 | 0.3579 | 0.2875 |
| Median Imputation | 0.6829 | 0.6497 | 0.6144 | 0.5743 | 0.5279 | 0.4776 | 0.4228 | 0.3693 | 0.3327 |
| Mode Imputation | 0.6577 | 0.6150 | 0.5813 | 0.5532 | 0.5299 | 0.5119 | 0.4983 | 0.4840 | **0.4736** |
| 0 Imputation | 0.6243 | 0.5681 | 0.5342 | 0.5127 | 0.4979 | 0.4867 | 0.4767 | 0.4717 | 0.4664 |
| 1 Imputation | 0.6289 | 0.5728 | 0.5317 | 0.5023 | 0.4816 | 0.4618 | 0.4449 | 0.4285 | 0.4037 |
| LOCF Imputation | 0.3759 | 0.3803 | 0.3839 | 0.3864 | 0.3858 | 0.3904 | 0.3907 | 0.3902 | 0.3935 |
| NOCB Imputation | 0.3766 | 0.3794 | 0.3831 | 0.3847 | 0.3880 | 0.3894 | 0.3921 | 0.3922 | 0.3932 |
| MICE (linear) | 0.6975 | 0.6780 | 0.6578 | 0.6291 | 0.5969 | **0.5699** | **0.5283** | **0.4902** | 0.4397 |
| GAIN | 0.6658 | 0.6803 | 0.6302 | 0.5909 | 0.5436 | 0.5054 | 0.4931 | 0.4697 | 0.4669 |
| DiffImpute w/ MLP | 0.6461 | 0.5903 | 0.5494 | 0.5183 | 0.4972 | 0.4797 | 0.4664 | 0.4585 | 0.4541 |
| DiffImpute w/ ResNet | 0.6905 | 0.6658 | 0.6409 | 0.5183 | 0.5724 | 0.5308 | 0.4937 | 0.4707 | 0.4572 |
| DiffImpute w/ Transformer | **0.6998** | **0.6838** | **0.6624** | **0.6379** | **0.6045** | 0.5637 | 0.5177 | 0.4608 | 0.3970 |
| DiffImpute w/ U-Net | 0.6421 | 0.5881 | 0.5477 | 0.5197 | 0.4973 | 0.4797 | 0.4651 | 0.4586 | 0.4527 |

Table 46: Downstream task performance comparison in random mask setting on HI, evaluated by accuracy score, the best results are in **bold**.

| Imputation Methods | 10% | 20% | 30% | 40% | 50% | 60% | 70% | 80% | 90% |
|---|---|---|---|---|---|---|---|---|---|
| Mean Imputation | 0.6931 | 0.6713 | 0.6515 | 0.6305 | **0.6135** | **0.5950** | **0.5786** | **0.5629** | **0.5463** |
| Median Imputation | 0.6929 | 0.6708 | 0.6506 | 0.6305 | 0.6114 | 0.5907 | 0.5736 | 0.5573 | 0.5430 |
| Mode Imputation | 0.6915 | 0.6670 | 0.6441 | 0.6232 | 0.6026 | 0.5840 | 0.5671 | 0.5528 | 0.5409 |
| 0 Imputation | 0.6823 | 0.6507 | 0.6242 | 0.5984 | 0.5741 | 0.5516 | 0.5276 | 0.5040 | 0.4867 |
| 1 Imputation | 0.6385 | 0.5844 | 0.5447 | 0.5188 | 0.5004 | 0.4872 | 0.4791 | 0.4747 | 0.4724 |
| LOCF Imputation | 0.5014 | 0.4994 | 0.4997 | 0.4976 | 0.5006 | 0.5013 | 0.5017 | 0.5017 | 0.5017 |
| NOCB Imputation | 0.4994 | 0.4978 | 0.4977 | 0.4990 | 0.4986 | 0.4973 | 0.4974 | 0.4974 | 0.4974 |
| MICE (linear) | 0.6890 | 0.6669 | 0.6453 | 0.6114 | 0.5906 | 0.5645 | 0.5480 | 0.5286 | 0.5119 |
| GAIN | 0.6849 | 0.6527 | 0.6280 | 0.6105 | 0.5945 | 0.5544 | 0.5378 | 0.5102 | 0.4874 |
| DiffImpute w/ MLP | 0.6768 | 0.6394 | 0.6120 | 0.5881 | 0.5674 | 0.5483 | 0.5340 | 0.5175 | 0.5050 |
| DiffImpute w/ ResNet | 0.6909 | 0.6664 | 0.6420 | 0.6176 | 0.5917 | 0.5670 | 0.5383 | 0.5044 | 0.4836 |
| DiffImpute w/ Transformer | **0.6979** | **0.6767** | **0.6545** | **0.6340** | 0.6097 | 0.5870 | 0.5652 | 0.5406 | 0.5196 |
| DiffImpute w/ U-Net | 0.6665 | 0.6243 | 0.5922 | 0.5681 | 0.5459 | 0.5284 | 0.5139 | 0.5016 | 0.4939 |

Table 47: Downstream task performance comparison in random mask setting on AL, evaluated by accuracy score, the best results are in **bold**.

| Imputation Methods | 10% | 20% | 30% | 40% | 50% | 60% | 70% | 80% | 90% |
|---|---|---|---|---|---|---|---|---|---|
| Mean Imputation | 0.8002 | 0.6321 | 0.4549 | 0.2964 | 0.1756 | 0.0927 | 0.0421 | 0.0160 | 0.0052 |
| Median Imputation | 0.8325 | 0.7148 | 0.5730 | 0.4247 | 0.2877 | 0.1724 | 0.0891 | 0.0359 | 0.0098 |
| Mode Imputation | 0.8080 | 0.6604 | 0.4953 | 0.3371 | 0.2104 | 0.1155 | 0.0557 | 0.0229 | 0.0072 |
| 0 Imputation | 0.7102 | 0.4903 | 0.3057 | 0.1729 | 0.0915 | 0.0448 | 0.0211 | 0.0092 | 0.0036 |
| 1 Imputation | 0.1194 | 0.0272 | 0.0064 | 0.0021 | 0.0013 | 0.0012 | 0.0011 | 0.0011 | 0.0011 |
| LOCF Imputation | 0.0009 | 0.0009 | 0.0009 | 0.0009 | 0.0009 | 0.0009 | 0.0009 | 0.0009 | 0.0009 |
| NOCB Imputation | 0.0010 | 0.0010 | 0.0010 | 0.0010 | 0.0010 | 0.0010 | 0.0010 | 0.0010 | 0.0010 |
| MICE (linear) | 0.8724 | 0.7969 | 0.6883 | 0.5724 | 0.4309 | **0.2951** | **0.1693** | **0.0788** | **0.0202** |
| GAIN | 0.8724 | 0.7575 | 0.5574 | 0.3936 | 0.2470 | 0.1364 | 0.0551 | 0.0168 | 0.0040 |
| **DiffImpute w/ MLP** | 0.4176 | 0.1748 | 0.0751 | 0.0344 | 0.0169 | 0.0085 | 0.0045 | 0.0029 | 0.0019 |
| **DiffImpute w/ ResNet** | 0.8519 | 0.7366 | 0.5801 | 0.3987 | 0.2309 | 0.1063 | 0.0390 | 0.0125 | 0.0039 |
| **DiffImpute w/ Transformer** | **0.8875** | **0.8301** | **0.7386** | **0.6070** | **0.4427** | 0.2702 | 0.1313 | 0.0469 | 0.0103 |
| **DiffImpute w/ U-Net** | 0.8321 | 0.7061 | 0.5542 | 0.3925 | 0.2477 | 0.1345 | 0.0598 | 0.0221 | 0.0060 |

Table 48: Downstream task performance comparison in random mask setting on YE, evaluated by RMSE, the best results are in **bold**.

| Imputation Methods | 10% | 20% | 30% | 40% | 50% | 60% | 70% | 80% | 90% |
|---|---|---|---|---|---|---|---|---|---|
| Mean Imputation | 9.6483 | 9.9895 | 10.3056 | 10.5864 | 10.8372 | 11.0496 | **11.2184** | **11.3274** | **11.3629** |
| Median Imputation | 9.6279 | 9.9625 | 10.2814 | 10.5704 | 10.8363 | 11.0667 | 11.2547 | 11.3902 | 11.4502 |
| Mode Imputation | 9.7211 | 10.1028 | 10.4576 | 10.7646 | 11.1054 | 11.4239 | 11.7990 | 12.3536 | 13.2784 |
| 0 Imputation | 10.2651 | 10.8614 | 11.2272 | 11.4203 | 11.5104 | 11.5486 | 11.5515 | 11.5434 | 11.5288 |
| 1 Imputation | 10.4652 | 11.0338 | 11.329 | 11.4941 | 11.5855 | 11.6359 | 11.6544 | 11.6536 | 11.6344 |
| LOCF Imputation | 12.4934 | 12.4969 | 12.4953 | 12.5030 | 12.5114 | 12.5117 | 12.4934 | 12.4934 | 12.4934 |
| NOCB Imputation | 12.4883 | 12.4909 | 12.4963 | 12.5015 | 12.5267 | 12.5402 | 12.4883 | 12.4883 | 12.4883 |
| MICE (linear) | 9.9231 | 9.8463 | 10.1061 | 10.4166 | 10.7099 | **11.0431** | 11.3486 | 11.6996 | 11.9950 |
| GAIN | 9.9231 | 9.8463 | 10.8024 | 10.4166 | 11.3067 | 11.5499 | 11.4964 | 11.5453 | 11.5261 |
| **DiffImpute w/ MLP** | 10.2733 | 10.8953 | 11.2566 | 11.4651 | 11.5683 | 11.6109 | 11.6202 | 11.6075 | 11.5891 |
| **DiffImpute w/ ResNet** | 9.6229 | 9.9908 | 10.3924 | 10.8053 | 11.0905 | 11.2565 | 11.4274 | 11.4886 | 11.4806 |
| **DiffImpute w/ Transformer** | **9.5022** | **9.7544** | **10.0342** | **10.3449** | **10.6919** | 11.0639 | 11.4675 | 11.8724 | 12.2635 |
| **DiffImpute w/ U-Net** | 9.8339 | 10.2640 | 10.5960 | 10.8568 | 11.0618 | 11.2840 | 11.5149 | 11.6760 | 11.7223 |

Table 49: Downstream task performance comparison in random mask setting on CO, evaluated by accuracy score, the best results are in **bold**.

| Imputation Methods | 10% | 20% | 30% | 40% | 50% | 60% | 70% | 80% | 90% |
|---|---|---|---|---|---|---|---|---|---|
| Mean Imputation | 0.8379 | 0.7526 | 0.6826 | 0.6252 | 0.5801 | 0.5447 | 0.5172 | 0.4978 | **0.4869** |
| Median Imputation | 0.8397 | 0.7549 | 0.6850 | 0.6280 | 0.5827 | 0.5471 | 0.5206 | 0.5015 | 0.4905 |
| Mode Imputation | 0.8270 | 0.7327 | 0.6549 | 0.5896 | 0.5343 | 0.4877 | 0.4473 | 0.4132 | 0.3853 |
| 0 Imputation | 0.8118 | 0.7020 | 0.6093 | 0.5284 | 0.4587 | 0.3985 | 0.3458 | 0.2982 | 0.2300 |
| 1 Imputation | 0.6544 | 0.5354 | 0.4691 | 0.4253 | 0.3940 | 0.3734 | 0.3633 | 0.3711 | 0.3942 |
| LOCF Imputation | 0.4004 | 0.3872 | 0.3918 | 0.3951 | 0.3979 | 0.4001 | 0.4015 | 0.4035 | 0.4047 |
| NOCB Imputation | 0.4001 | 0.3877 | 0.3927 | 0.3956 | 0.3982 | 0.3994 | 0.4015 | 0.4035 | 0.4043 |
| MICE (linear) | 0.7608 | 0.6504 | 0.5881 | 0.4820 | 0.4332 | 0.3916 | 0.3852 | 0.4534 | 0.3657 |
| GAIN | 0.8502 | 0.7707 | 0.6961 | 0.5760 | 0.4926 | 0.3988 | 0.3396 | 0.3098 | 0.2302 |
| **DiffImpute w/ MLP** | 0.7997 | 0.6870 | 0.6032 | 0.5397 | 0.4905 | 0.4522 | 0.4180 | 0.3898 | 0.3639 |
| **DiffImpute w/ ResNet** | 0.8557 | 0.7796 | 0.7114 | 0.6523 | 0.6008 | 0.5556 | 0.5165 | 0.4889 | 0.4630 |
| **DiffImpute w/ Transformer** | **0.8622** | **0.7904** | **0.7244** | **0.6646** | **0.6144** | **0.5710** | **0.5351** | **0.5031** | 0.4766 |
| **DiffImpute w/ U-Net** | 0.8086 | 0.7027 | 0.6185 | 0.5505 | 0.4963 | 0.4490 | 0.4073 | 0.3700 | 0.3373 |

**Column Mask.** In this section, we assess the imputation performance using the Pearson correlation metric, specifically under the column mask settings. These settings are representative of the Missing at Random (MAR). Our evaluation spans across seven benchmark datasets, as detailed in Tabs. 50 to 56. It's important to highlight that the NOCB imputation method is not applicable in this context, given the absence of a subsequent observation for backfilling missing values.

Table 50: Downstream task performance comparison in column mask setting (MAR) on CA, evaluated by RMSE, the best results are in **bold**.

| Imputation Methods | 1 | 2 | 3 | 4 |
|---|---|---|---|---|
| Mean Imputation | 0.8321 | 0.9880 | 1.2584 | 1.2831 |
| Median Imputation | 0.8474 | 1.0118 | 1.3759 | 1.2288 |
| Mode Imputation | 0.925 | 1.0672 | 1.5293 | 1.2891 |
| 0 Imputation | 0.9295 | 1.7578 | 1.5229 | 1.7217 |
| 1 Imputation | 1.1986 | 1.1815 | 1.8452 | 1.8931 |
| LOCF Imputation | 0.9175 | 1.0747 | 1.5489 | 1.3072 |
| NOCB Imputation | / | / | / | / |
| MICE (linear) | 0.7302 | **0.6850** | 1.2246 | **0.8795** |
| GAIN | 0.7107 | 0.6862 | **0.9819** | 1.1849 |
| `DiffImpute w/ MLP` | 0.8775 | 1.2106 | 1.6318 | 1.6548 |
| `DiffImpute w/ ResNet` | 0.9440 | 1.8283 | 1.5211 | 1.8269 |
| `DiffImpute w/ Transformer` | **0.7228** | 0.7790 | 1.0002 | 1.0263 |
| `DiffImpute w/ U-Net` | 1.0677 | 1.9387 | 1.8962 | 2.0328 |

Table 51: Downstream task performance comparison in column mask setting (MAR) on HE, evaluated by accuracy score, the best results are in **bold**

| Imputation Methods | 1 | 2 | 3 | 4 |
|---|---|---|---|---|
| Mean Imputation | 0.3547 | 0.3279 | 0.2832 | 0.2696 |
| Median Imputation | 0.3550 | 0.3277 | 0.2816 | 0.2681 |
| Mode Imputation | 0.3489 | 0.3141 | 0.2297 | 0.2364 |
| 0 Imputation | 0.3160 | 0.2528 | 0.1727 | 0.1808 |
| 1 Imputation | 0.3428 | 0.2626 | 0.1436 | 0.1376 |
| LOCF Imputation | 0.3546 | 0.3250 | 0.2646 | 0.2667 |
| NOCB Imputation | / | / | / | / |
| MICE (linear) | 0.3576 | 0.3567 | 0.3232 | 0.2657 |
| GAIN | 0.3574 | **0.3571** | **0.3346** | **0.2809** |
| `DiffImpute w/ MLP` | 0.3416 | 0.2945 | 0.2186 | 0.2137 |
| `DiffImpute w/ ResNet` | 0.3340 | 0.2900 | 0.1712 | 0.1888 |
| `DiffImpute w/ Transformer` | **0.3566** | 0.3393 | 0.3199 | 0.3117 |
| `DiffImpute w/ U-Net` | 0.3352 | 0.2561 | 0.2634 | 0.2160 |

Table 52: Downstream task performance comparison in column mask setting (MAR) on JA, evaluated by accuracy score, the best results are in **bold**.

| Imputation Methods | 1 | 2 | 3 | 4 |
|---|---|---|---|---|
| Mean Imputation | 0.7108 | 0.7060 | 0.7005 | 0.6783 |
| Median Imputation | 0.7103 | 0.7056 | 0.7009 | 0.6774 |
| Mode Imputation | 0.7011 | 0.6987 | 0.6857 | 0.6532 |
| 0 Imputation | 0.7100 | 0.6806 | 0.6793 | 0.6158 |
| 1 Imputation | 0.6897 | 0.6862 | 0.6716 | 0.6021 |
| LOCF Imputation | 0.7101 | 0.7056 | 0.6960 | 0.6608 |
| NOCB Imputation | / | / | / | / |
| MICE (linear) | **0.7131** | 0.6706 | 0.7097 | 0.6915 |
| GAIN | 0.7129 | 0.6843 | 0.6980 | **0.6995** |
| **DiffImpute w/ MLP** | 0.7082 | 0.6919 | 0.6908 | 0.6524 |
| **DiffImpute w/ ResNet** | 0.7103 | 0.6781 | 0.6825 | 0.6158 |
| **DiffImpute w/ Transformer** | 0.7123 | **0.7078** | **0.7108** | 0.6937 |
| **DiffImpute w/ U-Net** | 0.7061 | 0.6732 | 0.6755 | 0.6815 |

Table 53: Downstream task performance comparison in column mask setting (MAR) on HI, evaluated by accuracy score, the best results are in **bold**.

| Imputation Methods | 1 | 2 | 3 | 4 |
|---|---|---|---|---|
| Mean Imputation | 0.6964 | **0.6998** | 0.7022 | 0.6914 |
| Median Imputation | 0.6970 | 0.6961 | 0.7006 | 0.6873 |
| Mode Imputation | 0.6961 | 0.6961 | 0.7009 | 0.6856 |
| 0 Imputation | 0.6842 | 0.6832 | **0.7035** | 0.6718 |
| 1 Imputation | 0.6842 | 0.6263 | 0.6367 | 0.5959 |
| LOCF Imputation | 0.6558 | 0.6954 | 0.6918 | 0.6888 |
| NOCB Imputation | / | / | / | / |
| MICE (linear) | 0.6350 | 0.6950 | 0.6840 | **0.6981** |
| GAIN | 0.6473 | 0.6943 | 0.6849 | 0.6898 |
| **DiffImpute w/ MLP** | 0.6764 | 0.6669 | 0.6895 | 0.6544 |
| **DiffImpute w/ ResNet** | 0.6773 | 0.6756 | 0.7030 | 0.6647 |
| **DiffImpute w/ Transformer** | **0.7032** | 0.6989 | 0.7027 | 0.6910 |
| **DiffImpute w/ U-Net** | 0.6726 | 0.6434 | 0.6564 | 0.6562 |

Table 54: Downstream task performance comparison in column mask setting (MAR) on AL, evaluated by accuracy score, the best results are in **bold**.

| Imputation Methods | 1 | 2 | 3 | 4 |
|---|---|---|---|---|
| Mean Imputation | 0.9164 | 0.9045 | 0.9047 | 0.8852 |
| Median Imputation | 0.9167 | 0.905 | 0.9052 | 0.8820 |
| Mode Imputation | 0.9167 | 0.9023 | 0.9052 | 0.8830 |
| 0 Imputation | 0.9167 | 0.8954 | 0.9052 | 0.8458 |
| 1 Imputation | 0.7638 | 0.5757 | 0.4265 | 0.3502 |
| LOCF Imputation | 0.9167 | 0.8247 | 0.8547 | 0.7762 |
| NOCB Imputation | / | / | / | / |
| MICE (linear) | 0.9108 | 0.9003 | 0.9116 | 0.9000 |
| GAIN | 0.9157 | 0.8958 | 0.9121 | 0.9011 |
| **DiffImpute w/ MLP** | 0.8783 | 0.8264 | 0.7922 | 0.7469 |
| **DiffImpute w/ ResNet** | 0.9161 | 0.8753 | 0.8917 | 0.8200 |
| **DiffImpute w/ Transformer** | **0.9177** | 0.9124 | **0.9146** | **0.9047** |
| **DiffImpute w/ U-Net** | 0.9161 | **0.9141** | 0.8978 | 0.8879 |

Table 55: Downstream task performance comparison in column mask setting (MAR) on YE, evaluated by RMSE (MAR), the best results are in **bold**.

| Imputation Methods | 1 | 2 | 3 | 4 |
|---|---|---|---|---|
| Mean Imputation | 9.2610 | 9.4197 | 9.3024 | 9.4945 |
| Median Imputation | 9.2610 | 9.3982 | 9.2909 | 9.4762 |
| Mode Imputation | 9.2610 | 9.3931 | 9.2842 | 9.4635 |
| 0 Imputation | 9.2610 | 9.6935 | 9.3141 | 10.1599 |
| 1 Imputation | 9.2606 | 10.1696 | 9.6535 | 10.2094 |
| LOCF Imputation | 9.2610 | 9.4576 | 9.2906 | 9.6248 |
| NOCB Imputation | / | / | / | / |
| MICE (linear) | 9.261 | 9.4699 | **9.2610** | **9.3314** |
| GAIN | 9.261 | 9.5965 | **9.2610** | 9.3885 |
| **DiffImpute w/ MLP** | **9.2609** | 9.9062 | 9.4708 | 10.2741 |
| **DiffImpute w/ ResNet** | 9.2611 | 9.6901 | 9.3116 | 10.1554 |
| **DiffImpute w/ Transformer** | **9.2609** | 9.3298 | 9.2727 | 9.4193 |
| **DiffImpute w/ U-Net** | **9.2609** | **9.2640** | 9.4906 | 9.3764 |

Table 56: Downstream task performance comparison in column mask setting (MAR) on CO, evaluated by accuracy score, the best results are in **bold**.

| Imputation Methods | 1 | 2 | 3 | 4 |
|---|---|---|---|---|
| Mean Imputation | 0.8919 | 0.8890 | 0.8187 | 0.7491 |
| Median Imputation | 0.8951 | 0.8924 | 0.8257 | 0.7610 |
| Mode Imputation | 0.8799 | 0.8875 | 0.8099 | 0.7271 |
| 0 Imputation | 0.8784 | 0.8807 | 0.8064 | 0.7159 |
| 1 Imputation | 0.8370 | 0.7896 | 0.6767 | 0.6398 |
| LOCF Imputation | 0.8939 | 0.8717 | 0.8223 | 0.7630 |
| NOCB Imputation | / | / | / | / |
| MICE (linear) | NaN | NaN | NaN | NaN |
| GAIN | NaN | NaN | NaN | NaN |
| **DiffImpute w/ MLP** | 0.8836 | 0.8703 | 0.8077 | 0.7247 |
| **DiffImpute w/ ResNet** | 0.8938 | 0.8882 | 0.8233 | 0.7564 |
| **DiffImpute w/ Transformer** | **0.8988** | 0.8962 | 0.8318 | 0.7745 |
| **DiffImpute w/ U-Net** | 0.8870 | **0.9281** | **0.8746** | **0.7861** |

**Downstream Tasks Performance Rankings.** This section presents overall downstream tasks performance rankings under different mask settings (MCAR, and MAR) across seven datasets (Tabs. 57 and 58).

Table 57: Downstream task performance comparison under the random mask setting (MCAR) across the seven datasets. As different datasets apply different metrics, we report the performance rankings as the measurement. DiffImpute with Transformer has the best overall performance, the best results are in **bold**.

| Imputation Methods | CA | HE | JA | HI | AL | YE | CO | Mean | Std |
|---|---|---|---|---|---|---|---|---|---|
| Mean Imputation | 3.8 | 4.7 | 7.7 | **1.4** | 7.6 | **3.0** | 4.0 | 4.6 | 2.1 |
| Median Imputation | 5.3 | 6.1 | 8.1 | 2.3 | 3.6 | 3.0 | **3.0** | 4.5 | 1.9 |
| Mode Imputation | 6.4 | 7.3 | 4.8 | 3.8 | 6.0 | 8.1 | 5.3 | 6.0 | 1.4 |
| 0 Imputation | 10.7 | 8.3 | 7.6 | 8.6 | 9.0 | 7.8 | 10.0 | 8.8 | 1.1 |
| 1 Imputation | 10.8 | 11.2 | 10.0 | 12.0 | 11.0 | 10.0 | 10.9 | 10.8 | 0.6 |
| LOCF Imputation | 8.4 | 12.7 | 12.0 | 11.0 | 13.0 | 12.6 | 10.1 | 11.4 | 1.5 |
| NOCB Imputation | 9.2 | 12.1 | 12.1 | 12.0 | 12.0 | 12.2 | 10.0 | 11.4 | 1.1 |
| MICE | 3.1 | 1.8 | **2.3** | 5.4 | 1.6 | 4.1 | 9.6 | 4.0 | 2.6 |
| GAIN | 6.3 | 4.4 | 4.0 | 7.1 | 4.9 | 5.7 | 7.7 | 5.7 | 1.3 |
| **DiffImpute w/ MLP** | 8.7 | 8.4 | 7.7 | 8.2 | 10.0 | 9.0 | 8.2 | 8.6 | 0.7 |
| **DiffImpute w/ ResNet** | 4.8 | 2.8 | 4.0 | 6.6 | 5.8 | 4.6 | 2.8 | 4.5 | 1.3 |
| **DiffImpute w/ Transformer** | **1.2** | **2.0** | 2.7 | 2.4 | **1.4** | 3.7 | **1.3** | 2.1 | 0.8 |
| **DiffImpute w/ U-Net** | 12.2 | 9.0 | 7.9 | 10.0 | 5.2 | 7.0 | 7.9 | 8.5 | 2.1 |

Table 58: Downstream task performance comparison under the column mask setting (MAR) across the seven datasets. As different datasets apply different metrics, we report the performance rankings as the measurement. `DiffImpute` with Transformer has the best overall performance, the best results are in **bold**.

| Imputation Methods | CA | HE | JA | HI | AL | YE | CO | Mean | Std |
|---|---|---|---|---|---|---|---|---|---|
| Mean Imputation | 4.3 | 4.0 | 3.8 | 2.5 | 5.5 | 6.0 | 5.3 | 4.5 | 1.1 |
| Median Imputation | 4.8 | 4.5 | 4.3 | 4.3 | 4.0 | 5.3 | 3.0 | 4.3 | 0.6 |
| Mode Imputation | 7.0 | 7.3 | 8.0 | 4.8 | 4.3 | 4.3 | 7.0 | 6.1 | 1.5 |
| 0 Imputation | 8.8 | 11.3 | 9.3 | 5.5 | 5.5 | 8.5 | 8.5 | 8.2 | 1.9 |
| 1 Imputation | 10.5 | 10.5 | 10.8 | 10.3 | 12.0 | 9.0 | 10.0 | 10.4 | 0.8 |
| LOCF Imputation | 7.5 | 5.8 | 5.8 | 6.8 | 8.3 | 6.0 | 4.8 | 6.4 | 1.1 |
| NOCB Imputation | / | | / | / | / | / | | / | / |
| MICE | 2.0 | 2.8 | 4.5 | 7.3 | 5.5 | 3.5 | NaN | 4.3 | 1.8 |
| GAIN | **1.8** | **1.5** | 4.0 | 7.8 | 5.0 | 4.3 | NaN | 4.0 | 2.1 |
| **DiffImpute w/ MLP** | 8.3 | 8.8 | 7.8 | 9.3 | 10.8 | 8.8 | 8.0 | 8.8 | 0.9 |
| **DiffImpute w/ ResNet** | 9.3 | 10.3 | 8.5 | 6.8 | 8.5 | 9.5 | 4.5 | 8.2 | 1.8 |
| **DiffImpute w/ Transformer** | 2.3 | 2.5 | **1.8** | **2.3** | **1.3** | **2.8** | **1.8** | 2.1 | 0.5 |
| **DiffImpute w/ U-Net** | 11.8 | 9.0 | 9.0 | 10.3 | 5.0 | 4.0 | 2.3 | 7.3 | 3.3 |

### D.4 Time Performance.

**Training Time.** In the subsequent tables, we present the training durations associated with various denoising models employed in our study. Notably, these durations exclude the time taken for `Harmonization` and `Impute-DDIM` processes. All time measurements are provided in seconds, as detailed in Tab. 59.

Table 59: The training time performance, measured in seconds, reveals that the U-Net model exhibits the longest training duration.

| Methods | CA | HE | JA | HI | AL | YE | CO |
|---|---|---|---|---|---|---|---|
| **DiffImpute w/ MLP** | 16 | 58 | 54 | 78 | 72 | 343 | 488 |
| **DiffImpute w/ ResNet** | 26 | 92 | 82 | 122 | 107 | 526 | 743 |
| **DiffImpute w/ Transformer** | 88 | 295 | 267 | 404 | 386 | 1762 | 2428 |
| **DiffImpute w/ U-Net** | 267 | 926 | 856 | 1252 | 1180 | 5555 | 7572 |

**Inference Time.** The subsequent tables detail the inference durations for the various denoising models incorporated in our research. It's noteworthy to mention that, based on the `Harmonization` algorithm (as seen in code snippet. 2), the inference time for models utilizing `Harmonization` witnessed an approximately fivefold increase. All durations are quantified in seconds, as elaborated in Tab. 60.

Table 60: The inference time performance, measured in seconds, reveals that the U-Net model exhibits the longest training duration.

| Methods | CA | HE | JA | HI | AL | YE | CO |
|---|---|---|---|---|---|---|---|
| **DiffImpute w/ MLP** | 3 | 9 | 19 | 12 | 13 | 36 | 71 |
| **DiffImpute w/ ResNet** | 4 | 12 | 24 | 15 | 16 | 42 | 89 |
| **DiffImpute w/ Transformer** | 11 | 74 | 298 | 107 | 553 | 677 | 913 |
| **DiffImpute w/ U-Net** | 27 | 157 | 869 | 236 | 959 | 1827 | 2519 |

## D.5 ABLATION RESULTS WITHOUT TIME STEP TOKENIZER.

This section demonstrates the ablation results after excluding the `Time Step Tokenizer`. The evaluations are specifically conducted under various missingness mechanisms, focusing on the CA dataset.

**Random Mask.**   Below, we present tables detailing the imputation outcomes under random mask settings. These outcomes are quantified using three metrics: mean squared error (MSE), Pearson correlation, and performance on downstream tasks. The respective results can be referenced in Tabs. 61 to 62.

Table 61: Imputation MSE performance comparison without `Time Step Tokenizer` in random mask (MCAR) setting on CA. The best results are in **bold**.

| Imputation Methods | 10% | 20% | 30% | 40% | 50% | 60% | 70% | 80% | 90% |
|---|---|---|---|---|---|---|---|---|---|
| MLP w/o Time Step Tokenizer | 0.0173 | 0.0187 | 0.0199 | 0.0212 | 0.0226 | 0.0238 | 0.0251 | 0.0263 | 0.0275 |
| ResNet w/o Time Step Tokenizer | **0.0157** | **0.0171** | **0.0184** | **0.0198** | **0.0220** | 0.0255 | 0.0321 | 0.0448 | 0.0658 |
| Transformer w/o Time Step Tokenizer | 0.0169 | 0.0184 | 0.0199 | 0.0210 | 0.0224 | 0.0236 | 0.0250 | 0.0264 | 0.0277 |
| U-Net w/o Time Step Tokenizer | 0.0176 | 0.0189 | 0.0200 | 0.0212 | 0.0224 | **0.0234** | **0.0245** | **0.0257** | **0.0266** |

Table 62: Pearson correlation performance comparison without `Time Step Tokenizer` in random mask (MCAR) setting on CA. The best results are in **bold**.

| Imputation Methods | 10% | 20% | 30% | 40% | 50% | 60% | 70% | 80% | 90% |
|---|---|---|---|---|---|---|---|---|---|
| MLP w/o Time Step Tokenizer | 0.8515 | 0.8379 | 0.8284 | 0.8167 | 0.8035 | 0.7920 | 0.7797 | 0.7678 | 0.7569 |
| ResNet w/o Time Step Tokenizer | **0.8648** | **0.8527** | **0.8426** | **0.8332** | **0.8180** | **0.7984** | 0.7602 | 0.6794 | 0.5192 |
| Transformer w/o Time Step Tokenizer | 0.8531 | 0.8389 | 0.8268 | 0.8174 | 0.8041 | 0.7931 | 0.7790 | 0.7651 | 0.7527 |
| U-Net w/o Time Step Tokenizer | 0.8493 | 0.8372 | 0.8286 | 0.8188 | 0.8074 | 0.7981 | **0.7865** | **0.7756** | **0.7661** |

Table 63: Downstream task performance comparison without `Time Step Tokenizer` in random mask (MCAR) setting on CA, evaluated by RMSE. The best results are in **bold**.

| Imputation Methods | 10% | 20% | 30% | 40% | 50% | 60% | 70% | 80% | 90% |
|---|---|---|---|---|---|---|---|---|---|
| MLP w/o Time Step Tokenizer | 0.7916 | 0.8922 | 0.9683 | 1.0452 | 1.1141 | 1.1766 | 1.2294 | 1.2723 | 1.3099 |
| ResNet w/o Time Step Tokenizer | 0.7909 | 0.8914 | 0.9656 | 1.0409 | 1.1169 | 1.2139 | 1.3766 | 1.6312 | 1.8705 |
| Transformer w/o Time Step Tokenizer | **0.7844** | **0.8816** | **0.9588** | **1.0334** | **1.1041** | **1.1665** | 1.2242 | 1.2687 | 1.3095 |
| U-Net w/o Time Step Tokenizer | 0.7975 | 0.8994 | 0.9713 | 1.0449 | 1.1101 | 1.1680 | **1.2166** | **1.2536** | **1.2892** |

**Column Mask.**   Below, we present tables detailing the imputation outcomes under random mask settings. These outcomes are quantified using three metrics: mean squared error (MSE), Pearson correlation, and performance on downstream tasks. The respective results can be referenced in Tabs. 64 to 66.

Table 64: Imputation performance comparison without `Time Step Tokenizer` in column mask (MAR) setting on CA, evaluated by MSE. The best results are in **bold**.

| Imputation Methods | 1 | 2 | 3 | 4 |
|---|---|---|---|---|
| MLP w/o Time Step Tokenizer | 0.0196 | 0.0223 | 0.0198 | 0.0112 |
| ResNet w/o Time Step Tokenizer | 0.0741 | 0.0951 | 0.0914 | 0.0722 |
| Transformer w/o Time Step Tokenizer | 0.0191 | 0.0224 | 0.0193 | 0.0106 |
| U-Net w/o Time Step Tokenizer | 0.2000 | 0.0180 | 0.0268 | 0.0205 |

## D.6 ABLATION RESULTS OF HARMONIZATION.

This section delves into the imputation efficacy of four distinct denoising models when integrated with the `Harmonization` technique. The evaluations are specifically conducted under various missingness mechanisms, focusing on the CA dataset.

Table 65: Pearson correlation performance comparison without `Time Step Tokenizer` in column mask (MAR) setting on CA. The best results are in **bold**.

| Imputation Methods | 1 | 2 | 3 | 4 |
|---|---|---|---|---|
| **MLP w/o `Time Step Tokenizer`** | 0.1728 | 0.5812 | 0.7376 | 0.8899 |
| **ResNet w/o `Time Step Tokenizer`** | 0.1983 | 0.3260 | -0.0072 | 0.3305 |
| **Transformer w/o `Time Step Tokenizer`** | 0.1908 | 0.5899 | 0.7426 | 0.8977 |
| **U-Net w/o `Time Step Tokenizer`** | 0.1658 | 0.5232 | 0.7896 | 0.7782 |

Table 66: Downstream task performance comparison without `Time Step Tokenizer` in column mask (MAR) setting on CA, evaluated by RMSE. The best results are in **bold**.

| Imputation Methods | 1 | 2 | 3 | 4 |
|---|---|---|---|---|
| **MLP w/o `Time Step Tokenizer`** | 0.7566 | 0.8494 | 1.0102 | 1.1316 |
| **ResNet w/o `Time Step Tokenizer`** | 0.9282 | 1.7319 | 1.5977 | 1.5334 |
| **Transformer w/o `Time Step Tokenizer`** | 0.7498 | 0.8399 | 1.0759 | 1.0995 |
| **U-Net w/o `Time Step Tokenizer`** | 0.7637 | 0.9363 | 0.9413 | 1.1991 |

**Random Mask.** Below, we present tables detailing the imputation outcomes under random mask settings. These outcomes are quantified using three metrics: mean squared error (MSE), Pearson correlation, and performance on downstream tasks. The respective results can be referenced in Tabs. 67 to 69.

Table 67: Imputation MSE performance comparison with `Harmonization` in random mask (MCAR) setting on CA. The best results are in **bold**.

| Imputation Methods | 10% | 20% | 30% | 40% | 50% | 60% | 70% | 80% | 90% |
|---|---|---|---|---|---|---|---|---|---|
| `Harmonization w/ MLP` | 0.0253 | 0.0258 | 0.0265 | 0.0268 | 0.0274 | 0.0280 | 0.0285 | 0.0292 | 0.0298 |
| `Harmonization w/ ResNet` | **0.0146** | **0.0157** | **0.0169** | **0.0178** | **0.0189** | **0.0198** | **0.0208** | **0.0218** | **0.0229** |
| `Harmonization w/ Transformer` | 0.0155 | 0.0168 | 0.0180 | 0.0191 | 0.0206 | 0.0219 | 0.0232 | 0.0246 | 0.0258 |
| `Harmonization w/ U-Net` | 2.0681 | 2.6099 | 3.1769 | 3.9142 | 4.7691 | 5.6382 | 6.6880 | 7.9535 | 9.2977 |

Table 68: Pearson correlation performance comparison with `Harmonization` in random mask (MCAR) setting on CA. The best results are in **bold**.

| Imputation Methods | 10% | 20% | 30% | 40% | 50% | 60% | 70% | 80% | 90% |
|---|---|---|---|---|---|---|---|---|---|
| `Harmonization w/ MLP` | 0.7817 | 0.7774 | 0.7747 | 0.7733 | 0.7694 | 0.7655 | 0.7614 | 0.7569 | 0.7533 |
| `Harmonization w/ ResNet` | 0.8752 | 0.8645 | 0.8554 | **0.8474** | **0.8373** | **0.8287** | **0.8184** | **0.8085** | **0.7986** |
| `Harmonization w/ Transformer` | **0.8772** | **0.8662** | **0.8566** | 0.8473 | 0.8352 | 0.8240 | 0.8115 | 0.7994 | 0.7883 |
| `Harmonization w/ U-Net` | 0.0781 | 0.0726 | 0.0683 | 0.0677 | 0.0663 | 0.0668 | 0.0671 | 0.0652 | 0.656 |

Table 69: Downstream task performance comparison with `Harmonization` in MCAR setting on CA, evaluated by RMSE. The best results are in **bold**.

| Imputation Methods | 10% | 20% | 30% | 40% | 50% | 60% | 70% | 80% | 90% |
|---|---|---|---|---|---|---|---|---|---|
| `Harmonization w/ MLP` | 0.8692 | 1.0101 | 1.1407 | 1.1852 | 1.2500 | 1.3100 | 1.3547 | 1.3843 | 1.4057 |
| `Harmonization w/ ResNet` | 0.7679 | 0.8574 | 0.9255 | 1.0000 | 1.0723 | 1.1369 | 1.2031 | 1.2612 | 1.3190 |
| `Harmonization w/ Transformer` | **0.7486** | **0.8162** | **0.8705** | **0.9335** | **0.9943** | **1.0509** | **1.1076** | **1.1657** | **1.2264** |
| `Harmonization w/ U-Net` | 1.1727 | 1.4774 | 1.6634 | 1.7959 | 1.8834 | 1.9391 | 1.9825 | 2.0146 | 2.0634 |

**Column Mask.** In the subsequent tables, we detail the imputation outcomes when operating under column mask settings. These results are gauged using three pivotal metrics: mean squared error (MSE), Pearson correlation, and efficacy on downstream tasks. For a comprehensive understanding, refer to Tabs. 70 to 72.

Table 70: Imputation performance comparison with `Harmonization` in column mask (MAR) setting on CA, evaluated by MSE. The best results are in **bold**.

| Imputation Methods | 1 | 2 | 3 | 4 |
|---|---|---|---|---|
| **Harmonization w/ MLP** | 0.02660 | 0.0296 | 0.0264 | 0.0189 |
| **Harmonization w/ ResNet** | 0.0184 | 0.0203 | 0.0173 | 0.0095 |
| **Harmonization w/ Transformer** | **0.0173** | **0.0202** | **0.0164** | **0.0084** |
| **Harmonization w/ U-Net** | 2.1512 | 0.1604 | 2.5408 | 4.2775 |

Table 71: Pearson correlation performance comparison with `Harmonization` in column mask (MAR) setting on CA. The best results are in **bold**.

| Imputation Methods | 1 | 2 | 3 | 4 |
|---|---|---|---|---|
| **Harmonization w/ MLP** | 0.0929 | 0.5159 | 0.6368 | 0.8193 |
| **Harmonization w/ ResNet** | 0.2462 | 0.6083 | 0.7690 | 0.9112 |
| **Harmonization w/ Transformer** | **0.4130** | **0.6877** | **0.8064** | **0.9286** |
| **Harmonization w/ U-Net** | 0.1795 | 0.3662 | 0.1771 | 0.0948 |

Table 72: Downstream task performance comparison with `Harmonization` in column mask (MAR) setting on CA, evaluated by RMSE. The best results are in **bold**.

| Imputation Methods | 1 | 2 | 3 | 4 |
|---|---|---|---|---|
| **Harmonization w/ MLP** | 0.8175 | 0.9961 | 1.2466 | 1.2839 |
| **Harmonization w/ ResNet** | 0.7557 | 0.8718 | 1.0723 | 1.0908 |
| **Harmonization w/ Transformer** | **0.7111** | **0.7647** | **0.9425** | **0.9991** |
| **Harmonization w/ U-Net** | 0.9452 | 1.6025 | 1.4419 | 1.9054 |

### D.7 ABLATION RESULTS OF IMPUTE-DDIM.

The tables below display the experimental results of imputation performance using the `Impute-DDIM` technique on the CA dataset, with the retraced step set to $j = 5$ and $\tau \in \{10, 25, 50, 100, 250\}$.

**Random Mask.** The tables below shows the imputation performance with `Impute-DDIM` as evaluated by mean squared error (MSE) setting $\tau \in \{10, 25, 50, 100, 250\}$, under the random mask settings (Tabs. 73 to 77).

Table 73: Imputation performance comparison with `Impute-DDIM` setting $\tau = 10$ under the random mask (MCAR) setting on CA, evaluated by MSE. The best results are in **bold**.

| Imputation Methods | 10% | 20% | 30% | 40% | 50% | 60% | 70% | 80% | 90% |
|---|---|---|---|---|---|---|---|---|---|
| **Impute-DDIM w/ MLP** | 0.2725 | 0.2775 | 0.2807 | 0.2814 | 0.2825 | 0.2829 | 0.2835 | 0.2842 | 0.2849 |
| **Impute-DDIM w/ ResNet** | 0.2483 | 0.2539 | 0.2580 | **0.2594** | **0.2608** | **0.2615** | **0.2623** | **0.2633** | **0.2640** |
| **Impute-DDIM w/ Transformer** | **0.2438** | **0.2511** | **0.2571** | 0.2602 | 0.2634 | 0.2657 | 0.2677 | 0.2699 | 0.2718 |
| **Impute-DDIM w/ U-Net** | 0.2678 | 0.2719 | 0.2748 | 0.2752 | 0.2759 | 0.2760 | 0.2762 | 0.2766 | 0.2771 |

Table 74: Imputation performance comparison with `Impute-DDIM` setting $\tau = 25$ under the random mask (MCAR) setting on CA, evaluated by MSE. The best results are in **bold**.

| Imputation Methods | 10% | 20% | 30% | 40% | 50% | 60% | 70% | 80% | 90% |
|---|---|---|---|---|---|---|---|---|---|
| Impute-DDIM w/ MLP | 0.2301 | 0.2354 | 0.2398 | 0.2417 | 0.2437 | 0.2452 | 0.2467 | 0.2483 | 0.2499 |
| Impute-DDIM w/ ResNet | 0.1763 | 0.1822 | 0.1876 | 0.1904 | 0.1927 | 0.1946 | **0.1962** | **0.1980** | **0.1997** |
| Impute-DDIM w/ Transformer | **0.1601** | **0.1692** | **0.1774** | **0.1834** | **0.1890** | **0.1937** | 0.1980 | 0.2024 | 0.2064 |
| Impute-DDIM w/ U-Net | 0.2191 | 0.2236 | 0.2268 | 0.2279 | 0.2289 | 0.2296 | 0.2302 | 0.2311 | 0.2321 |

Table 75: Imputation performance comparison with `Impute-DDIM` setting $\tau = 50$ under the random mask (MCAR) setting on CA, evaluated by MSE. The best results are in **bold**.

| Imputation Methods | 10% | 20% | 30% | 40% | 50% | 60% | 70% | 80% | 90% |
|---|---|---|---|---|---|---|---|---|---|
| Impute-DDIM w/ MLP | 0.1778 | 0.1832 | 0.1881 | 0.1911 | 0.1940 | 0.1964 | 0.1990 | 0.2014 | 0.2039 |
| Impute-DDIM w/ ResNet | 0.1027 | 0.1077 | 0.1129 | 0.1163 | 0.1192 | 0.1217 | 0.1240 | 0.1264 | 0.1285 |
| Impute-DDIM w/ Transformer | **0.0801** | **0.0867** | **0.0934** | **0.0992** | **0.1049** | **0.1103** | **0.1152** | **0.1204** | **0.1253** |
| Impute-DDIM w/ U-Net | 0.1638 | 0.1673 | 0.1701 | 0.1720 | 0.1734 | 0.1750 | 0.1760 | 0.1774 | 0.1788 |

Table 76: Imputation performance comparison with `Impute-DDIM` setting $\tau = 100$ under the random mask (MCAR) setting on CA, evaluated by MSE. The best results are in **bold**.

| Imputation Methods | 10% | 20% | 30% | 40% | 50% | 60% | 70% | 80% | 90% |
|---|---|---|---|---|---|---|---|---|---|
| Impute-DDIM w/ MLP | 0.1135 | 0.1175 | 0.1224 | 0.1259 | 0.1291 | 0.1324 | 0.1358 | 0.1390 | 0.1420 |
| Impute-DDIM w/ ResNet | 0.0443 | 0.0466 | 0.0495 | 0.0518 | 0.0541 | 0.0560 | 0.0579 | 0.0599 | 0.0617 |
| Impute-DDIM w/ Transformer | **0.0281** | **0.0303** | **0.0329** | **0.0351** | **0.0375** | **0.0399** | **0.0423** | **0.0451** | **0.0477** |
| Impute-DDIM w/ U-Net | 0.1064 | 0.1091 | 0.1114 | 0.1131 | 0.1147 | 0.1165 | 0.1180 | 0.1199 | 0.1218 |

Table 77: Imputation performance comparison with `Impute-DDIM` setting $\tau = 250$ under the in random mask (MCAR) setting on CA, evaluated by MSE. The best results are in **bold**.

| Imputation Methods | 10% | 20% | 30% | 40% | 50% | 60% | 70% | 80% | 90% |
|---|---|---|---|---|---|---|---|---|---|
| Impute-DDIM w/ MLP | 0.0492 | 0.0512 | 0.0537 | 0.0555 | 0.0576 | 0.0596 | 0.0617 | 0.0641 | 0.0661 |
| Impute-DDIM w/ ResNet | 0.0210 | 0.0219 | 0.0230 | 0.0238 | 0.0248 | 0.0257 | **0.0266** | 0.0276 | 0.0285 |
| Impute-DDIM w/ Transformer | **0.0152** | **0.0165** | **0.0180** | **0.0191** | **0.0205** | **0.0215** | 0.0277 | **0.0240** | **0.0251** |
| Impute-DDIM w/ U-Net | 0.0748 | 0.0758 | 0.0777 | 0.0794 | 0.0808 | 0.0824 | 0.0845 | 0.0870 | 0.0900 |

**Column Mask.** The tables below shows the imputation performance with `Impute-DDIM` setting $\tau \in \{10, 25, 50, 100, 250\}$, as evaluated by mean squared error (MSE) under column mask settings (Tabs. 78 to 82).

Table 78: Imputation performance comparison with `Impute-DDIM` setting $\tau = 10$ under the column mask (MAR) setting on CA, evaluated by MSE. The best results are in **bold**.

| Imputation Methods | 1 | 2 | 3 | 4 |
|---|---|---|---|---|
| Impute-DDIM w/ MLP | 0.2770 | 0.2922 | 0.2715 | 0.2581 |
| Impute-DDIM w/ ResNet | 0.2557 | 0.2707 | 0.2505 | **0.2381** |
| Impute-DDIM w/ Transformer | **0.2438** | **0.2635** | **0.2477** | 0.2391 |
| Impute-DDIM w/ U-Net | 0.2732 | 0.2472 | 0.3016 | 0.2704 |

Table 79: Imputation performance comparison with `Impute-DDIM` setting $\tau = 25$ under the column mask (MAR) setting on CA, evaluated by MSE. The best results are in **bold**.

| Imputation Methods | 1 | 2 | 3 | 4 |
|---|---|---|---|---|
| Impute-DDIM w/ MLP | 0.2333 | 0.2471 | 0.2347 | 0.2195 |
| Impute-DDIM w/ ResNet | 0.1854 | 0.1978 | 0.1858 | 0.1731 |
| Impute-DDIM w/ Transformer | **0.1572** | **0.1758** | **0.1714** | **0.1667** |
| Impute-DDIM w/ U-Net | 0.2244 | 0.2067 | 0.2461 | 0.2302 |

Table 80: Imputation performance comparison with `Impute-DDIM` setting $\tau = 50$ under the column mask (MAR) setting on CA, evaluated by MSE. The best results are in **bold**.

| Imputation Methods | 1 | 2 | 3 | 4 |
|---|---|---|---|---|
| Impute-DDIM w/ MLP | 0.1791 | 0.1908 | 0.1874 | 0.1708 |
| Impute-DDIM w/ ResNet | 0.1128 | 0.1216 | 0.1157 | 0.1036 |
| Impute-DDIM w/ Transformer | **0.0791** | **0.0889** | **0.0916** | **0.0861** |
| Impute-DDIM w/ U-Net | 0.1679 | 0.1602 | 0.1815 | 0.1817 |

Table 81: Imputation performance comparison with `Impute-DDIM` setting $\tau = 100$ under the column mask (MAR) setting on CA, evaluated by MSE. The best results are in **bold**.

| Imputation Methods | 1 | 2 | 3 | 4 |
|---|---|---|---|---|
| **Impute-DDIM w/ MLP** | 0.1133 | 0.1216 | 0.1255 | 0.1096 |
| **Impute-DDIM w/ ResNet** | 0.0506 | 0.0547 | 0.0529 | 0.0425 |
| **Impute-DDIM w/ Transformer** | **0.0312** | **0.0334** | **0.0329** | **0.0225** |
| **Impute-DDIM w/ U-Net** | 0.1064 | 0.1098 | 0.1163 | 0.1250 |

Table 82: Imputation performance comparison with `Impute-DDIM` setting $\tau = 250$ under the column mask (MAR) setting on CA, evaluated by MSE. The best results are in **bold**.

| Imputation Methods | 1 | 2 | 3 | 4 |
|---|---|---|---|---|
| **Impute-DDIM w/ MLP** | 0.0492 | 0.0536 | 0.0555 | 0.0452 |
| **Impute-DDIM w/ ResNet** | 0.0236 | 0.0262 | 0.0238 | 0.0154 |
| **Impute-DDIM w/ Transformer** | **0.0177** | **0.0205** | **0.0168** | **0.0085** |
| **Impute-DDIM w/ U-Net** | 0.0622 | 0.0772 | 0.0851 | 0.0770 |

## D.8    INFERENCE TIME ABLATION STUDY.

In the subsequent tables, we present the inference durations associated with four distinct denoising networks. Specifically, we focus on the impact of integrating the `Harmonization` and `Impute-DDIM` techniques on the CA dataset.

**Impact of `Harmonization` on Inference Time.**    The table that follows delineates the inference durations for four denoising networks when the `Harmonization` technique is employed with a retraced step of $j = 5$. For a comprehensive understanding, we also provide a comparative analysis against scenarios where the `Harmonization` technique is not utilized (Tab. 83).

Table 83: Ablation of inference time comparison for `Harmonization`. The inference time is about five times longer when employing the `Harmonization` technique, which aligns with our algorithm 2. Time is measured in seconds.

| Technique | MLP | ResNet | Transformer | U-Net |
|---|---|---|---|---|
| w/o `Harmonization` | 3 | 4 | 27 | 11 |
| `Harmonization` | 15 | 19 | 53 | 29 |

**`Impute-DDIM` Inference Time.**   The table below illustrates the inference time of four denoising networks when utilizing the `Impute-DDIM` technique, with $\tau$ sequentially taking values from $10, 25, 50, 100, 250, 500$. The retraced step $j$ remains fixed at 5 in this context. Time is measured in seconds (Tab. 84).

Table 84: Imputation performance comparison with `Impute-DDIM` in random mask setting on CA, measured in seconds. Note that when $\tau = 500$, no `Impute-DDIM` is applied.

| Imputation Methods | $\tau = 10$ | $\tau = 25$ | $\tau = 50$ | $\tau = 100$ | $\tau = 250$ | $\tau = 500$ |
|---|---|---|---|---|---|---|
| **Impute-DDIM w/ MLP** | 2 | 1 | 2 | 3 | 8 | 15 |
| **Impute-DDIM w/ ResNet** | 1 | 1 | 2 | 4 | 10 | 19 |
| **Impute-DDIM w/ Transformer** | 1 | 2 | 5 | 11 | 26 | 53 |
| **Impute-DDIM w/ U-Net** | 1 | 7 | 15 | 30 | 74 | 149 |

D.9    COMPARISON RESULTS WITH MIWAE (VAE-BASED METHOD).

**Random Mask.**   In the subsequent tables, we present the imputation results when employing the MIWAE method (Mattei & Frellsen, 2019), a VAE-based approach, gauged using MSE under random mask conditions. This evaluation spans five datasets, specifically Tabs. 85 to 87. It's worth noting that our experiments with MIWAE were confined to the CA, HE, JA, HI, and AL datasets. This limitation arises from the memory-intensive nature of the MIWAE method. Despite utilizing high-end GPUs like the NVIDIA A100, MIWAE often results in memory errors, underscoring its significant memory demands.

Table 85: Imputation performance in terms of random mask setting (MCAR), using the MIWAE method, evaluated with MSE and downstream task metrics across five datasets. According to the experimental results from Tabs. 9 to 15, MIWAE method is inferior to `DiffImpute` in most of the mask settings.

| Dataset | 10% | 20% | 30% | 40% | 50% | 60% | 70% | 80% | 90% |
|---|---|---|---|---|---|---|---|---|---|
| CA | 0.0228 | 0.0233 | 0.0233 | 0.0231 | 0.0234 | 0.0236 | 0.0235 | 0.0234 | 0.0235 |
| HE | 0.0414 | 0.0413 | 0.0405 | 0.0395 | 0.0385 | 0.0372 | 0.0373 | 0.0346 | 0.0352 |
| JA | 0.0388 | 0.0395 | 0.0430 | 0.0402 | 0.0412 | 0.0390 | 0.0380 | 0.0369 | 0.0350 |
| HI | 0.0631 | 0.0629 | 0.0628 | 0.0628 | 0.0629 | 0.0629 | 0.0628 | 0.0628 | 0.0627 |
| AL | 0.0199 | 0.0199 | 0.0199 | 0.0199 | 0.0200 | 0.0200 | 0.0200 | 0.0200 | 0.0200 |

Table 86: Imputation performance in terms of random mask setting (MCAR), using the MI-WAE method, evaluated with Pearson correlation and downstream task metrics across five datasets.According to the experimental results from Tabs. 26 to 32, MIWAE method is inferior to `DiffImpute` in most of the mask settings.

| Dataset | 10% | 20% | 30% | 40% | 50% | 60% | 70% | 80% | 90% |
|---------|-----|-----|-----|-----|-----|-----|-----|-----|-----|
| CA | 0.7995 | 0.7962 | 0.7950 | 0.7957 | 0.7938 | 0.7926 | 0.7924 | 0.7942 | 0.7940 |
| HE | 0.6857 | 0.6861 | 0.6895 | 0.6954 | 0.7008 | 0.7087 | 0.7079 | 0.7247 | 0.7191 |
| JA | 0.6501 | 0.6450 | 0.6253 | 0.6402 | 0.6349 | 0.6461 | 0.6510 | 0.6569 | 0.6714 |
| HI | 0.5762 | 0.5762 | 0.5810 | 0.5807 | 0.5805 | 0.5811 | 0.5817 | 0.5821 | 0.5825 |
| AL | 0.6399 | 0.6402 | 0.6400 | 0.6408 | 0.6400 | 0.6399 | 0.6393 | 0.6393 | 0.6391 |

Table 87: Imputation performance in terms of random mask setting (MCAR), using the MI-WAE method, evaluated with downstream task metrics and downstream task metrics across five datasets. According to the experimental results from Tabs. 43 to 49, MIWAE method is inferior to `DiffImpute` in most of the mask settings.

| Dataset | 10% | 20% | 30% | 40% | 50% | 60% | 70% | 80% | 90% |
|---------|-----|-----|-----|-----|-----|-----|-----|-----|-----|
| CA | 0.8768 | 1.0215 | 1.1207 | 1.2059 | 1.2682 | 1.3132 | 1.3511 | 1.3535 | 1.3535 |
| HE | 0.3017 | 0.2489 | 0.2036 | 0.1625 | 0.1306 | 0.1048 | 0.0791 | 0.0574 | 0.0370 |
| JA | 0.6792 | 0.6423 | 0.6088 | 0.5766 | 0.5428 | 0.5054 | 0.4717 | 0.4302 | 0.3858 |
| HI | 0.6934 | 0.6683 | 0.6451 | 0.6224 | 0.6006 | 0.5815 | 0.5597 | 0.5437 | 0.5241 |
| AL | 0.8210 | 0.6897 | 0.5375 | 0.3893 | 0.2558 | 0.1477 | 0.0735 | 0.0284 | 0.0081 |

**Column Mask.** In the following tables, we detail the imputation results using the MIWAE method, a VAE-based approach, assessed by the mean squared error (MSE) under column mask conditions. This assessment encompasses five datasets, as referenced in Tabs. 88 to 90.

Table 88: Imputation performance in terms of column mask setting (MAR), using the MIWAE method, evaluated with MSE across five datasets. According to the experimental results from Tabs. 16 to 22, MIWAE method is inferior to `DiffImpute` in most of the mask settings.

| Dataset | 1 | 2 | 3 | 4 |
|---------|-----|-----|-----|-----|
| CA | 0.0658 | 0.0007 | 0.0067 | 0.0112 |
| HE | 0.0008 | 0.0148 | 0.0324 | 0.0627 |
| JA | 0.0308 | 0.0386 | 0.0571 | 0.0286 |
| HI | 0.0022 | 0.0036 | 0.0339 | 0.0968 |
| AL | 0.0242 | 0.0487 | 0.0192 | 0.0192 |

Table 89: Imputation performance in terms of column mask setting (MAR), using the MIWAE method, evaluated with Pearson correlation across five datasets. According to the experimental results from Tabs. 33 to 39, MIWAE method is inferior to `DiffImpute` in most of the mask settings.

| Dataset | 1 | 2 | 3 | 4 |
|---------|-----|-----|-----|-----|
| CA | 0.2132 | 0.0073 | 0.7670 | 0.8795 |
| HE | -0.0152 | 0.8110 | 0.4187 | 0.3196 |
| JA | -0.0016 | 0.1886 | 0.4394 | 0.6895 |
| HI | 0.0129 | 0.0356 | 0.7108 | 0.4627 |
| AL | -0.0039 | 0.3794 | 0.1198 | 0.5804 |

Table 90: Imputation performance in terms of column mask setting (MAR), using the MIWAE method, evaluated with downstream task metrics across five datasets. According to the experimental results from Tabs. 50 to 56, MIWAE method is inferior to `DiffImpute` in most of the mask settings.

| Dataset | 1 | 2 | 3 | 4 |
|---------|--------|--------|--------|--------|
| CA | 0.7122 | 0.6853 | 1.0053 | 1.2930 |
| HE | 0.3571 | 0.3570 | 0.3065 | 0.2556 |
| JA | 0.7130 | 0.6845 | 0.6699 | 0.6951 |
| HI | 0.6566 | 0.6882 | 0.6834 | 0.6794 |
| AL | 0.9126 | 0.8780 | 0.8977 | 0.8887 |