# OpenReview forum: "DiffImpute: Tabular Data Imputation With Denoising Diffusion Probabilistic Model"
_ICLR.cc/2024/Conference — ICLR 2024 Conference Withdrawn Submission_

### Official Review · Reviewer_kcNk · 2023-10-31

**Soundness:** 1 poor
**Presentation:** 2 fair
**Contribution:** 1 poor
**Rating:** 3
**Confidence:** 4

**Summary:**

The authors introduce a method for imputation they term DiffImpute, based on DDPM.  The authors consider four different denoising architectures: MLP, ResNet, and Transformer, and U-Net, and compare their imputation methodology to standard baselines including mean imputation and MICE (multivariate imputation by chained equations).

**Strengths:**

The proposed methodology is an interesting idea and the results upon using a Transformer seem to indicate superior performance to standard baselines in the selected datasets.

**Weaknesses:**

The empirical evaluation and comparison of techniques is not thorough enough for any conclusions to be drawn, and I do not feel that this paper is ready for publication.  Even the empirical results given are not convincing that this is a reasonable method and improves in a meaningful way over far simpler baselines. In more detail:

- The authors do not state what the downstream task is/are in Section 4.2.  This makes it very difficult to interpret Table 1.  In particular, it is not clear to me how significant the gains are over the (much) simpler mean imputation.

- Related to the above point, I am not sure why in the empirical evaluation, the authors are restricting themselves to MCAR and MAR data for the experiments.  If this were a theoretical study, these are reasonable and tractable models under which one may hope to give guarantees.  However, in order to actually evaluate your imputation methods, it would be much stronger and more compelling to pick datasets which themselves have missing data and apply your methods.  For instance, one could think of a supervised setting in which the labels are fully observed and the data contains missing entries, and train a sequence of models using different imputation strategies, subsequently comparing the test error.

**Questions:**

- Could the authors please clarify what they mean by “dominant normal distributions and scant tail densities” and why this yields better performance for mean imputation?

- Could the authors please specify what the downstream task used in the experiments was?

- Do the authors have experimental results in which the missingness was not synthetic?

---

### Official Review · Reviewer_Y6jJ · 2023-11-01

**Soundness:** 2 fair
**Presentation:** 2 fair
**Contribution:** 2 fair
**Rating:** 5
**Confidence:** 5

**Summary:**

The author proposes a diffusion-based imputation method for tabular data. They tailor four architectures for handling the tabular features, propose a resample framework for enhancing the coherence between observed and imputed data, and propose to adopt DDIM for sampling. The author conducts experiments on seven dataset to demonstrate the effectiveness of DiffImpute.

**Strengths:**

The authors conduct a careful exploration of which architectures, MLP, ResNet, Transformer, and U-Net, provide the best performance on diffusion models for modeling tabular data. They also adopt three evaluation criteria for the comparison, which is better than previous work on imputation that only uses MSE as the evaluation criteria.

**Weaknesses:**

1. The overall contribution of this paper is limited.

All of the content except the transformer conditioning architecture is already known. The architecture design is heuristic, which has no theoretical guarantees of the performance. Moreover, they build upon Variance Preserving (VP) SDE (e.g., DDPM or TabDDPM in tabular data). The author does not mention whether their method work for Variance Exploding (VE) SDE (e.g, Score-based generative model, StaSy [1] in tabular data).
[1]: Kim, J., Lee, C.E., & Park, N. STaSy: Score-based Tabular data Synthesis. ICLR 2023.

2. Overclaiming the contribution of transformer conditioning architecture.

* Diffusion model can work on imputation together with generation (conditional generation) without the proposed transformer conditioning architecture. They are well-studied in the literature [1,2].

[1]: Tashiro, Y., Song, J., Song, Y., & Ermon, S. CSDI: Conditional Score-based Diffusion Models for Probabilistic Time Series Imputation. NIPS 2021.

[2]: Ouyang, Y., Xie, L., Li, C., & Cheng, G. (2023). MissDiff: Training Diffusion Models on Tabular Data with Missing Values. ArXiv, abs/2307.00467.

3. The effectiveness of the proposed method is not well supported.
* : The standard evaluation of imputation performance is the mean squared error of imputed value against oracle value instead of the efficiency criterion used in paragraph "Machine Learning efficiency - Data imputation". Otherwise, it faces the problem of "when the generative model needs to fill in the most significant feature or a feature that has a minimal impact on XGBoost output" as mentioned in the paper. If the authors adopt the traditional evaluation on this task, many designs in this paragraph will not be needed.

* : To evaluate the performance of TabGenDDPM on imputation tasks, it should be compared with other imputation methods, e.g., [3,4], rather than only compared with TabDDPM.

[3]: Yoon, J., Jordon, J., & Schaar, M.V. GAIN: Missing Data Imputation using Generative Adversarial Nets. ICML 2018.

[4]: Mattei, P., & Frellsen, J. MIWAE: Deep Generative Modelling and Imputation of Incomplete Data Sets. ICML 2019.


* : The author should compare with other diffusion based model on tabular data, e.g., StaSy [1]. Also, some discussion and experimental results of whether transformer conditioning can be developed on Variance Exploding (VE) SDE.

* : The of illumination the experimental setup should be clarified. Currently, it brings some confusion.
- The baseline in Figure 3 stands for which method? In my point of view, it is not the methods mentioned in section 5.2.
- The Table 4 is confusing. In my point of view, three different evaluation criteria have different properties, i.e., the smaller the correlation is, the better the performance is, which is different with privacy risk. Why do the authors use Up arrow/Down arrow beside the name of the dataset. It is also not clear why the authors only report the experimental results on six datasets rather than eight datasets in Table 2.
- It would be helpful to have the performance on each dataset for Table 3 in the appendix.

4. Minor

The paper has many typos, e.g.,
- adding period for the caption of Table 1, 3, 4, and Figure 3;
- what is the meaning of "4+2" and "2(4+40)" in Table 1;
- "in this situation, the generative model can employ the no-missing values to condition the missing data generation." is hard to understand.

**Questions:**

Previous works on imputation tasks, e.g., [1,2,3], train their model on the data containing missing values, which is important for real applications. However, DiffImpute is trained on complete tabular datasets. Why does the author adopt the CSDI [3] architecture for imputation on tabular data? CSDI [3] is also a diffusion based model for imputation tasks. Then, their model can train their model on the data containing missing values.

[1]: Mattei, Pierre-Alexandre and Jes Frellsen. “MIWAE: Deep Generative Modelling and Imputation of Incomplete Data Sets.” International Conference on Machine Learning (2019).
[2]: Yoon, Jinsung et al. “GAIN: Missing Data Imputation using Generative Adversarial Nets.” ArXiv abs/1806.02920 (2018): n. pag.
[3]: Tashiro, Yusuke et al. “CSDI: Conditional Score-based Diffusion Models for Probabilistic Time Series Imputation.” ArXiv abs/2107.03502 (2021): n. pag.

---

### Official Review · Reviewer_gUrr · 2023-11-08

**Soundness:** 2 fair
**Presentation:** 2 fair
**Contribution:** 2 fair
**Rating:** 3
**Confidence:** 2

**Summary:**

The authors propose a DDPM for missingness imputation. They propose 1) a novel time step tokeniser to embed temporal information, 2) harmonization to align synthetically generated tabular entries with observed data, and 3) a modification of DDIM for faster inference. Empirically, they explore 4 different denoising network architectures.

**Strengths:**

1. The paper proposed two modifications of DDPM+DDIM that can lead to improved imputation results.
2. The experimental setting also considers a number of denoising networks.

**Weaknesses:**

1. The works closely related to the paper, most prominently Zheng & Charoenphakdee (2022), but also Tashiro et al. (2021) and Ouyang et al., 2023) are not sufficiently discussed and compared to.
2. The presentation is unclear in a couple of parts. 3. For instance, $x_{r-1+j}$ is not being re-used in the algorithm (alg 1). Also it is not clear how the time step tokenizer is related to tabular data imputation, and how the change was motivated. The presentation of the contributions could be improved by highlighting in the algorithms which parts are new. There could additionally be information on the size of the data sets.
3. The experimental section might be misleading. A) results are averaged across percentage ranges from 10%-90%. 2) Std error per experiment is not presented. 3) It is not clear what LOCF and NOCB imputation are. 4) No other deep learning approaches than GAIN are compared to. 4) It is not clear how the hyperparameters have been tuned. They might have been chosen in a way to make harmonization and TST overperform.

**Questions:**

1. Is there a typo in Alg 1 or is  $x_{r-1+j}$ not being reused?
2. How were the hyperparameters chosen?
3. What do LOCF and NOCB stand for?

---

### Official Review · Reviewer_eAqk · 2023-11-11

**Soundness:** 2 fair
**Presentation:** 1 poor
**Contribution:** 2 fair
**Rating:** 5
**Confidence:** 4

**Summary:**

The authors present an imputation method for tabular data via employing Denoising Diffusion Probabilistic Model (DDPM). In order to do so they train DPMM on observed data and then impute the missing values at the denoising step. In order for the model to enhance consistency during the sampling stage, they use harmonisation via infusing the data back several time points and denoising them multiple times. The authors present four variants of denoising networks, namely, spanning MLP, ResNet, Transformer, and U-Net in order to handle tabular data. Finally, they present empirical results comparing their method to others for various datasets.

**Strengths:**

-- a diffusion model based imputation framework tailored for tabular data;
-- introduction of harmonisation during the imputation process through infusing the data back and denoising them multiple times during the sampling stage

**Weaknesses:**

- lack of theoretical justification on why the algorithm is expected to impute data from an unbiased distribution and/or converge to the stationary distribution
- the method is not compared to the state of the art GAN- and diffusion model- based methods namely MisGAN (Li et al 2019) - PBiGAN (Li et al. Learning from Irregularly-Sampled Time Series: A Missing Data Perspective. ICML, 2020)  and TabCSDI (Zheng et al 2023);
- acronyms of competing methods "LOCF Imputation" and "NOCB Imputation" presented in the results sections are not defined inthe main paper;

**Questions:**

- what do the authors mean by saying that the method works for MCAR and MAR? Under what assumptios?
- what is the method's performance under MNAR?
- at section 4.2, Table 1 it is said that the authors "report the performance rankings as the measurement" which is an unclear statement: what is meant by this? Also, why is this metric sufficient and representative ?
- could sensitivity analysis of the method be provided?